# Spatial epigenome–transcriptome co-profiling of mammalian tissues

Di Zhang[1,21], Yanxiang Deng[1,2,20,21 ✉], Petra Kukanja[3], Eneritz Agirre[3], Marek Bartosovic[3], Mingze Dong[4,5], Cong Ma[6], Sai Ma[7], Graham Su[1,2], Shuozhen Bao[1], Yang Liu[1,2], Yang Xiao[8], Gorazd B. Rosoklija[9,10,11], Andrew J. Dwork[9,10,11,12], J. John Mann[9,10,13], Kam W. Leong[8,14], Maura Boldrini[9,10], Liya Wang[15], Maximilian Haeussler[16], Benjamin J. Raphael[6], Yuval Kluger[4,5,17], Gonçalo Castelo-Branco[3,18 ✉] & Rong Fan[1,2,4,19 ✉]

Emerging spatial technologies, including spatial transcriptomics and spatial epigenomics, are becoming powerful tools for profiling of cellular states in the tissue context[1–5]. However, current methods capture only one layer of omics information at a time, precluding the possibility of examining the mechanistic relationship across the central dogma of molecular biology. Here, we present two technologies for spatially resolved, genome-wide, joint profiling of the epigenome and transcriptome by cosequencing chromatin accessibility and gene expression, or histone modifications (H3K27me3, H3K27ac or H3K4me3) and gene expression on the same tissue section at near-single-cell resolution. These were applied to embryonic and juvenile mouse brain, as well as adult human brain, to map how epigenetic mechanisms control transcriptional phenotype and cell dynamics in tissue. Although highly concordant tissue features were identified by either spatial epigenome or spatial transcriptome we also observed distinct patterns, suggesting their differential roles in defining cell states. Linking epigenome to transcriptome pixel by pixel allows the uncovering of new insights in spatial epigenetic priming, differentiation and gene regulation within the tissue architecture. These technologies are of great interest in life science and biomedical research.

Single-cell multiomics allows uncovering of the mechanisms of gene regulation across different omics layers[6–9] but lacks spatial information, which is crucial to understanding cellular function in tissue. Spatial epigenomics, transcriptomics and proteomics emerged recently[1–5] but most of these can capture only one layer of the omics information. Although computational methods can integrate data from multiple omics[10], they cannot readily uncover the mechanistic links between different omics layers. Previously we developed deterministic barcoding in tissue for spatial omics sequencing for spatially resolved comeasurement of the transcriptome and a panel of proteins[1], and recently it was also realized in the 10X Visium platform[11]. Herein, to further investigate the epigenetic mechanisms underlying gene expression regulation, we developed spatially resolved, genome-wide comapping of the epigenome and transcriptome by simultaneous profiling of chromatin accessibility and messenger RNA expression (spatial assay for transposase-accessible chromatin and RNA using sequencing (spatial ATAC–RNA-seq)), or histone modifications and

mRNA expression (spatial assay of cleavage under targets and tagmentation and RNA using sequencing (spatial CUT&Tag–RNA-seq); applied to H3K27me3, H3K27ac or H3K4me3 histone modifications) on the same tissue section at the cellular level via deterministic cobarcoding, to integrate the chemistry of spatial-ATAC-seq[3] or spatial-CUT&Tag[2] with that for spatial transcriptomics. We applied these techniques to embryonic and juvenile mouse brain, as well as adult human brain hippocampus, to dissect epigenetic and transcriptional states and their role in the regulation of cell types dynamics in tissue. This work opens new frontiers in spatial omics and may provide unprecedented opportunities to biological and biomedical research.

## Technology workflow and data quality

Spatial ATAC–RNA-seq is shown schematically in Fig. 1a and Extended Data Fig. 1a–c. A frozen tissue section was fixed with formaldehyde and treated with Tn5 transposition complex preloaded with a DNA

[1]Department of Biomedical Engineering, Yale University, New Haven, CT, USA. [2]Yale Stem Cell Center and Yale Cancer Center, Yale School of Medicine, New Haven, CT, USA. [3]Laboratory of Molecular Neurobiology, Department of Medical Biochemistry and Biophysics, Karolinska Institutet, Stockholm, Sweden. [4]Department of Pathology, Yale University School of Medicine, New Haven, CT, USA. [5]Interdepartmental Program in Computational Biology and Bioinformatics, Yale University, New Haven, CT, USA. [6]Department of Computer Science, Princeton University, Princeton, NJ, USA. [7]Klarman Cell Observatory, Broad Institute of MIT and Harvard, Cambridge, MA, USA. [8]Department of Biomedical Engineering, Columbia University, New York, NY, USA. [9]Department of Psychiatry, Columbia University, New York, NY, USA. [10]Division of Molecular Imaging and Neuropathology, New York State Psychiatric Institute, New York, NY, USA. [11]Macedonian Academy of Sciences & Arts, Skopje, Republic of Macedonia. [12]Department of Pathology and Cell Biology, Columbia University, New York, NY, USA. [13]Department of Radiology, Columbia University, New York, NY, USA. [14]Department of Systems Biology, Columbia University Irving Medical Center, New York, NY, USA. [15]AtlasXomics, Inc., New Haven, CT, USA. [16]Genomics Institute, University of California Santa Cruz, Santa Cruz, CA, USA. [17]Applied Mathematics Program, Yale University, New Haven, CT, USA. [18]Ming Wai Lau Centre for Reparative Medicine, Stockholm Node, Karolinska Institutet, Stockholm, Sweden. [19]Human and Translational Immunology Program, Yale School of Medicine, New Haven, CT, USA. [20]Present address: Department of Pathology and Laboratory Medicine, Epigenetics Institute, Perelman School of Medicine, University of Pennsylvania, Philadelphia, PA, USA. [21]These authors contributed equally: Di Zhang, Yanxiang Deng. ✉e-mail: yanxiang.deng@pennmedicine.upenn.edu; goncalo.castelo-branco@ki.se; rong.fan@yale.edu

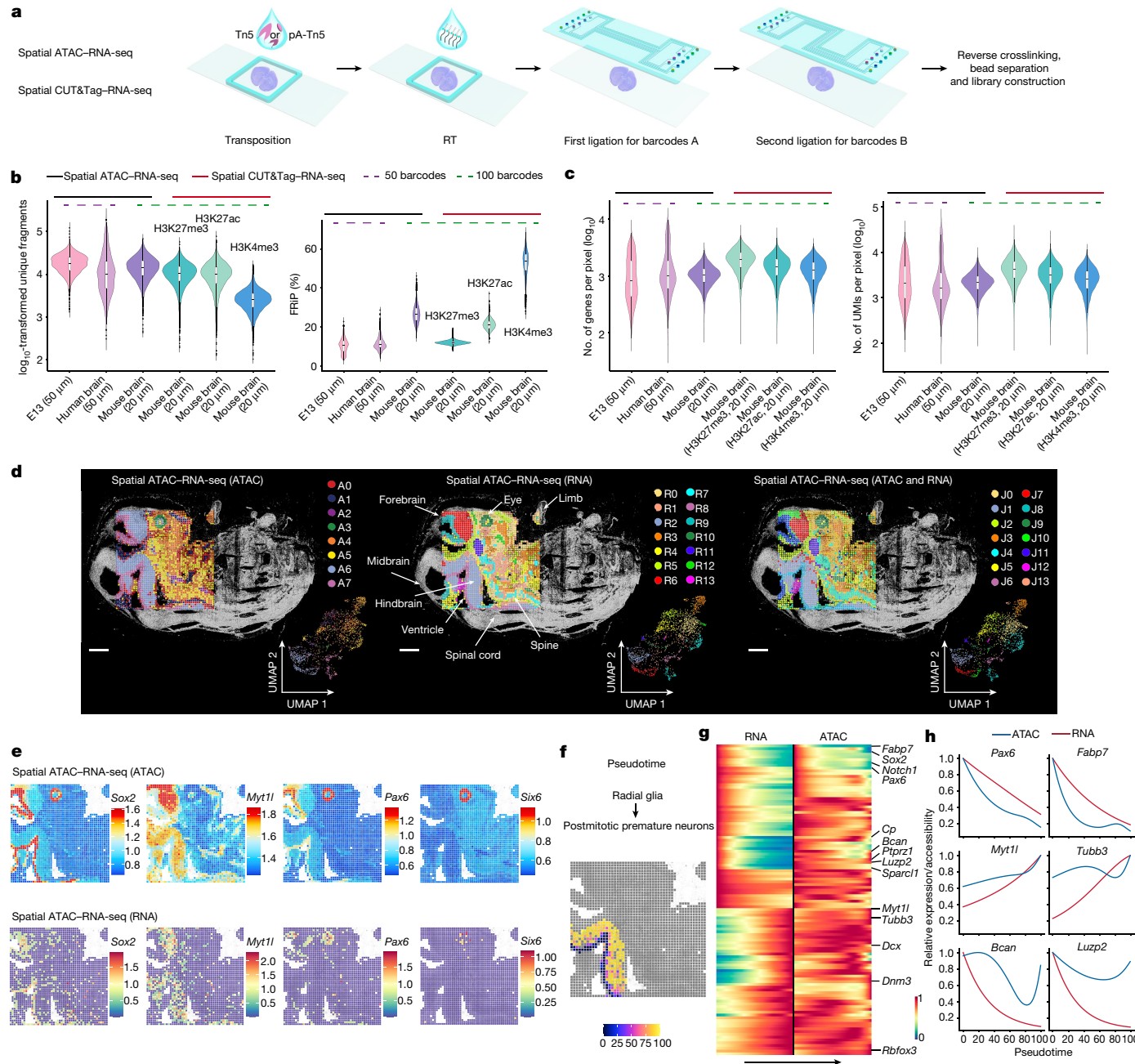

**Fig. 1 | Design and evaluation of spatial epigenome–transcriptome cosequencing with E13 mouse embryo. a**, Schematic workflow. **b**, Comparison of number of unique fragments and fraction of reads in peaks (FRiP) in spatial ATAC–RNA-seq and spatial CUT&Tag–RNA-seq. **c**, Gene and UMI count distribution in spatial ATAC–RNA-seq and spatial CUT&Tag–RNA-seq. Number of pixels in E13, 2,187; in human brain, 2,500; in mouse brain (ATAC), 9,215; in mouse brain (H3K27me3), 9,752; in mouse brain (H3K27ac), 9,370; in mouse brain (H3K4me3), 9,548. Box plots show the median (centre line), the first and third quartiles (box limits) and 1.5× interquartile range (whiskers). **d**, Spatial distribution and UMAP of all clusters for ATAC, RNA and joint clustering of ATAC and RNA data. Overlay of clusters with the tissue image shows that spatial clusters precisely match anatomic regions. Pixel size, 50 µm; scale bars, 1 mm. **e**, Spatial mapping of GAS and gene expression for selected marker genes in different clusters for ATAC and RNA in spatial ATAC–RNA-seq. **f**, Pseudotime analysis from radial glia to postmitotic premature neurons visualized at the spatial level. **g**, Heatmaps delineating gene expression and GAS for marker genes. **h**, Dynamic changes in GAS and gene expression across pseudotime.

adaptor containing a universal ligation linker that can be inserted into transposase-accessible genomic DNA loci. The same tissue section was then incubated with a biotinylated DNA adaptor containing a universal ligation linker and a poly-T sequence that binds to the poly-A trail of mRNAs to initiate reverse transcription (RT) in tissue. A microfluidic channel array chip was then placed on the tissue section to introduce spatial barcodes Ai (i = 1–50 or 100) that were covalently conjugated to the universal ligation linker via templated ligation. Next,

another microchip with microchannels perpendicular to the first flow direction was used to introduce spatial barcodes Bj (j = 1–50 or 100) that were then ligated to barcodes Ai (i = 1–50 or 100), resulting in a two-dimensional grid of spatially barcoded tissue pixels, each defined by the unique combination of barcodes Ai and Bj (i = 1–50 or 100, j = 1–50 or 100; barcoded pixels, n = 2,500 or 10,000). Finally, barcoded complementary DNA and genomic DNA fragments were released after reverse crosslinking. cDNAs were enriched with streptavidin beads and gDNA

fragments were retained in the supernatant. The libraries of gDNA and cDNA were constructed separately for next-generation sequencing (NGS). Spatial CUT&Tag–RNA-seq was performed by applying an antibody against a specific histone modification (H3K27me3, H3K27ac or H3K4me3) to the tissue section and then using a protein A-tethered Tn5-DNA complex to perform CUT&Tag. The remaining steps were similar to those for spatial ATAC–RNA-seq (Fig. 1a and Extended Data Fig. 1a,d,e), but resulted in spatial co-profiling of genome-wide histone modification occupancy and transcriptome.

We performed spatial ATAC–RNA-seq experiments on embryonic day 13 (E13) mouse embryo (pixel size, 50 μm), mouse postnatal day 21/22 (P21/22) brains (pixel size, 20 μm) and adult human brain hippocampus tissue (pixel size, 50 μm). For the 50-μm pixel size we obtained an average of 25 cells for E10 mouse embryo[1] and between one and nine cells for human hippocampus[3]. Most 20-μm pixels contained one to three cells per pixel in juvenile mouse brain (Supplementary Fig. 1b). Using a 100 × 100 barcode scheme, the mapping area covers nearly the entire hemisphere of a P22 mouse brain coronal section. From this sample we obtained a median of 14,284 unique fragments per pixel, of which 19% were enriched in the transcription start site regions and 26% located in peaks (Fig. 1b and Supplementary Figs. 1a and 2a). For the RNA portion a total of 22,914 genes were detected, with an average of 1,073 genes and 2,358 unique molecular identifiers (UMIs) per pixel (spatial ATAC–RNA-seq) (Fig. 1c). We performed spatial CUT&Tag–RNA-seq for H3K27me3, H3K27ac and H3K4me3 on mouse P21/22 brains (pixel size, 20 μm). With the 100 × 100 barcode device we obtained a median of 10,644 (H3K27me3), 10,002 (H3K27ac) and 2,507 (H3K4me3) unique fragments per pixel, of which 12% (H3K27me3), 17% (H3K27ac) and 67% (H3K4me3) overlapped with transcription start site regions and 12% (H3K27me3), 21% (H3K27ac) and 54% (H3K4me3) were located in peaks, respectively (Fig. 1b and Supplementary Figs. 1a and 2a). For the RNA data, 25,881 (H3K27me3), 23,415 (H3K27ac) and 22,731 (H3K4me3) genes were detected with an average of 2,011 (H3K27me3), 1,513 (H3K27ac) and 1,329 (H3K4me3) genes per pixel (4,734 (H3K27me3), 3,580 (H3K27ac) and 2,885 (H3K4me3) UMIs per pixel, respectively (Fig. 1c)). Assessment of data quality for mouse embryo, P21 mouse brain and human brain samples with 50 × 50 barcodes is also included in Fig. 1b,c, Supplementary Figs. 1a and 2a–c and Supplementary Tables 2 and 3.

Moreover, the insert size distributions of chromatin accessibility (spatial ATAC–RNA-seq) and histone modification (spatial CUT&Tag–RNA-seq, H3K27me3, H3K27ac or H3K4me3) fragments were consistent with the captured nucleosomal fragments in all tissues (Supplementary Fig. 1c,d). Correlation analysis between replicates showed high reproducibility ($r = 0.98$ for ATAC, $r = 0.98$ for RNA in spatial ATAC–RNA-seq, $r = 0.96$ for CUT&Tag (H3K27ac), $r = 0.89$ for RNA in spatial CUT&Tag–RNA-seq; $r$ denotes Pearson correlation coefficient; Supplementary Fig. 2d,e). Tissue type, preparation and quality can all influence analytical metrics (Methods).

## Spatial comapping of mouse embryo

Spatial ATAC–RNA-seq on E13 mouse embryo identified eight major ATAC clusters and 14 RNA clusters, suggesting that, at this stage of development, chromatin accessibility may not allow discrimination of all cell types defined by transcriptional profiles. Spatial distribution of these clusters agrees with tissue histology (see haematoxylin and eosin staining of an adjacent tissue section in Fig. 1d and Extended Data Fig. 2a). In the ATAC data, cluster A3 represents the embryonic eye field with open chromatin accessibility at the loci of genes including *Six6* (Fig. 1d (left) and Fig. 1e). Clusters A4–A5 are associated with several developing internal organs. Clusters A6–A7 cover the central nervous system (CNS). To benchmark the ATAC result, we aggregated the chromatin accessibility profiles in pixels within specific organs and compared these with organ-specific ENCODE E13.5 ATAC-seq reference

data (Supplementary Fig. 3b). Additionally, the peaks obtained from our spatial ATAC data are also consistent with the ENCODE reference (Supplementary Fig. 3c). We integrated spatial ATAC data with single-cell RNA-seq[12] data for cell type assignment in each cluster (Extended Data Fig. 2c,d). As expected, radial glia (neural stem/progenitor cells) were observed predominantly in the ventricular layer, and differentiated cell types such as postmitotic premature neurons and inhibitory interneurons were enriched in those regions distant from the ventricular layer (Extended Data Fig. 2d).

Cell type-specific marker genes were identified for individual clusters, and their expression was inferred from chromatin accessibility (Fig. 1e and Extended Data Fig. 2b,e) and predicted by gene activity score (GAS[13]; Methods). *Sox2*, which is involved in the development of nervous tissue and optic nerve formation[14,15], showed high chromatin accessibility in the embryonic eye field and in the ventricular layer containing neural stem/progenitor cells. *Pax6* exhibited a similar spatial pattern of chromatin accessibility. *Myt1l*, which encodes myelin transcription factor 1-like protein, presented a higher ATAC signal in the embryonic brain and neural tube[16]. *Six6*, a key gene involved in eye development[17], showed highest GAS in the eye region. *Nrxn2*, which encodes Neurexin 2, a key gene in the vertebrate nervous system[18], was extensively accessible in most neural cell regions. Accessibility at *Rbfox3*, which encodes RNA binding protein fox-1 homolog 3 (a splicing factor known as NeuN)[19], was observed in neurons[19] whereas *Sox1* and *Sox2* presented enriched chromatin accessibility in the ventricular layer (Fig. 1e and Extended Data Fig. 2b). Cell type-specific enrichment of transcription factor (TF) regulators was also examined using ChromVAR[20] analysis of deviation in TF motifs (*Sox2* and *Nfix*), and identified the positive TF regulators (Extended Data Fig. 2g,h). We observed that the *Sox2* motif was enriched in cluster A7, consistent with its function in embryonic brain development[21]. GREAT analysis[22] further verified the strong concordance between gene regulatory pathways and anatomical annotation (Extended Data Fig. 2i,j).

For the RNA spatial data, 14 distinct clusters were identified and characterized by specific marker genes (Fig. 1d (middle) and Extended Data Fig. 2f). For example, cluster R10 (*Six6*) was correlated with the embryonic eye, clusters R2, R6 and R8 were related to the CNS and cluster R7 was associated with the formation of cartilage. To evaluate data quality we performed correlation analysis with organ-specific ENCODE E13.5 RNA-seq reference data, which showed high concordance (Supplementary Fig. 3a). Cell type-specific marker genes were also identified for individual RNA clusters—for example, *Mapt* (R2) may function in establishing and maintaining neuronal polarity[23]. *Epha5* (R6) is involved in axon guidance, whereas *Slc1a3* (R8) is involved in regulation of excitatory neurotransmission in the CNS[18]. *Myh3* (R3) is associated with muscle contraction[24]. *Col2a1* (R7), which shows specific expression in cartilaginous tissues, has an essential role in normal embryonic skeleton development[18]. Pathway analysis[25] results agree with anatomical annotation (Supplementary Fig. 4a).

Integration of spatial RNA data with scRNA-seq mouse organogenesis data[12] was performed to determine cell identities in each pixel (Extended Data Fig. 2c,d). We observed that radial glia, postmitotic premature neurons and inhibitory interneurons (clusters R8, R6 and R2) were present in the same major clusters as shown in the ATAC analysis (clusters A7 and A6), which verified the use of multiple omics information for more robust cell type identification. We attempted joint clustering of spatial ATAC and RNA data to refine the spatial patterns. A new neuronal cluster (J10) was identified in the joint clustering analysis, which was not readily resolved by single modalities alone (Fig. 1d, right). This result highlights the value of using joint multiomics profiles to improve spatial cell type mapping[26].

Co-profiling of the epigenome and transcriptome facilitates investigation of the correlation between accessible peaks and expressed genes pixel by pixel in the tissue context. We observed distinct signals at predicted enhancers for some genes (*Sox2*, *Pax6* and *Sox1*) (Extended

Data Fig. 2e and Supplementary Fig. 4b). For example, enhancers for *Sox2* and *Pax6* had higher chromatin accessibility in clusters A7 and A3, respectively, suggesting their roles in these tissue regions in the regulation of *Sox2* and *Pax6* expression. Although spatial RNA distribution of these genes corresponded with their chromatin accessibility, the expression level may differ significantly from the degree of accessibility (Fig. 1e and Extended Data Fig. 2b). Some marker genes (*Pax6*, *Sox2* and *Myt1l*) were highly accessible in some regions of the embryonic brain but showed modest or low RNA expression (Fig. 1e and Extended Data Fig. 2b), which may indicate lineage priming[10,27] of these genes in embryonic brain development. Despite a strong correlation between replicates (Supplementary Fig. 2d), this observation could be due in part to the sequencing depth and RNA detection sensitivity. Nevertheless, these results still highlight the potential to link genome-wide epigenetic regulation to transcription in the spatial tissue context.

To investigate the spatiotemporal relationship between chromatin accessibility and gene expression in embryonic development, we analysed the differentiation trajectory from radial glia to various types of neurons such as postmitotic premature neurons[28]. Pseudotime analysis[29] was conducted under the ATAC pseudotime coordinate system, and developmental trajectories were directly visualized in the spatial tissue map (Fig. 1f). Chromatin accessibility GAS and gene expression along this trajectory showed dynamic changes in selected marker genes (Fig. 1g,h). As expected, the expression levels of *Sox2*, *Pax6* and other genes involved in progenitor maintenance and proliferation (Gene Ontology (GO) *Fabp7* to *Pax6* in Extended Data Fig. 3a) were downregulated during the transition to postmitotic neurons. The loss of chromatin accessibility at the *Pax6* and radial glia marker *Fabp7* loci preceded downregulation of the corresponding RNAs (Fig. 1h). In turn, genes involved in neuronal identity, axonogenesis and synapse organization (GO *Myt1l* to *Dnm3* in Extended Data Fig. 3a), such as *Dcx* and *Tubb3*, showed increased expression in spatial pseudotime but chromatin accessibility at their loci was already elevated at earlier stages, suggesting lineage priming of these genes in expression[10,27]. We also found a cohort of genes whose expression rapidly declined during spatial pseudotime but whose chromatin accessibility was maintained throughout pseudotime or declined only at very late stages. Many of these genes, such as *Ptprz1*, *Bcan* and *Luzp2*, are characteristic of oligodendrocyte precursor cells, and the corresponding GO biological processes are negative regulation of myelination and regulation of gliogenesis (GO *Cp* to *Sparcl1* in Extended Data Fig. 3a), suggesting that the neuronal lineage might retain the potential to acquire an oligodendrocyte identity even when it has already migrated away from the ventricular zone in the embryonic brain. Monocle2 pseudotime analysis showed a bifurcation in chromatin accessibility but not in RNA expression; one path led to regions close to the ventricular zone (green pixels, Extended Data Fig. 3d,e) and the other terminated in regions distal to the ventricular zone (blue pixels). In contrast to the green path, the blue path presented an increase in chromatin accessibility for genes involved in axonogenesis and dendrite formation (Extended Data Fig. 3 (red box), 3f), suggesting that the chromatin state of neural cells distal to the ventricle corresponds to a more differentiated neuronal state. Thus, our spatial ATAC–RNA-seq can be used to decipher the gene regulation mechanism and spatiotemporal dynamics during tissue development.

## Spatial ATAC–RNA-seq of mouse brain

We developed a microchip with 100 serpentine microchannels to barcode 100 × 100 or, in total, 10,000 pixels per tissue section and up to five samples per chip (Fig. 2a, 20-μm pixel size). This was applied to P22 mouse brain coronal sections (at bregma 1) for joint profiling of chromatin accessibility with transcriptome. In contrast to E13 mouse embryonic brain, we found a higher number of ATAC clusters (14) compared with RNA clusters (11), indicating terminal differentiation of most

major cell types in the juvenile brain relative to that in embryo, which consists of many undifferentiated or multipotent cell states. Spatial distribution of clusters agreed with the anatomical annotation defined by Nissl staining, reflecting arealization of the juvenile brain (Fig. 2b). Moreover, spatial clusters between ATAC and RNA showed concordance in cluster assignment (Supplementary Fig. 5a). Chromatin accessibility of marker genes identified major regions such as striatum (*Pde10a* and *Adcy5*, markers of medium spiny neurons, cluster A1), corpus callosum (*Sox10*, *Mbp* and *Tspan2*, markers of oligodendroglia, cluster A3), cortex (*Mef2c*, *Neurod6* and *Nrn1*, clusters A0 and A4, markers of excitatory neurons; *Cux2*, cluster A4) and lateral ventricle (*Dlx1*, *Pax6*, *Notch1* and *Sox2*, markers of ependymal/neural progenitor cells, cluster A11) (Fig. 2e and Extended Data Fig. 4a,b). GREAT analysis confirmed that the major pathways correlated with tissue functions in different anatomical regions (Supplementary Fig. 5b,c). Spatial RNA clusters also showed region-specific gene expression such as striatum (*Pde10a*, cluster R2, medium spiny neurons), corpus callosum (*Mbp* and *Tspan2*, cluster R5, oligodendroglia) and cortex (*Mef2c*, clusters R0 and R1, excitatory neurons) (Fig. 2e and Extended Data Fig. 4a).

Integration of single-cell ATAC-seq mouse brain atlas data[30] with spatial ATAC-seq data facilitated identification of all major cell types, and label transfer was then used to assign cell types to spatial locations where the epigenetic state may control specific cell type formation (Fig. 2c and Extended Data Fig. 4c). For example, we observed immature oligodendrocytes and oligodendrocytes in cluster A3 corresponding to corpus callosum. A thin layer of radial glia-like cells was found in the lateral ventricle, medium spiny neurons (D1MSN and D2MSN) in striatum and inhibitory neurons in the lateral septal nucleus[30] (Extended Data Fig. 4c) based on chromatin accessibility. Furthermore, the subclasses of excitatory neurons in different layers of cortex (ITL23GL, Cortex L2/3; ITL4GL, Cortex L4; ITL5GL, Cortex L5; PTGL, Cortex PT; NPGL, Cortex NP; ITL6GL, Cortex L6; CTGL, Cortex CT; and L6bGL, Cortex L6b) were clearly identified[31] (Extended Data Fig. 4c). Integration of spatial RNA data and the juvenile CNS scRNA-seq atlas[32] allowed for visualization of dominant transcriptional cell types in tissue (Fig. 2d and Extended Data Fig. 4d). We observed a high degree of concordance in spatial cell type distribution as determined by ATAC versus RNA, suggesting a general congruence between chromatin accessibility and transcriptome in defining cell identities in tissue. For instance, neuronal intermediate progenitor cells (SZNBL, included in radial glial-like cells from scATAC-seq[30]), mature oligodendrocytes (MOL, included in oligodendrocytes from scATAC-seq), newly formed oligodendrocytes (included in immature oligodendrocytes from scATAC-seq) and medium spiny neurons (MSN) shown by the RNA data were enriched in the same regions as identified by the ATAC data (Extended Data Fig. 4c,d). Furthermore, the detection of a thin layer of SZNBL in the lateral ventricle and vascular leptomeningeal cells (VLMC) in the regions with preserved meninges demonstrated a high spatial resolution for our technology in identifying low-abundance cells, even at near-single-cell resolution (for VLMC) (Extended Data Fig. 4c,d).

Joint profiling of ATAC and RNA facilitates inferring the gene regulatory landscape by searching correlated peak accessibility and gene expression. We detected 21,417 significant peak-to-gene linkages between regulatory elements and target genes (Extended Data Fig. 5c,d). Some potential enhancers with dynamically regulated promoter interactions were found to be enriched in specific clusters such as *Dlx1* and *Sox2* (cluster A11), *Tspan2* (cluster A3) and *Adcy5* (cluster A1) (Extended Data Fig. 4b), indicating the ability of spatial ATAC–RNA-seq to identify the key regulatory regions for target genes. Spatial patterns of TF motif enrichment (*Dnajc21*, *Pax6*, *Sox2*, *Sox4* and *Mef2c*) provided further insights into the TF regulators[33] visualized in tissue (Extended Data Fig. 5a,b). For example, *Sox2* was examined simultaneously for chromatin accessibility, gene expression, putative enhancers and TF motif enrichment, enabling a more comprehensive understanding of gene regulation dynamics. As observed in the embryonic mouse brain,

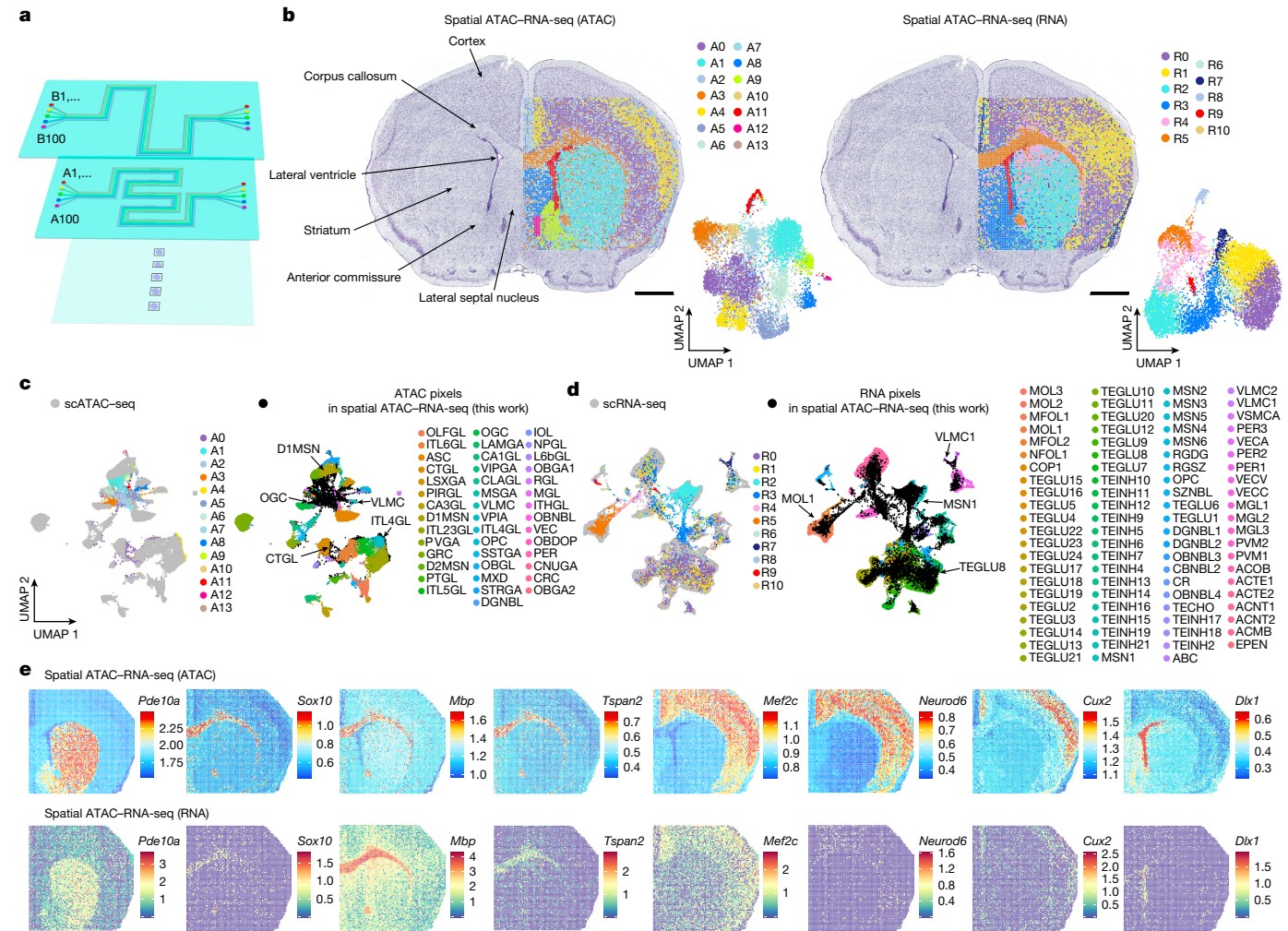

**Fig. 2 | Spatial chromatin accessibility and transcriptome co-profiling of P22 mouse brain. a**, Design of microfluidic chips for 100 × 100 barcodes with 20-μm channel size. **b**, Spatial distribution and UMAP of all clusters for ATAC and RNA in spatial ATAC–RNA-seq of mouse brain. Pixel size, 20 μm; scale bars, 1 mm. **c**, Integration of ATAC data and scATAC-seq data[30] from mouse brain.

**d**, Integration of RNA data and scRNA-seq data[32] from mouse brain. **e**, Spatial mapping of GAS and gene expression for selected marker genes in different clusters for ATAC and RNA in spatial ATAC–RNA-seq. A list of abbreviation definitions can be found in Supplementary Table 1.

some marker genes with open chromatin accessibility (*Sox10*, *Sox2*, *Neurod6*, *Pax6* and *Notch1*) were either not or lowly expressed (Fig. 2e and Extended Data Fig. 4a). A biological replicate experiment on P21 mouse brain was also performed (Extended Data Fig. 6; 20 μm pixel size, 50 × 50 barcodes). Many of these genes encode for transcription factors involved in neural development, suggesting the possibility of epigenetic—but not transcriptional—memory of these genes in brain development.

## Spatial CUT&Tag–RNA-seq of mouse brain

In addition to chromatin accessibility, histone modifications are also a key aspect of epigenetic regulation. Spatial CUT&Tag–RNA-seq was performed to co-profile transcriptome and H3K27me3 (repressing loci), H3K27ac (activating promoters and/or enhancers) or H3K4me3 (active promoters) histone modifications, respectively, in P22 mouse brain (20-μm pixel size, 100 × 100 barcodes). We identified 13 and 15 specific clusters for H3K27me3 and RNA, 12 and 13 clusters for H3K27ac and RNA and 11 and 12 clusters for H3K4me3 and RNA, respectively (Fig. 3a–c). These clusters agreed with the anatomical annotation by Nissl staining and showed good concordance between CUT&Tag and RNA in spatial patterns (Fig. 3a–c), which was further confirmed by Belayer[34] analysis (Supplementary Fig. 7c).

For H3K27me3, gene expression was predicted by calculation of chromatin silencing score (CSS[2,35]; Methods). High CSS indicates repressed gene expression because of the transcriptional repression function of H3K27me3. For H3K27ac and H3K4me3, active genes should correspond to high GAS[2,13] (Methods). Clustering of CSS or GAS based on different modifications resolved all major tissue regions (Fig. 3a–c and Supplementary Fig. 6) and identified region-specific marker gene modifications. *Cux2* showed high GAS in cluster C9 (H3K27ac) and cluster C6 (H3K4me3), indicating enrichment of excitatory neurons[5]. *Cux2* was observed in cortical layers 2/3, corresponding to superficial layers in the cortex. By contrast, H3K27me3 was depleted at *Cux2* in the same region (cluster C3) (Supplementary Fig. 6). GAS of *Fezf2* (a marker gene for cortical layer 5) was high in cluster C8 (H3K27ac) and cluster C7 (H3K4me3), corresponding to the deeper layer of the cortex. *Fezf2* was depleted for H3K27me3 (cluster C1) in this layer. *Satb2* showed the highest activity in clusters C8 and C9 (H3K27ac) and C6 and C7 (H3K4me3), but the lowest CSS in clusters C1 and C3 (H3K27me3) in the cortical layer. *Tspan2* was enriched in cluster C4 (H3K27ac) and cluster C1 (H3K4me3) but depleted in cluster C5 (H3K27me3) in corpus callosum, associated with oligodendrocyte lineage development (Supplementary Fig. 6). The corresponding RNA clusters also showed concordant region-specific signatures (Fig. 3a–c and Supplementary

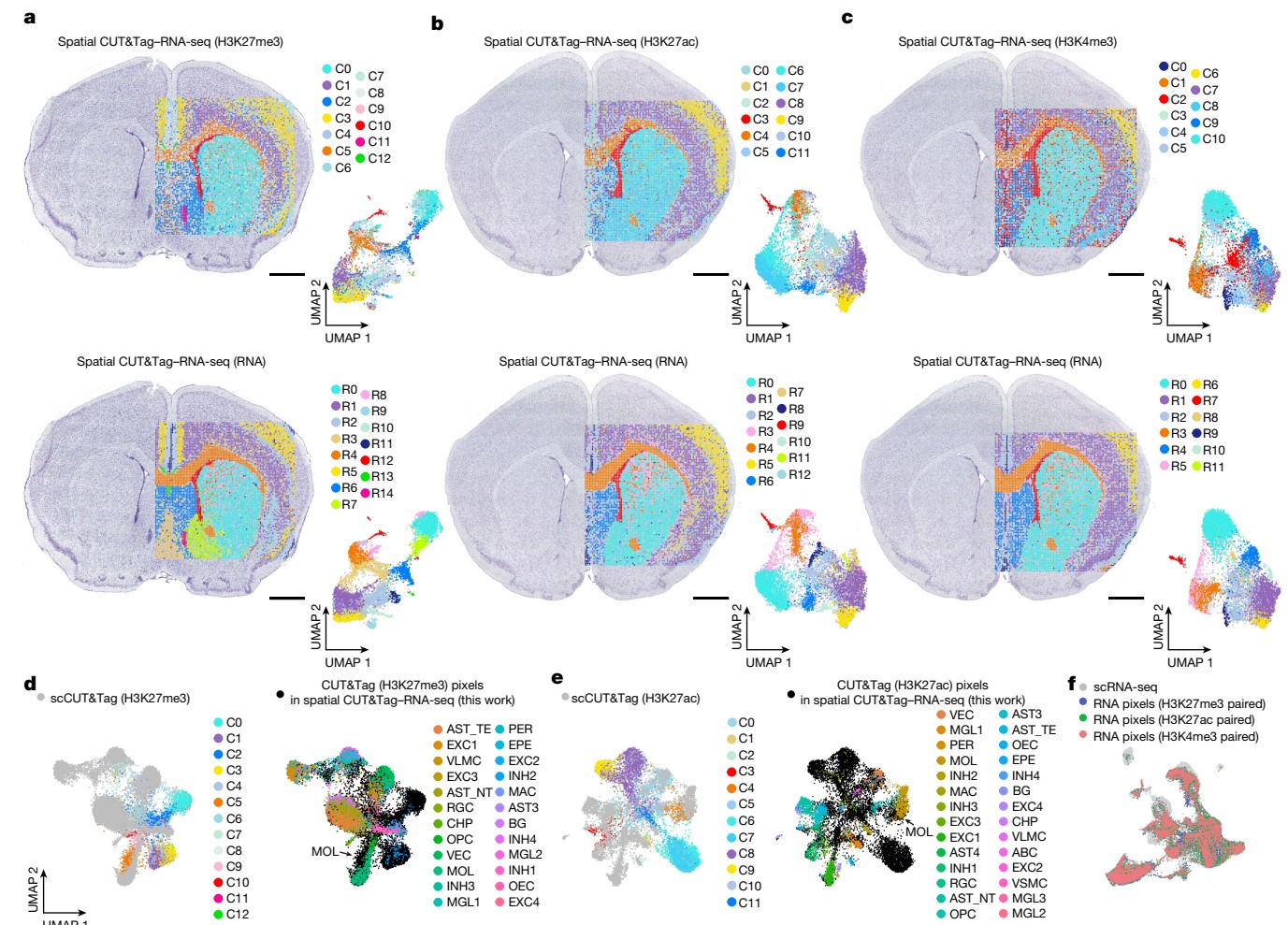

**Fig. 3 | Spatial histone modification and transcriptome co-profiling of P22 mouse brain. a–c**, Spatial distribution and UMAP of all clusters for H3K27me3 and RNA (**a**), H3K27ac and RNA (**b**) and H3K4me3 and RNA (**c**) in mouse brain. Pixel size, 20 μm; scale bars, 1 mm. **d**, Integration of H3K27me3 data with scCUT&Tag (H3K27me3) data[37] from mouse brain. **e**, Integration of H3K27ac data with scCUT&Tag (H3K27ac) data[37] from mouse brain. **f**, Integration of RNA data in spatial CUT&Tag (H3K27me3)–RNA-seq, spatial CUT&Tag (H3K27ac)–RNA-seq and spatial CUT&Tag (H3K4me3)–RNA-seq with scRNA-seq data[32] from mouse brain.

Fig. 6), as exemplified by *Cux2* in cortical layer 2/3 excitatory neurons (cluster R5 (H3K27me3 paired), cluster R5 (H3K27ac paired) and cluster R6 (H3K4me3 paired)), *Fezf2* in cortical layer 5 excitatory neurons (cluster R1 (H3K27me3 paired), cluster R1 (H3K27ac paired), cluster R1 (H3K4me3 paired)), *Tspan2* in oligodendrocyte lineage cells in corpus callosum (cluster R4 (H3K27me3 paired), cluster R4 (H3K27ac paired) and cluster R3 (H3K4me3 paired) Supplementary Fig. 6).

Integration and co-embedding of spatial CUT&Tag and scCUT&Tag data[36,37] (Fig. 3d,e and Extended Data Fig. 7a) showed that the epigenetic states in our data for H3K27me3, H3K27ac and H3K4me3 agreed with the corresponding projection in scCUT&Tag. Integration with the corresponding juvenile CNS scRNA-seq atlas[32] allowed for label transfer to assign transcriptional cell types to the spatial location of epigenetic identities/states (Extended Data Fig. 7g,h, bottom). For example, we observed an enrichment of MOL within the corpus callosum, a thin layer of ependymal cells in the lateral ventricle, excitatory neurons in cerebral cortex and MSN in the striatum. We also integrated paired RNA data with the scRNA-seq atlas[32] to identify dominant transcriptional cell types via label transfer (Fig. 3f and Extended Data Fig. 7b–h). MOL, a thin layer of ependymal cells, MSN and excitatory neurons were enriched in the same spatial regions identified by CUT&Tag (H3K27ac and H3K4me3) (Extended Data Fig. 7g,h, top). In particular for spatial H3K27me3, whereas integration with scCUT&Tag could not clearly show

the identity of several clusters in the epigenomic modalities (clusters C0, C1 and C3), label transfer of scRNA-seq data with paired RNA data from the same tissue section clearly showed cell identities in these clusters (Fig. 3a,d and Extended Data Fig. 7f), highlighting the power of combining CUT&Tag and RNA-seq (spatial CUT&Tag–RNA-seq) in the same tissue section to more accurately identify cell types or states. To directly infer interactions between genome-wide gene expression and the corresponding enhancers across all clusters, we identified a total of 19,468 significant peak-to-gene linkages from spatial CUT&Tag (H3K27ac)–RNA-seq (Supplementary Fig. 7a,b). A biological replicate was also performed on P21 mouse brain (Extended Data Fig. 8; 20-μm pixel size, 50 × 50 barcodes).

## Region-specific gene expression regulation

To further understand the spatial epigenetic regulation of gene expression at the genome scale, we compared the GAS or CSS obtained from ATAC and CUT&Tag (H3K27me3, H3K27ac and H3K4me3) with the corresponding RNA expression in all major tissue regions of P22 mouse brain (Fig. 4a–c and Supplementary Figs. 9a, 10a and 11a). For example, in oligodendrocyte-abundant corpus callosum we observed a robust anticorrelation between H3K27me3 and RNA (Fig. 4a). Genes such as *Mal*, *Mag* and *Car2* with low CSS and high RNA expression correspond

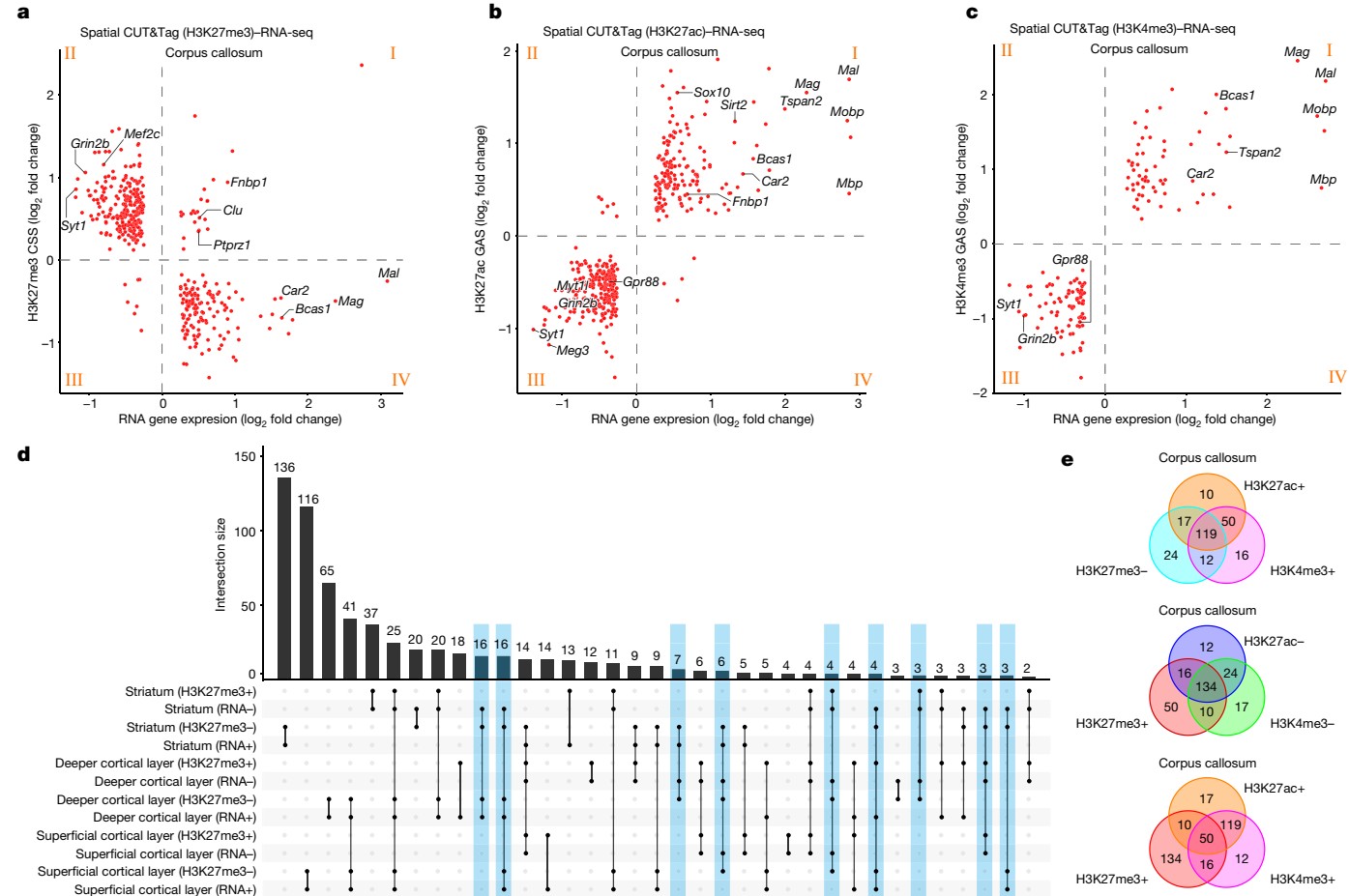

**Fig. 4 | Region-specific epigenetic regulation of gene expression.**
**a**–**c**, Correlation of H3K27me3 CSS and RNA gene expression (**a**), H3K27ac GAS and RNA gene expression (**b**) and H3K4me3 GAS and RNA gene expression (**c**) in corpus callosum. **d**, Upset plot of H3K27me3 CSS and RNA gene expression in striatum and deeper and superficial cortical layers; –, low CSS or gene expression; +, high CSS or gene expression. **e**, Venn diagrams showing high (+) or low (–) CSS/GAS for different histone modifications in corpus callosum with common RNA marker genes.

to GO biological processes such as myelination and regulation of oligodendrocyte differentiation (Extended Data Fig. 9a–c and Supplementary Fig. 8; GO for quadrant IV). By contrast, genes such as *Grin2b* and *Syt1* with high CSS and low RNA expression are related to neuronal processes such as synaptic transmission and neurotransmitter release (Extended Data Fig. 9d,e and Supplementary Fig. 8; GO for quadrant II). These results are in accordance with ATAC/H3K27me3/H3K27ac Nano-CUT&Tag analysis[37] of oligodendroglial differentiation in the juvenile brain, which showed two waves of H3K27me3 repression during this process[37]. We also found a small subset of genes (Fig. 4a and Supplementary Fig. 8; GO for quadrant I) with higher levels of both H3K27me3 and RNA in the corpus callosum, such as *Ptprz1*, a marker of oligodendrocyte precursor cells[38,39] (Extended Data Fig. 9g), which might indicate transcriptional poising of some of these genes. This could also be due in part to the presence in proximity of both oligodendrocyte precursor cells (expressing *Ptprz1*) and mature oligodendrocytes (where *Ptprz1* is repressed) in the corpus callosum. We performed a similar correlation analysis for RNA and H3K27me3 in other regions of the juvenile brain, including the striatum and superficial and deeper cortical layers (Supplementary Figs. 9a, 10a and 11a). We also observed strong anticorrelation with a cohort of genes involved in neuronal processes being activated or repressed in a region-specific manner. For instance, in the superficial cortical layer, genes for GABAergic regulation of synaptic transmission were enriched in H3K27me3 whereas glutamatergic synapse transmission genes had high expression and low H3K27me3 (Supplementary

Figs. 9b, 10b and 11b; GOs for striatum, superficial and deeper cortical layers). In contrast to corpus callosum, only a limited number of genes positive for both H3K27me3 and RNA were found in these regions. Despite a general anticorrelation between RNA expression and H3K27me3 deposition, we also observed regional differences (blue-shaded columns in Fig. 4d); *Nav3* and *Sncb*, with low levels of H3K27me3 in both the cortex and striatum, were expressed in the former but not the latter (Extended Data Fig. 9h). The opposite pattern was observed for genes such as *Ablim2* and *Gng7* (Extended Data Fig. 9h). *Car2*, a marker gene expressed in oligodendrocytes and abundant in the corpus callosum, presented H3K27me3 occupancy in the cortical layers, in contrast to the corpus callosum and, surprisingly, the striatum (Extended Data Fig. 9c). Thus, mechanisms other than Polycomb-mediated H3K27me3 deposition might be involved in the transcriptional repression of these genes in different areas of the CNS.

In contrast to H3K27me3, we observed a robust correlation between RNA and activating marks H3K4me3 or H3K27ac in the corpus callosum, with genes highly expressed and with high deposition in these two modalities regulating processes in oligodendroglia (Fig. 4b,c and Supplementary Fig. 12; GO for H3K27ac quadrant I and H3K4me3 quadrant I). For instance, *Mal* showed the highest activity or expression in the corpus callosum for ATAC, H3K27ac, H3K4me3 and paired RNA (Extended Data Fig. 9a). *Gpr88*, a marker gene of MSN, was enriched in striatum for ATAC, H3K27ac and H3K4me3 but with low CSS in striatum for H3K27me3 (Extended Data Fig. 9f). Furthermore, we examined collective regulation among different histone modifications in the

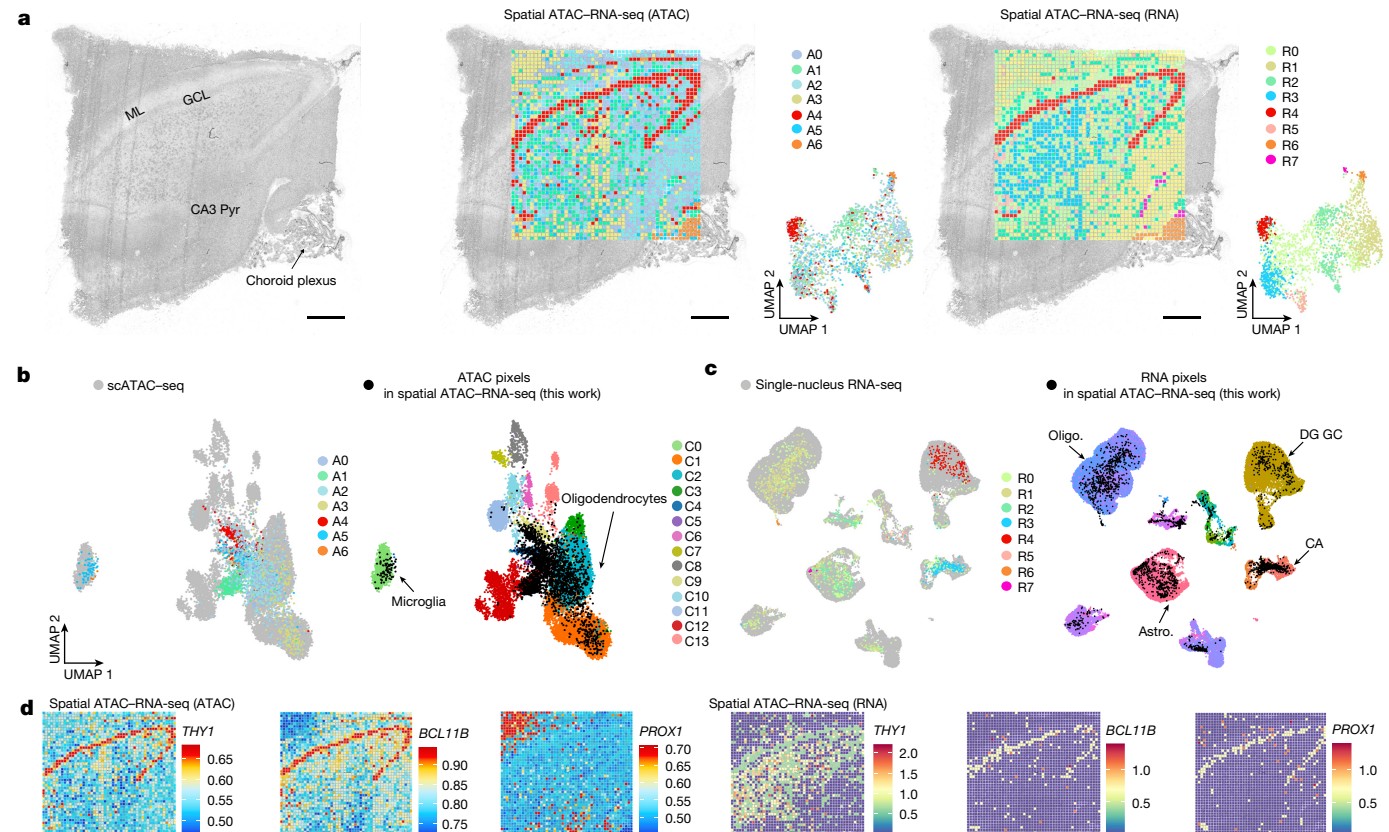

**Fig. 5 | Spatial chromatin accessibility and transcriptome co-profiling of human hippocampus. a**, Brightfield image, spatial distribution and UMAP of all clusters based on ATAC and RNA in the human hippocampus. ML, molecular layer; Pyr, pyramidal neurons. Pixel size, 50 μm, scale bars, 1 mm. **b**, Integration of our ATAC data with scATAC-seq data[42] from human hippocampus.

**c**, Integration of our RNA data with snRNA-seq data from human brain[43]. **d**, Spatial mapping of GAS and gene expression for selected marker genes in different clusters for ATAC and RNA. Oligo, oligodendrocytes; astro, astrocytes; DG, dentate gyrus.

corpus callosum (Fig. 4e and Supplementary Figs. 13 and 14). In general H3K4me3 and H3K27ac showed high correlations, with their signals anticorrelated with H3K27me3 (Fig. 4e). We also observed H3K4me3 or H3K27ac and H3K27me3 co-occupancy in several gene loci in this tissue region (Fig. 4e and Supplementary Figs. 13 and 14). These genes are involved in oligodendrocyte differentiation and myelination and other processes such as protein catabolism and localization (Supplementary Figs. 13 and 14). Some of these genes, such as *Ptprz1* and *Fnbp1*, also presented higher levels of H3K27me3 and RNA (Fig. 4a). As mentioned above, this potential H3K4me3/H3K27me3 bivalency may reflect transcriptional poising.

## Spatial comapping of human brain

We performed spatial ATAC–RNA-seq on adult (31-year-old male) human brain hippocampal formation, which is a complex brain region involved in cognitive functions and diseases such as major depression disease[40] and Alzheimer's disease[41]. We identified seven and eight major clusters for ATAC and RNA, respectively, and the spatial pattern agreed with major anatomical landmarks (Fig. 5a)[3], including specific marker genes (Fig. 5d). The ATAC cluster A4 represents the granule cell layer (GCL) (*THY1*, *BCL11B*) and cluster A6 corresponds to the choroid plexus (Fig. 5a,d). TF motifs (*NEUROD1* and *SNAI1*) and their spatial patterns were visualized in different tissue regions, and positive TF regulators were identified (Extended Data Fig. 10b,c). For the RNA data, we also detected distinct clusters with unique marker genes (Fig. 5a,d) such as *PROX1* and *BCL11B* enriched in cluster R4 (GCL).

We also conducted integration of our ATAC data and scATAC-seq data from human brain samples[42] (Fig. 5b), and our RNA data with adult

human brain snRNA-seq data[43], to show the dominant cell identities and states. Cell types identified by snRNA-seq[43] were assigned to each cluster via label transfer (Fig. 5c and Extended Data Fig. 10a). Granule cells (GC) were detected in the GCL, cornu ammonis (CA) neurons were enriched in CA3–4 regions and VLMC were strongly distinguished in choroid plexus as opposed to other regions.

In general, both ATAC and RNA can readily resolve all major tissue features in this region, but spatial cosequencing of epigenome and transcriptome may provide new insights into the dynamic gene regulation mechanism that cannot be realized by single modalities. For example, *PROX1*, a signature gene defining granule neuron identity during pyramidal neuron fate selection[44], is indeed highly expressed in GCL but showed modest chromatin accessibility (Fig. 5d). This might be attributed to a minimal demand for synthesis of new *PROX1* transcripts in postmitotic mature granule cells, and is thus not required to maintain an active open chromatin state.

## Discussion

Spatial omics technologies (spatial epigenomics, transcriptomics and proteomics), based on either NGS[1–4,45–47] or imaging[5,48], offer an unprecedented opportunity to generate new and rich insights into gene regulation in the spatial tissue context. However, to comprehensively understand the mechanism of gene regulation, different layers of omics information need to be profiled simultaneously. This was first demonstrated in dissociated single cells[6–9] but is yet to be realized directly in tissue. Imaging-based DNA sequential fluorescence in situ hybridization combined with RNA sequential fluorescence in situ hybridization detected spatial chromatin and gene expression for

target genes and selected genomic loci[49]. As of today, current technologies have not been able to perform unbiased genome-wide comapping of epigenome and transcriptome on the same tissue section at the cellular level. We developed spatial ATAC–RNA-seq and spatial CUT&Tag–RNA-seq (applied to H3K27me3, H3K27ac and H3K4me3) for co-profiling of genome-wide chromatin accessibility or histone modifications in conjunction with whole transcriptome on the same tissue section at near-single-cell resolution (available for spatial browsing, see 'Data availability'). These techniques were applied to comapping of embryonic and juvenile mouse brain, as well as adult human brain. Spatial epigenome–transcriptome cosequencing yielded two layers of spatial omics information directly in tissue, with data quality similar to that previously obtained by single modalities. These technologies allow us to examine spatiotemporal dynamics and genome-wide gene regulation mechanisms in the tissue context.

We have demonstrated either ATAC or CUT&Tag in conjunction with RNA for spatial multiomics mapping. It might be possible to combine all three—chromatin accessibility, histone modifications and transcriptome—to delineate a more comprehensive landscape of gene regulation network in tissue. It might be also possible to simultaneously measure multiple histone modifications to assess the multivalency effect on spatial gene expression regulation. Previously we demonstrated spatial profiling of transcriptome and a large panel of proteins[1]. We envision that it is possible and highly valuable to further combine epigenome, transcriptome and proteome[50] to dissect spatial patterns of cell type, state and gene regulation. Finally, spatial multiomics as reported herein may find widespread applications beyond neuroscience and developmental biology. For example, multiple modalities for spatial mapping of human disease tissues not only cross-validate one another but also better elucidate the mechanisms driving abnormal cell states that could not be readily discerned using single-modality methods. In addition, spatial information may further show how the local tissue environment affects cell state, dynamics and function across all layers of the central dogma. In this work, we developed a device with a larger mapping area and higher throughput that allows the mapping of a fourfold larger tissue area, using 100 × 100 barcodes (total 10,000 pixels, 20-µm pixel size) to cover almost the entire hemisphere of a juvenile P22 mouse brain coronal section. The serpentine channel design allows simultaneous processing of five tissue samples, each mapped for 10,000 pixels. It is feasible to further increase the mapping area by increasing the number of barcodes and further enhance throughput using automated liquid handling.

In summary, spatially resolved, genome-wide cosequencing of epigenome and transcriptome at the cellular level represents one of the most informative tools in spatial biology and can be applied to a wide range of biological and biomedical research.

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

# Methods

## Preparation of tissue slides

Mouse C57 embryo sagittal frozen sections (no. MF-104-13-C57) were purchased from Zyagen. Freshly harvested E13 mouse embryos were snap-frozen in optimal cutting temperature compounds and sectioned at 7–10 μm thickness. Tissue sections were collected on poly-L-lysine-coated glass slides (Electron Microscopy Sciences, no. 63478-AS).

Juvenile mouse brain tissue (P21–P22) was obtained from the *Sox10:Cre-RCE:LoxP* enhanced green fluorescent protein (eGFP) line on a C57BL/6xCD1 mixed genetic background maintained at Karolinska Institutet. The line was first generated by crossing *Sox10:Cre* animals (The Jackson Laboratory, no. 025807) with *RCE:loxP* (eGFP) animals (The Jackson Laboratory, no. 032037-JAX). This was established and maintained by breeding males lacking the *Cre* allele with females carrying a hemizygous *Cre* allele; the reporter allele was maintained in homozygosity or hemizygosity in both males and females. This resulted in specific labelling of oligodendrocyte lineage with eGFP. All animals were free from mouse bacterial and viral pathogens, ectoparasites and endoparasites. The following light/dark cycle was maintained for the mice: dawn, 06:00–07:00; daylight, 07:00–18:00; dusk, 18:00–19:00; night, 19:00–06:00. Mice were housed in individually ventilated cages at a maximum number of five per cage (IVC sealsafe GM500, tecniplast). General housing parameters including temperature, ventilation and relative humidity followed the European Convention for the Protection of Vertebrate Animals used for experimental and other scientific purposes. Air quality was controlled using stand-alone air-handling units equipped with a high-efficiency particulate air filter. Relative air humidity was consistently 55 ± 10%, at a temperature of 22 °C. Husbandry parameters were monitored with ScanClime (Scanbur) units. The cages contained a card box shelter, gnawing sticks and nesting material (Scanbur), placed on hardwood bedding (TAPVEI). Mice were provided with a regular chow diet, and water was supplied by water bottle and changed weekly. Cages were changed every 2 weeks in a laminar air-flow cabinet.

Mice were sacrificed at P21/P22 (both sexes were used) by anaesthesia with ketamine (120 mg kg$^{-1}$ body weight) and xylazine (14 mg kg$^{-1}$ body weight), followed by transcranial perfusion with cold oxygenated artificial cerebrospinal fluid (87 mM NaCl, 2.5 mM KCl, 1.25 mM NaH$_2$PO$_4$, 26 mM NaHCO$_3$, 75 mM sucrose, 20 mM glucose, 1 mM CaCl$_2$*2H$_2$O and 2 mM MgSO$_4$*7H$_2$O in distilled H$_2$O). Following isolation, brains were maintained for a minimal time period in artificial cerebrospinal fluid until embedding in Tissue-Tek O.C.T. compound (Sakura) and snap-freezing using a mixture of dry ice and ethanol. Coronal cryosections of 10 μm were mounted on poly-L-lysine-coated glass slides (no. 63478-AS, Electron Microscopy Sciences) or 2 × 3 square inch glass slides (AtlasXomics).

All experimental procedures were conducted following European directive no. 2010/63/EU, local Swedish directive no. L150/SJVFS/2019:9, Saknr no. L150 and Karolinska Institutet complementary guidelines for procurement and use of laboratory animals, no. Dnr. 1937/03-640. All procedures described were approved by the local committee for ethical experiments on laboratory animals in Sweden (Stockholms Norra Djurförsöksetiska nämnd, nos. 1995/2019 and 7029/2020).

Human brain tissue was obtained from the brain collection of the New York State Psychiatric Institute at Columbia University, which includes brain samples from the Republic of Macedonia. Brain tissue collection was conducted with New York State Psychiatric Institute Institutional Review Board approval, and informed consent obtained from next of kin who agreed to donate the brains and participate in psychological autopsy interviews.

We analysed brain hippocampus tissue from a 31-year-old Caucasian male individual, with no psychiatric or neurological diagnosis, who had died of a traumatic accident and had a high level of global functioning before death as measured by global assessment scale[51] score, which was 90 (scoring 1–100, with 100 the highest function), and with toxicology negative for psychotropic medications and drugs. Postmortem interval (time from demise to brain collection) was 6.5 h.

The anterior hippocampal region was dissected from a fresh-frozen coronal section (20-mm thickness) of the right brain hemisphere. The dentate gyrus region (around 10 × 10 mm$^2$) of the anterior hippocampal region was selected. Cryosections of 10 μm were collected on poly-L-lysine-coated glass slides (no. 63478-AS, Electron Microscopy Sciences). Samples were stored at −80 °C until further use.

## Preparation of transposome

Unloaded Tn5 transposase (no. C01070010) and pA-Tn5 (no. C01070002) were purchased from Diagenode, and the transposome was assembled following the manufacturer's guidelines. The oligos applied for transposome assembly were:

Tn5ME-A, 5′-TCGTCGGCAGCGTCAGATGTGTATAAGAGACAG-3′,
Tn5MErev, 5′-/5Phos/CTGTCTCTTATACACATCT-3′ and
Tn5ME-B, 5′-/5Phos/CATCGGCGTACGACTAGATGTGTATAAGAGACAG-3′

## DNA barcode sequences, DNA oligos and other key reagents

DNA oligos used for PCR and library construction are shown in Supplementary Table 4. All DNA barcode sequences are provided in Supplementary Tables 5 and 6 and all other chemicals and reagents in Supplementary Table 7.

## Fabrication of PDMS microfluidic device

Chrome photomasks were purchased from Front Range Photomasks, with a channel width of either 20 or 50 μm. The moulds for polydimethylsiloxane (PDMS) microfluidic devices were fabricated using standard photolithography. The manufacturer's guidelines were followed to spin-coat SU-8-negative photoresist (nos. SU-2025 and SU-2010, Microchem) onto a silicon wafer (no. C04004, WaferPro). The heights of the features were about 20 and 50 μm for 20- and 50-μm-wide devices, respectively. PDMS microfluidic devices were fabricated using the SU-8 moulds. We mixed the curing and base agents in a 1:10 ratio and poured the mixture into the moulds. After degassing for 30 min the mixture was cured at 70 °C for 2 h. Solidified PDMS was extracted for further use. We have published a detailed protocol for the fabrication and preparation of the PDMS device[52].

## Spatial ATAC–RNA-seq

Frozen tissue slides were thawed for 10 min at room temperature. Tissue was fixed with formaldehyde (0.2%, with 0.05 U μl$^{-1}$ RNase Inhibitor) for 5 min and quenched with 1.25 M glycine for a further 5 min. After fixation, tissue was washed twice with 1 ml of 0.5× DPBS-RI and cleaned with deionized (DI) H$_2$O.

Tissue permeabilization was carried out with 200 μl of lysis buffer (3 mM MgCl$_2$, 0.01% Tween-20, 10 mM Tris-HCl pH 7.4, 0.01% NP40, 10 mM NaCl, 1% bovine serum albumin (BSA), 0.001% digitonin, 0.05 U μl$^{-1}$ RNase inhibitor) for 15 min and washed twice with 200 μl of wash buffer (10 mM Tris-HCl pH 7.4, 10 mM NaCl, 3 mM MgCl$_2$, 1% BSA, 0.1% Tween-20) for 5 min. Transposition mix (5 μl of home-made transposome, 33 μl of 1× DPBS, 50 μl of 2× Tagmentation buffer, 1 μl of 1% digitonin, 1 μl of 10% Tween-20, 0.05 U μl$^{-1}$ RNase Inhibitor, 10 μl of nuclease-free H$_2$O) was added with incubation at 37 °C for 30 min. Next, 200 μl of 40 mM EDTA with 0.05 U μl$^{-1}$ RNase inhibitor was added with incubation for 5 min at room temperature, to stop transposition. Finally, tissue sections were washed twice with 200 μl of 0.5× PBS-RI for 5 min and cleaned with DI water.

For RT, the following mixture was used: 12.5 μl of 5× RT buffer, 4.5 μl of RNase-free water, 0.4 μl of RNase inhibitor, 0.8 μl of Superase In RNase inhibitor, 3.1 μl of 10 mM deoxynucleotide triphosphate each, 6.2 μl of Maxima H Minus Reverse Transcriptase, 25 μl of 0.5× PBS-RI and 10 μl

of RT primer. Tissues were incubated for 30 min at room temperature, then at 42°C for 90 min in a wet box. After the RT reaction, tissues were washed with 1× NEBuffer 3.1 and 1% RNase inhibitor for 5 min.

For first barcode (barcode A) in situ ligation, the first PDMS chip was covered to the tissue region of interest. For alignment purposes, a 10× objective (Thermo Fisher Scientific, EVOS FL Auto microscope no. AMF7000, EVOS FL Auto 2 Software Revision 2.0.2094.0) was used to take a brightfield image. The PDMS device and tissue slide were clamped tightly with a home-made acrylic clamp. Barcode A was first annealed with ligation linker 1, 10 µl of 100 µM ligation linker, 10 µl of 100 µM each barcode A and 20 µl of 2× annealing buffer (20 mM Tris pH 7.5–8.0, 100 mM NaCl, 2 mM EDTA) and mixed well. For each channel, 5 µl of ligation master mixture was prepared with 2 µl of ligation mix (27 µl of T4 DNA ligase buffer, 72.4 µl of RNase-free water, 5.4 µl of 5% Triton X-100, 11 µl of T4 DNA ligase), 2 µl of 1× NEBuffer 3.1 and 1 µl of each annealed DNA barcode A (A1–A50/A100, 25 µM). Vacuum was used to load the ligation master mixture into 50 channels of the device, followed by incubation at 37 °C for 30 min in a wet box. The PDMS chip and clamp were removed after washing with 1× NEBuffer 3.1 for 5 min. The slide was washed with water and dried in air.

For second barcode (barcode B) in situ ligation, the second PDMS chip was covered to the slide and a further brightfield image taken with the 10× objective. An acrylic clamp was applied to clamp the PDMS and tissue slide together. Annealing of barcodes B (B1–B50/B100, 25 µM) and preparation of the ligation master mix were carried out as for barcodes A. The device was incubated at 37 °C for 30 min in a wet box. The PDMS chip and clamp were removed after washing with 1× DPBS with SUPERase In RNase inhibitor for 5 min. The slide was washed with water and dried in air. A brightfield image was then taken for further alignment.

For tissue lysis, the region of interest was digested with 100 µl of reverse crosslinking mixture (0.4 mg ml$^{-1}$ proteinase K, 1 mM EDTA, 50 mM Tris-HCl pH 8.0, 200 mM NaCl, 1% SDS) at 58 °C for 2 h in a wet box. The lysate was collected in a 1.5 m tube and incubated at 65 °C overnight.

For DNA and cDNA separation, the lysate was purified with Zymo DNA Clean & Concentrator-5 and eluted to 100 µl of RNase-free water. The 1× B&W buffer with 0.05% Tween-20 was used to wash 40 µl of Dynabeads MyOne Streptavidin C1 beads three times. Then, 100 µl of 2× B&W buffer with 2.5 µl of SUPERase In RNase inhibitor was used to resuspend the beads, which were then mixed with the lysate and allowed to bind at room temperature for 1 h with agitation. A magnet was used to separate beads and supernatant in the lysate.

The supernatant was removed for ATAC library construction, then purified with Zymo DNA Clean & Concentrator-5 and eluted to 20 µl of RNase-free water. PCR solution (25 µl of 2× NEBNext Master Mix, 2.5 µl of 25 µM indexed i7 primer, 2.5 µl of 25 µM P5 PCR primer) was added and mixed well. PCR was first performed with the following programme: 72 °C for 5 min, 98 °C for 30 s and cycling at 98 °C for 10 s, 63 °C for 30 s and 72 °C for 1 min, five times. To determine additional cycles, the pre-amplified mixture (5 µl) was mixed with quantitative PCR (qPCR) solution (5 µl of 2× NEBNext Master Mix, 0.24 µl of 25× SYBR Green, 0.5 µl of 25 µM new P5 PCR primer, 3.76 µl of nuclease-free H$_2$O, 0.5 µl of 25 µM indexed i7 primer). The qPCR reaction was then carried out with the following programme: 98 °C for 30 s with cycling at 98 °C for 10 s, 63 °C for 30 s and 72 °C for 1 min, 20 times. The remaining pre-amplified DNA (45 µl) was amplified by running additional cycles as determined by qPCR (to reach one-third of saturated signal). The final PCR product was purified by 1× Ampure XP beads (45 µl) and eluted in 20 µl of nuclease-free H$_2$O.

The beads were used for cDNA library construction. They were first washed twice with 400 µl of 1× B&W buffer with 0.05% Tween-20 and once with 10 mM Tris pH 8.0 containing 0.1% Tween-20. Streptavidin beads with bound cDNA molecules were resuspended in a TSO solution (22 µl of 10 mM deoxynucleotide triphosphate each, 44 µl of 5× Maxima

RT buffer, 44 µl of 20% Ficoll PM-400 solution, 88 µl of RNase-free water, 5.5 µl of 100 uM template switch primer (AAGCAGTGGTATCAACGCA GAGTGAATrGrG+G), 11 µl of Maxima H Minus Reverse Transcriptase, 5.5 µl of RNase Inhibitor). The beads were incubated at room temperature for 30 min and then at 42 °C for 90 min, with gentle shaking. After washing beads once with 400 µl of 10 mM Tris and 0.1% Tween-20 and once with water, they were resuspended in a PCR solution (110 µl of 2× Kapa HiFi HotStart Master Mix, 8.8 µl of 10 µM primers 1 and 2, 92.4 µl of RNase-free water). PCR thermocycling was carried out using the following programme: 95 °C for 3 min and cycling at 98 °C for 20 s, 65 °C for 45 s and 72 °C for 3 min, five times. After five cycles, beads were removed from the PCR solution and 25× SYBR Green was added at 1× concentration. Samples were again placed in a qPCR machine with the following thermocycling conditions: 95 °C for 3 min, cycling at 98 °C for 20 s, 65 °C for 20 s and 72 °C for 3 min, 15 times, followed by 5 min at 72 °C. The reaction was removed once the qPCR signal began to plateau. The PCR product was purified with 0.8× Ampure XP beads and eluted in 20 µl of nuclease-free H$_2$O.

A Nextera XT Library Prep Kit was used for library preparation. Purified cDNA (1 ng) was diluted in RNase-free water to a total volume of 5 µl, then 10 µl of Tagment DNA buffer and 5 µl of Amplicon Tagment mix were added with incubation at 55 °C for 5 min; 5 µl of NT buffer was then added, with incubation at room temperature for 5 min. PCR master solution (15 µl of PCR master mix, 1 µl of 10 µM P5 primer, 1 µl of 10 µM indexed P7 primer, 8 µl of RNase-free water) was added and the PCR reaction performed with the following programme: 95 °C for 30 s, cycling at 95 °C for 10 s, 55 °C for 30 s, 72 °C for 30 s and 72 °C for 5 min, for 12 cycles. The PCR product was purified with 0.7× Ampure XP beads to obtain the library.

An Agilent Bioanalyzer High Sensitivity Chip was used to determine size distribution and concentration of the library before sequencing. NGS was conducted on an Illumina NovaSeq 6000 sequencer (paired-end, 150-base-pair mode).

### Spatial CUT&Tag–RNA-seq

The frozen tissue slide was thawed for 10 min at room temperature. Tissue was fixed with formaldehyde (0.2%, with 0.05 U µl$^{-1}$ RNase inhibitor) for 5 min and quenched with 1.25 M glycine for a further 5 min. After fixation, the tissue was washed twice with 1 ml of wash buffer (150 mM NaCl, 20 mM HEPES pH 7.5, one tablet of protease inhibitor cocktail, 0.5 mM Spermidine) and dipped in DI water. The tissue section was permeabilized with NP40-digitonin wash buffer (0.01% digitonin, 0.01% NP40 in wash buffer) for 5 min. The primary antibody (1:50 dilution with antibody buffer (0.001% BSA, 2 mM EDTA in NP40-digitonin wash buffer) was added with incubation at 4 °C overnight. The secondary antibody (guinea pig anti-rabbit IgG, 1:50 dilution with NP40-digitonin wash buffer) was added with incubation for 30 min at room temperature. The tissue was then washed with wash buffer for 5 min. A 1:100 dilution of pA-Tn5 adaptor complex in 300-wash buffer (one tablet of Protease inhibitor cocktail, 300 mM NaCl, 0.5 mM Spermidine, 20 mM HEPES pH 7.5) was added with incubation at room temperature for 1 h, followed by a 5 min wash with 300-wash buffer. Tagmentation buffer (10 mM MgCl$_2$ in 300-wash buffer) was added with incuation at 37 °C for 1 h. Next, 40 mM EDTA with 0.05 U µl$^{-1}$ RNase inhibitor was added with incubation at room temperature for 5 min to stop tagmentation. Tissue was washed twice with 0.5× DPBS-RI for 5 min for further use.

For RT, two ligations and bead separation the protocols were as for spatial ATAC–RNA-seq. For construction of the CUT&Tag library, the supernatant was purified with Zymo DNA Clean & Concentrator-5 and eluted to 20 µl of RNase-free water. PCR solution (2 µl each of 10 µM P5 PCR primer and indexed i7 primer, 25 µl of NEBNext Master Mix) was added and mixed well. PCR was performed with the following programme: 58 °C for 5 min, incubation at 72 °C for 5 min and 98 °C for 30 s then cycling at 98 °C for 10 s, with incubation at 60 °C for 10 s 12 times and a final incubation at 72 °C for 1 min. The PCR product was purified

by 1.3× Ampure XP beads using the standard protocol and eluted in 20 μl of nuclease-free $H_2O$. cDNA library construction followed the spatial ATAC–RNA-seq protocol.

An Agilent Bioanalyzer High Sensitivity Chip was used to determine size distribution and concentration of the library before sequencing. NGS was conducted on an Illumina NovaSeq 6000 sequencer (paired-end, 150-base-pair mode).

## Data preprocessing

For ATAC and CUT&Tag data, linkers 1 and 2 were used to filter read 2 and sequences were converted to Cell Ranger ATAC v.1.2 format (10X Genomics). Genome sequences were in the newly formed read 1, and barcodes A and B were included in newly formed read 2. Human reference (GRCh38) or mouse reference (GRCm38) was used to align fastq files. The BED-like fragments thus obtained were used to conduct downstream analysis. The fragments file includes fragments of information on spatial locations (barcode A × barcode B) and the genome.

For RNA data, read 2 was refined to extract barcode A, barcode B and UMI. ST pipeline v.1.7.2 (ref. [53]) was used to map the processed read 1 against the mouse genome (GRCm38) or human genome (GRCh38), which created the gene matrix for downstream analysis that contains information on genes and spatial locations (barcode A × barcode B).

## Data clustering and visualization

We first identified the location of pixels on tissue from the brightfield image using MATLAB 2020b (https://github.com/edicliuyang/Hiplex_proteome).

Signac v.1.8 (ref. [54]) was loaded in R v.4.1. ATAC, CUT&Tag and RNA matrices were read into Signac v.1.8 (ref. [54]). The 'DefaultAssay' function was used for the RNA assay. For RNA data visualization, the feature was set to 3,000 with the 'FindVariableFeatures' function, then data were normalized using the 'SCTransform' function. Normalized RNA data were clustered and RNA UMAP was built. The DefaultAssay function was applied to the ATAC/CUT&Tag assay. For ATAC/CUT&Tag data visualization, minimum cutoff was set with the 'FindTopFeatures' function. Data were normalized and dimensionally reduced using latent semantic indexing, then ATAC/CUT&Tag data were clustered and ATAC/CUT&Tag UMAP was built.

The DefaultAssay function was used for the joint ATAC/CUT&Tag and RNA assay. For visualization of joint ATAC/CUT&Tag and RNA data[55], the 'FindMultiModalNeighbors' function was used. The reduction list was set to ('pca', 'lsi'), the dimensions list was set to that for RNA and ATAC/CUT&Tag, the modality weight.name was set to RNA weight and the joint UMAP was built.

To plot the above-generated UMAPs together, DefaultAssay was set to RNA and the UMAPs for ATAC/CUT&Tag, RNA or joint ATAC/CUT&Tag and RNA were visualized separately using 'DimPlot'.

In regard to RNA spatial data visualization, the gene matrix obtained from RNA was loaded into Seurat v.4.1 (ref. [56]) as a Seurat object, and RNA metadata obtained from Signac were read into the Seurat object. All spatial maps were then plotted with the 'SpatialPlot' function.

In regard to ATAC/CUT&Tag spatial data visualization, the fragment file obtained from ATAC/CUT&Tag was read into ArchR v.1.0.1 (ref. [13]) as an ArchRProject and the ATAC/CUT&Tag metadata obtained from Signac were read into the ArchRProject. The data from ArchRProject were normalized and dimensionally reduced using iterative latent semantic indexing. For GAS and CSS calculation we used the Gene Score model in ArchR. A gene score matrix was obtained for downstream analysis. The 'getMarkerFeatures' and 'getMarkers' functions in ArchR (testMethod = "Wilcoxon", cutOff = "FDR <= 0.05", groupBy = "seurat_cluster") were used to find the marker genes/regions for each cluster. To visualize spatial data, results obtained from ArchR were input to Seurat v.4.1 to map the data back to the tissue. Pixel size was scaled using the 'pt.size.factor' parameter in the Seurat package for better visualization.

For peak-to-gene links we input RNA Seurat object using the 'addGeneIntegrationMatrix' function in ArchR, then peak-to-gene links were drawn with the 'addPeak2GeneLinks' function. Co-accessibility of peaks was calculated using the 'addCoAccessibility' function in ArchR.

## Integrative data analysis and cell type identification

Seurat v.4.1 (ref. [56]) was used for RNA data integration and cell type identification, and the 'SCTransform' function to normalize our spatial RNA and scRNA-seq data. The 'SelectIntegrationFeatures' function was used to obtain features common to the two datasets. The 'FindIntegrationAnchors' function was applied to find anchors, and the 'IntegrateData' function to create an integrated dataset through the identified anchors. The obtained integrated dataset was clustered, showing a good match between our spatial RNA and scRNA-seq data. The 'FindTransferAnchors' function was used to find transfer anchors, which were then used to conduct label transfer with the 'TransferData' function (if more than one cell type was presented in one pixel, the major cell type was assigned).

Signac v.1.8 and Seurat v.4.1 were used for integration of our ATAC/CUT&Tag and scATAC-seq/scCUT&Tag data. The scATAC-seq/scCUT&Tag data were quantified according to our ATAC/CUT&Tag data to ensure that there were features common across both datasets. The FindIntegrationAnchors function (reduction = "rlsi") was used to identify anchors between the two datasets. The 'IntegrateEmbeddings' function was used to obtain an integrated dataset through the identified anchors. The obtained integrated dataset was clustered, showing a good match between our spatial ATAC/CUT&Tag and scATAC-seq/scCUT&Tag data. For ATAC data, the FindTransferAnchors function was used to find transfer anchors, which were then used to map scATAC-seq to our spatial ATAC data with the 'MapQuery' function.

ArchR v.1.0.1 was used for cell type identification for our ATAC/CUT&Tag data from scRNA-seq data. The gene score matrix of our ATAC/CUT&Tag was compared with the gene expression matrix from scRNA-seq, and aligned pixels from our ATAC/CUT&Tag data with cells from scRNA-seq. The function 'GeneIntegrationMatrix' was used to add pseudo-scRNA-seq profiles and cell identities.

## Correlation of CSS/GAS and gene expression

Correlation analysis was performed for different tissue regions. The mouse brain hemisphere was separated into seven clusters (corpus callosum, striatum, superficial cortical layer, deeper cortical layer, lateral ventricle, lateral septal nucleus and others) according to RNA clusters and anatomical annotation, and named 'tissue_clusters'. For Fig. 4a–c, the 'FindMarkers' function (settings: min.pct = 0.25, logfc.threshold = 0.25) was used to calculate our RNA data, and genes with adjusted $P < 10^{-5}$ selected as marker genes. The getMarkerFeatures function (settings: groupBy = "tissue_clusters") was applied to calculate the GAS or CSS of genes from the marker gene list (cutOff = "FDR <= 0.05" & (cutOff = "Log2FC >= 0.1" or cutOff = "Log2FC <= −0.1")). If avg_log2FC > 0 (RNA) and Log2FC > 0 (CUT&Tag) for a specific gene, it showed in quadrant I. GO enrichment analysis was conducted with the 'enrichGO' function (qvalueCutoff = 0.05) in the clusterProfiler v.4.2 package[25].

## Reporting summary

Further information on research design is available in the Nature Portfolio Reporting Summary linked to this article.

# Data availability

Raw and processed data reported in this paper are deposited in the Gene Expression Omnibus with accession code GSE205055. These datasets are available as web resources and can be browsed within the tissue spatial coordinates in the UCSC Cell and Genome Browser (https://brain-spatial-omics.cells.ucsc.edu), and in our own data portal generated with AtlasXplore (https://web.atlasxomics.com/visualization/

Fan). Data are also available at https://ki.se/en/mbb/oligointernode. The resulting fastq files were aligned to either the human reference genome (GRCh38) (https://hgdownload.soe.ucsc.edu/goldenPath/hg38/chromosomes/) or mouse reference genome (GRCm38) (https://hgdownload.soe.ucsc.edu/goldenPath/mm10/chromosomes/). Published data for integration and quality comparison are available online: ENCODE ATAC-seq (E13.5 mouse embryo) (https://www.encodeproject.org/search/?type=Experiment&status=released&related_series.@type=OrganismDevelopmentSeries&replicates.library.biosample.organism.scientific_name=Mus+musculus&assay_title=ATAC-seq&life_stage_age=embryonic%2013.5%20days); ENCODE RNA-seq (E13.5 mouse embryo): forebrain (https://www.ncbi.nlm.nih.gov/geo/query/acc.cgi?acc=GSE78493); midbrain (https://www.ncbi.nlm.nih.gov/geo/query/acc.cgi?acc=GSE78456); hindbrain (https://www.ncbi.nlm.nih.gov/geo/query/acc.cgi?acc=GSE78481); mouse organogenesis cell atlas (https://oncoscape.v3.sttrcancer.org/atlas.gs.washington.edu.mouse.rna/downloads); atlas of gene regulatory elements in adult mouse cerebrum (http://catlas.org/mousebrain/#!/downloads); atlas of the adolescent mouse brain (http://mousebrain.org/adolescent/downloads.html); mouse brain scCUT&Tag H3K27ac data (https://www.ncbi.nlm.nih.gov/geo/query/acc.cgi?acc=GSM5949207); mouse brain scCUT&Tag H3K27me3 data (https://www.ncbi.nlm.nih.gov/geo/query/acc.cgi?acc=GSM5949205); mouse brain scCUT&Tag H3K4me3 data (https://www.ncbi.nlm.nih.gov/geo/query/acc.cgi?acc=GSE163532); human hippocampus (snRNA-seq) (https://www.ncbi.nlm.nih.gov/geo/query/acc.cgi?acc=GSE186538; and human hippocampus (scATAC-seq) (https://www.ncbi.nlm.nih.gov/geo/query/acc.cgi?acc=GSE147672). Source data are provided with this paper.

## Code availability

Code for sequencing data analysis is available at Github (https://github.com/di-0579/Spatial_epigenome-transcriptome_co-sequencing) and archived at Zenodo (https://doi.org/10.5281/zenodo.7395313).

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

**Acknowledgements** We thank the Yale Center for Research Computing for guidance and use of the research computing infrastructure. The moulds for microfluidic devices were fabricated at the Yale University School of Engineering and Applied Science Nanofabrication Center. NGS was conducted at the Yale Center for Genome Analysis and at the Yale Stem Cell Center Genomics Core Facility, supported by the Connecticut Regenerative Medicine Research Fund and the Li Ka Shing Foundation. Service provided by the Genomics Core of Yale Cooperative Center of Excellence in Hematology (no. U54DK106857) was used. We thank T. Jimenez-Beristain for writing laboratory animal ethics permits and assistance with animal experiments, and the staff at Comparative Medicine-Biomedicum. M. Bartosovic was funded by the Vinnova Seal of Excellence Marie-Sklodowska Curie Actions grant RNA-centric view on oligodendrocyte lineage development (RODent). E.A. was funded by the European Union, Horizon 2020, Marie-Sklodowska Curie Actions and the grant SOLO (no. 794689). Work in G.C.-B.'s research group was supported by the Swedish Research Council (grant no. 2019-01360), the Swedish Cancer Society (Cancerfonden, no. 190394 Pj), the Knut and Alice Wallenberg Foundation (grant nos. 2019-0107 and 2019-0089), The Swedish Society for Medical Research (grant no. JUB2019), the Göran Gustafsson Foundation for Research in Natural Sciences and Medicine, the Ming Wai Lau Centre for Reparative Medicine and Karolinska Institutet. This research was supported by Packard Fellowship for Science and Engineering (to R.F.), Yale Stem Cell Center Chen Innovation Award (to R.F.) and the US National Institutes of Health (nos. RF1MH128876, U54AG076043, U54AG079759, UG3CA257393, UH3CA257393, R01CA245313 and U54CA274509 to R.F.). Y.L. was supported by the Society for ImmunoTherapy of Cancer Fellowship.

**Author contributions** R.F. conceptualized the study. D.Z., Y.D. and Y.L. were responsible for the methodology. D.Z. and Y.D. carried out the experimental investigation. D.Z., Y.D., P.K., E.A., M. Bartosovic, M.C., C.M., S.M., B.J.R., Y.K., G.C.-B. and R.F. were responsible for data analysis. P.K., G.S., S.B., Y.X., K.W.L., G.B.R., A.J.D., J.J.M. and M. Boldrini were responsible for getting resources. L.W. and M.H. were responsible for the data browser. D.Z., Y.D. and R.F. wrote the original draft. All authors reviewed, edited and approved the manuscript.

**Funding** Open access funding provided by Karolinska Institute.

**Competing interests** R.F., D.Z. and Y.D. are inventors of a patent provisional disclosure related to this work. R.F. is scientific founder and advisor of IsoPlexis, Singleron Biotechnologies and AtlasXomics. The interests of R.F. were reviewed and managed by Yale University Provost's Office in accordance with the university's conflict of interest policies. The remaining authors declare no competing interests.

**Additional information**
**Correspondence and requests for materials** should be addressed to Yanxiang Deng, Gonçalo Castelo-Branco or Rong Fan.

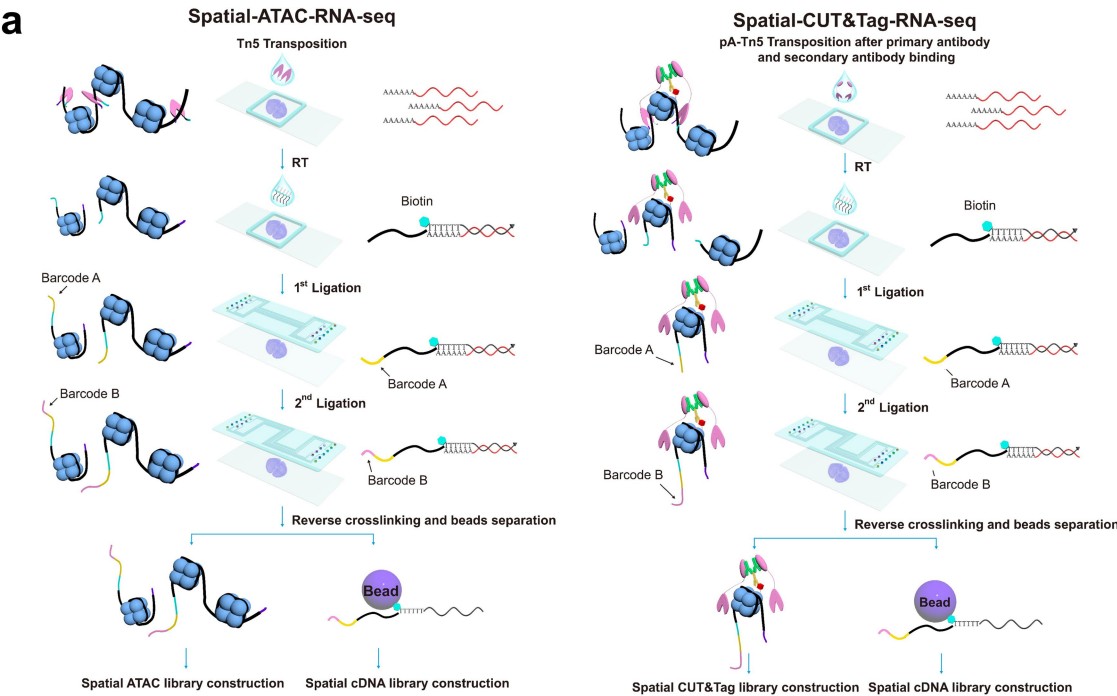

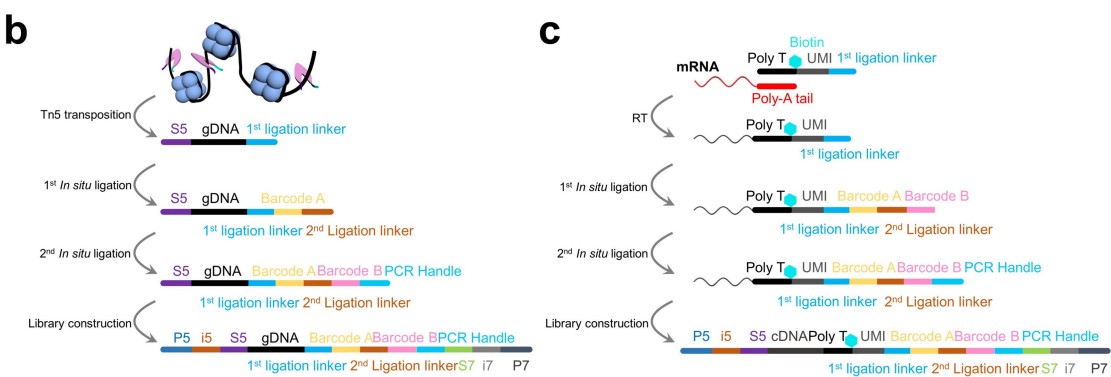

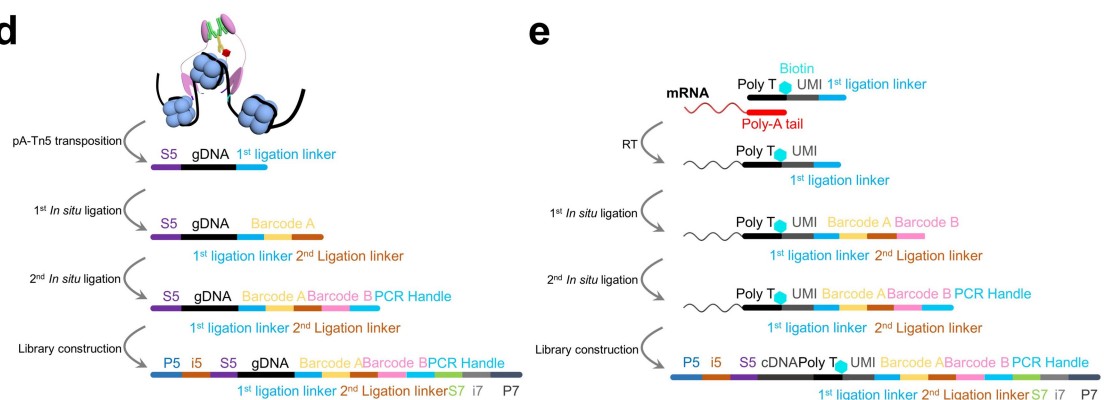

**Extended Data Fig. 1 | Workflow of spatial-ATAC-RNA-seq and spatial-CUT &Tag-RNA-seq. a**, Schematic workflow of spatial-ATAC-RNA-seq and spatial-CUT&Tag-RNA-seq. **b–c**, Chemistry workflow of ATAC (**b**) and RNA (**c**) in spatial-ATAC-RNA-seq. **d–e**, Chemistry workflow of CUT&Tag (**d**) and RNA (**e**) in spatial-CUT&Tag-RNA-seq.

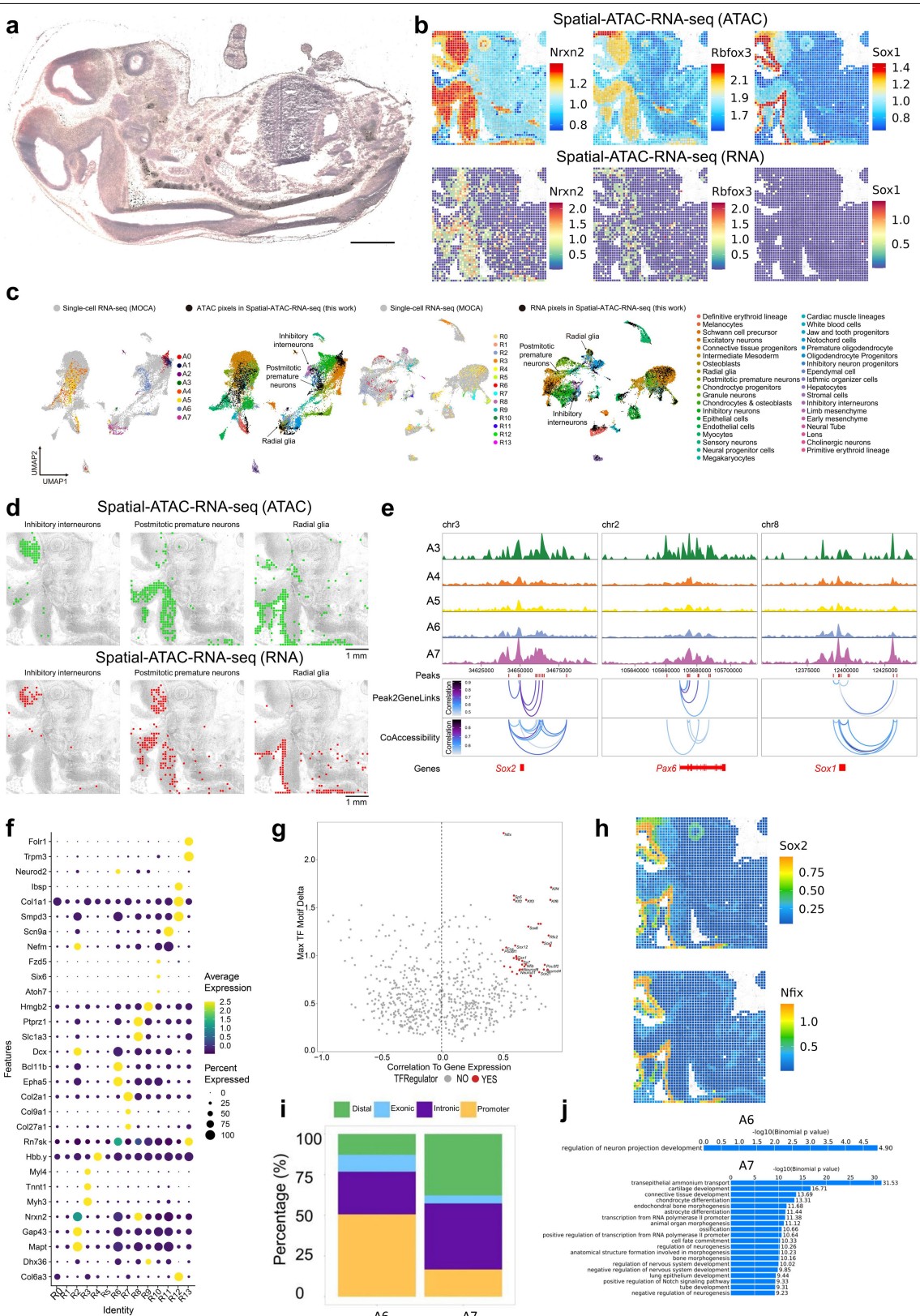

**Extended Data Fig. 2 | Further analysis of spatial-ATAC-RNA-seq for E13 mouse embryo. a**, H&E image from an adjacent tissue section of E13 mouse embryo. **b**, Spatial mapping of GAS and gene expression for selected marker genes in spatial-ATAC-RNA-seq. **c**, Integration of scRNA-seq data[12] from E13.5 mouse embryos with ATAC and RNA data in spatial ATAC-RNA-seq. MOCA, Mouse Organogenesis Cell Atlas. **d**, Spatial mapping of cell types identified by label transfer from scRNA-seq[12] to ATAC (top) and RNA (bottom). **e**, Genome track visualization of marker genes with peak-to-gene links for distal regulatory elements and peak co-accessibility. **f**, The expression level and the percentage of pixels in all clusters (marker genes for each cluster) for RNA data in spatial-ATAC-RNA-seq. **g**, Dot plot showing the identification of positive TF regulators. **h**, Spatial mapping of deviation scores for selected TF motifs. **i**, Annotation of marker peaks in different clusters. **j**, GREAT enrichment analysis of marker peaks in different clusters (Binomial and hypergeometric tests).

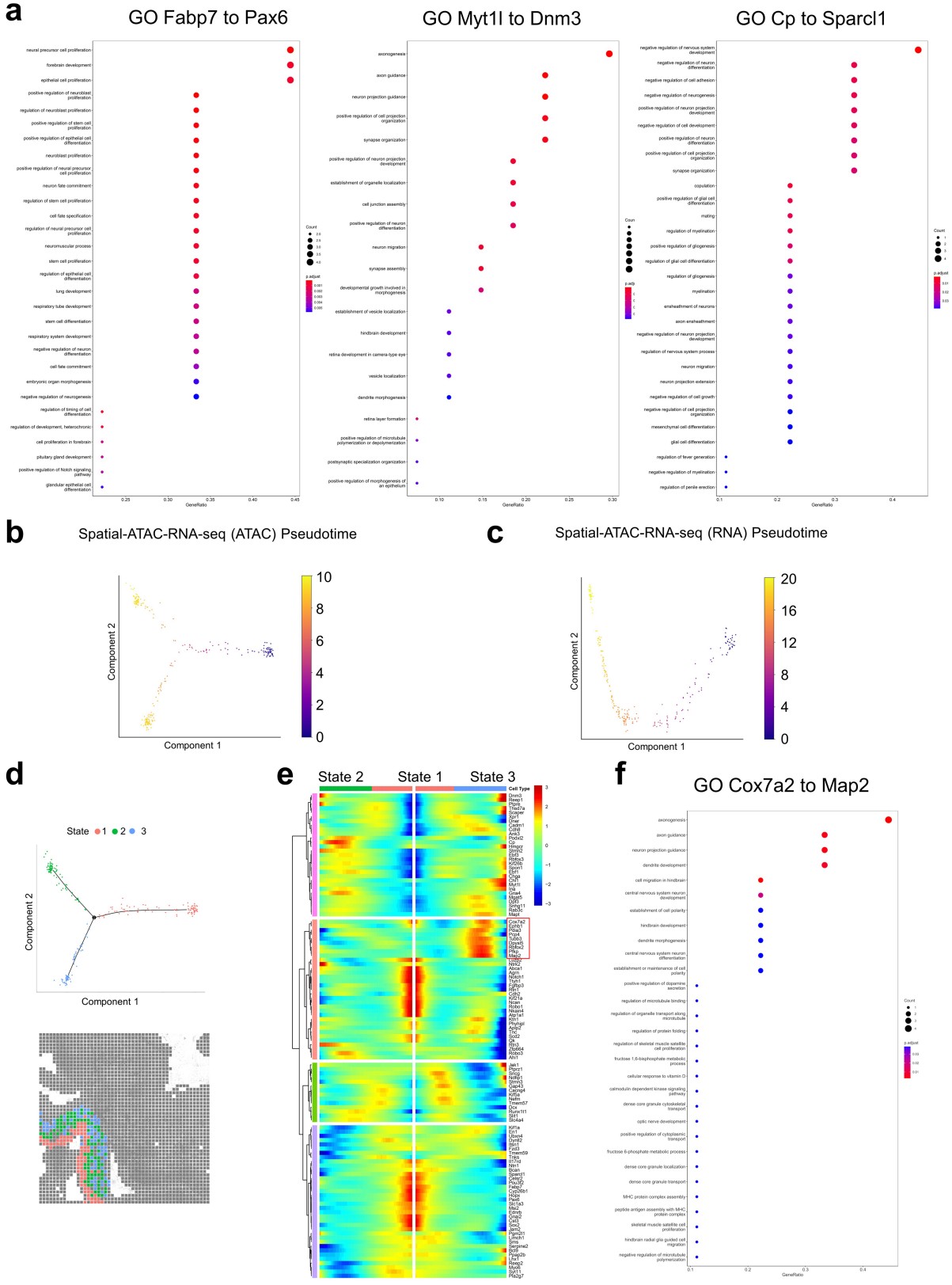

**Extended Data Fig. 3 | Further pseudotime analysis of radial glia and postmitotic premature neurons in spatial-ATAC-RNA-seq. a**, GO enrichment analysis for genes from Fig. 1g. **b,c**, Pseudotime analysis from radial glia to postmitotic premature neurons with GAS (**b**) and gene expression (**c**). **d**, Monocle2 analyses showing different states in (**b**). **e**, Heatmap of different states along the pseudotime trajectory. **f**, GO analysis of genes in red box of (**e**) (One-sided version of Fisher's exact test, p-value was adjusted for multiple comparisons by Benjamini & Hochberg method).

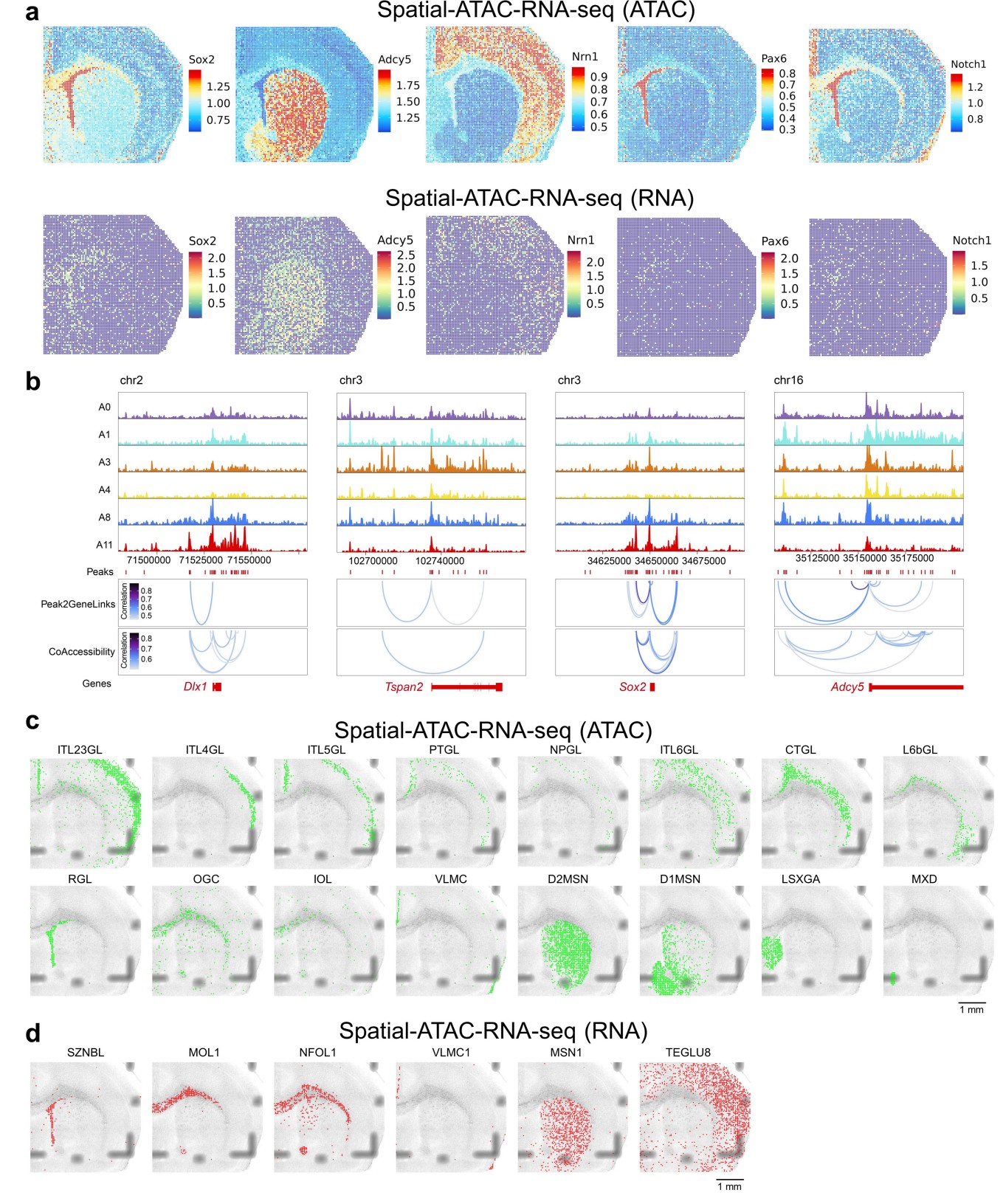

**Extended Data Fig. 4 | Further analysis of spatial-ATAC-RNA-seq for P22 mouse brain. a**, Spatial mapping of gene activity scores and gene expression for selected marker genes in spatial-ATAC-RNA-seq. **b**, Genome track visualization of marker genes with peak-to-gene links for distal regulatory elements and peak co-accessibility. **c**, Spatial mapping of cell types identified by label transfer from scATAC-seq[30] to ATAC data. IT: intratelencephalic. PT: pyramidal tract. NP: near-projecting. CT: corticothalamic. L: layer. **d**, Spatial mapping of cell types identified by label transfer from scRNA-seq[32] to RNA data.

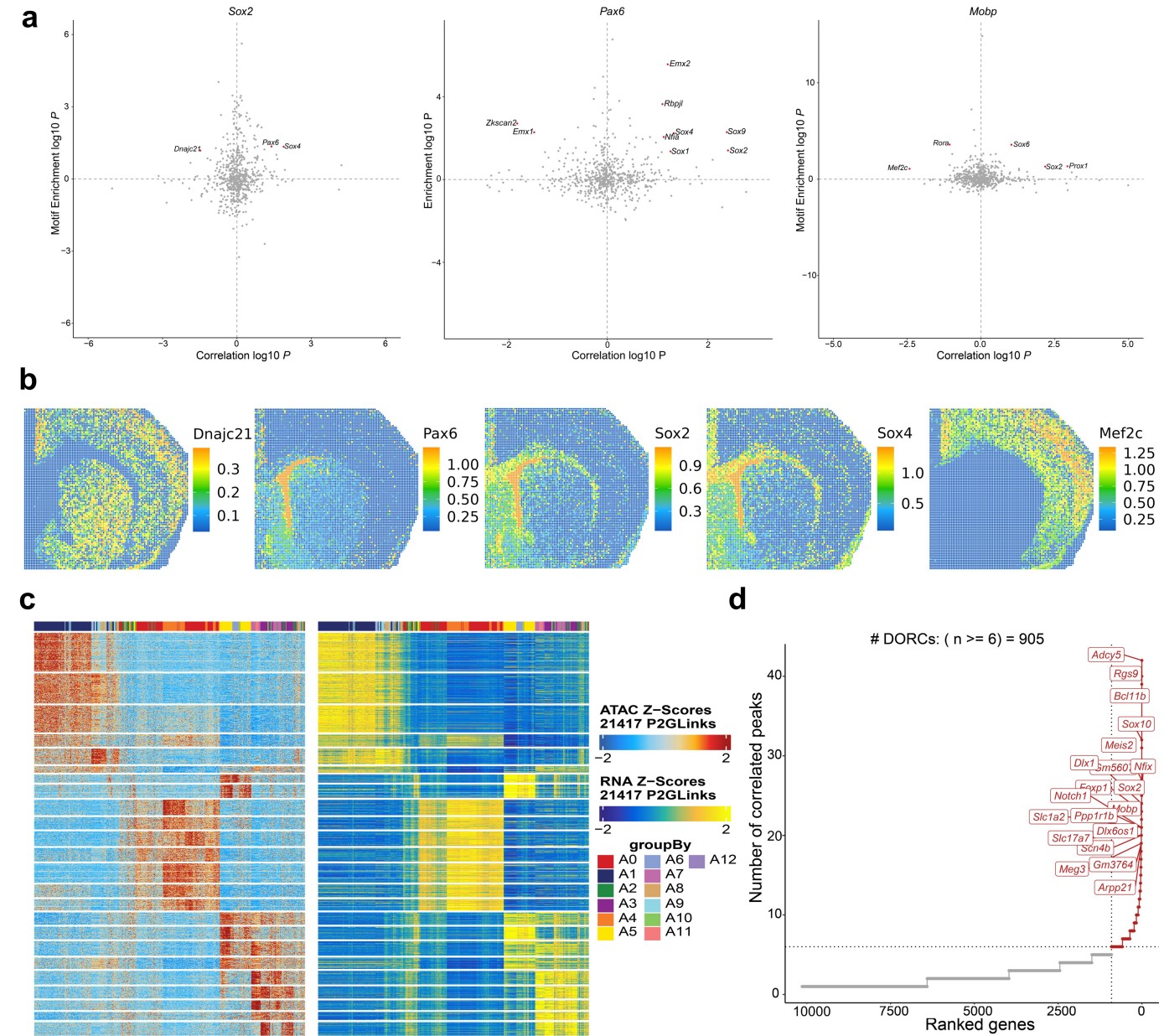

**Extended Data Fig. 5 | Further analysis of P22 mouse brain in spatial-ATAC-RNA-seq. a**, Candidate TF regulators of *Sox2*, *Pax6*, and *Mobp*. Highlighted points are TFs with abs(regulation score) ≥1 (−log10 scale)[33], with all other TFs shown in gray (*Z* test). **b**, Spatial mapping of deviation scores for selected TF motifs. **c**, Heatmaps of peak-to-gene links in spatial-ATAC-RNA-seq. **d**, The number of significantly correlated peaks for each gene.

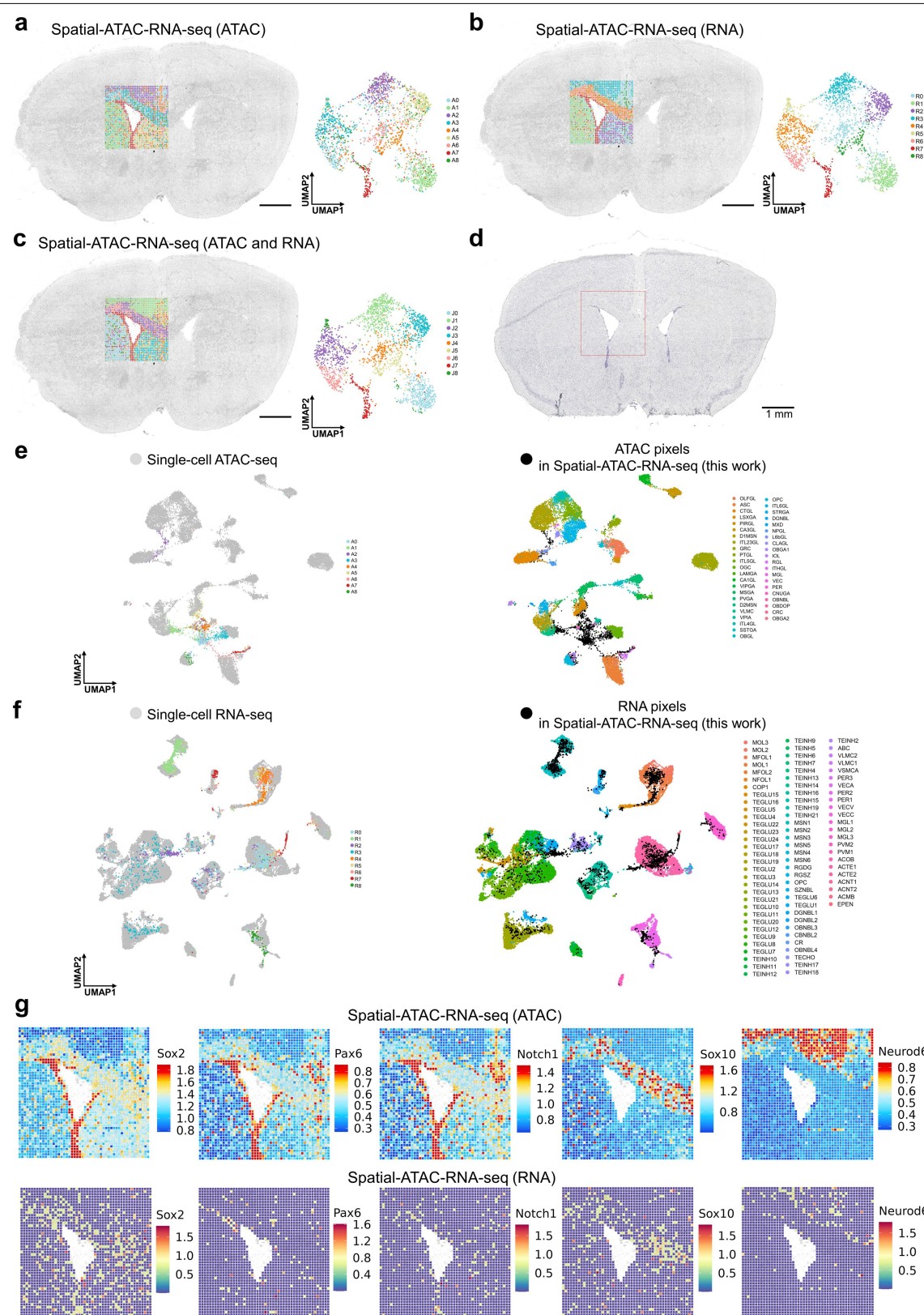

**Extended Data Fig. 6 | Spatial chromatin accessibility and transcriptome co-sequencing of P21 mouse brain. a–c**, Spatial distribution and UMAP of all the clusters for ATAC (**a**), RNA (**b**), and joint clustering of ATAC and RNA (**c**) in spatial-ATAC-RNA-seq for the mouse brain. Pixel size, 20 μm. Scale bar, 1 mm. **d**, Nissl-stained image from an adjacent tissue section of P21 mouse brain. Scale bar, 1 mm. **e**, Integration of ATAC data in spatial-ATAC-RNA-seq with scATAC-seq data[30] from mouse brain. **f**, Integration of RNA data in spatial-ATAC-RNA-seq with scRNA-seq data[32] from mouse brain. **g**, Spatial mapping of GAS and gene expression for selected marker genes in different clusters for ATAC and RNA in spatial-ATAC-RNA-seq.

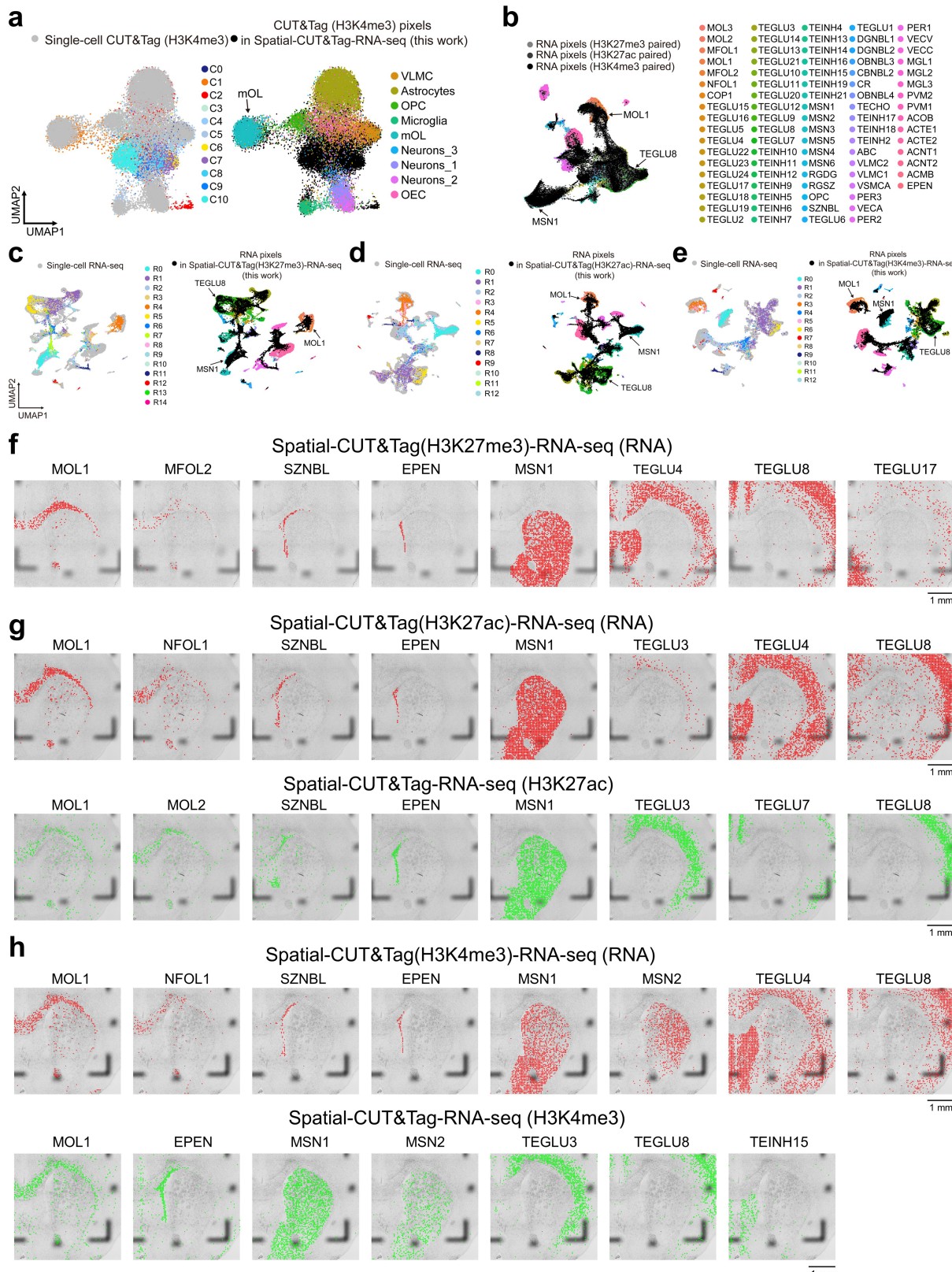

**Extended Data Fig. 7** | See next page for caption.

**Extended Data Fig. 7 | Further analysis for spatial-CUT&Tag-RNA-seq with P22 mouse brain. a**, Integration of CUT&Tag (H3K4me3) data in spatial-CUT&Tag-RNA-seq with scCUT&Tag (H3K4me3) data[36] from mouse brain. **b**, Integration of RNA data in spatial- CUT&Tag(H3K27me3)-RNA-seq, spatial-CUT&Tag (H3K27ac)-RNA-seq, and spatial-CUT&Tag(H3K4me3)-RNA-seq with scRNA-seq data[32] from mouse brain. **c-e**, Integration of RNA data in spatial-CUT&Tag (H3K27me3)-RNA-seq (**c**), RNA data in data in spatial-CUT&Tag(H3K27ac)-RNA-seq (**d**), and RNA data in spatial-CUT&Tag(H3K4me3)-RNA-seq (**e**) with scRNA- seq data[32] from mouse brain. **f**, Spatial mapping of cell types identified by label transfer from scRNA-seq[32] to RNA data in spatial-CUT&Tag(H3K27me3)-RNA-seq. **g**, Spatial mapping of cell types identified by label transfer from scRNA-seq[32] to RNA (top) and from scRNA-seq[32] to CUT&Tag (H3K27ac, bottom) data in spatial-CUT&Tag(H3K27ac)-RNA-seq. **h**, Spatial mapping of cell types identified by label transfer from scRNA-seq[32] to RNA (top) and from scRNA-seq[32] to CUT&Tag (H3K4me3, bottom) data in spatial-CUT&Tag(H3K4me3)-RNA-seq.

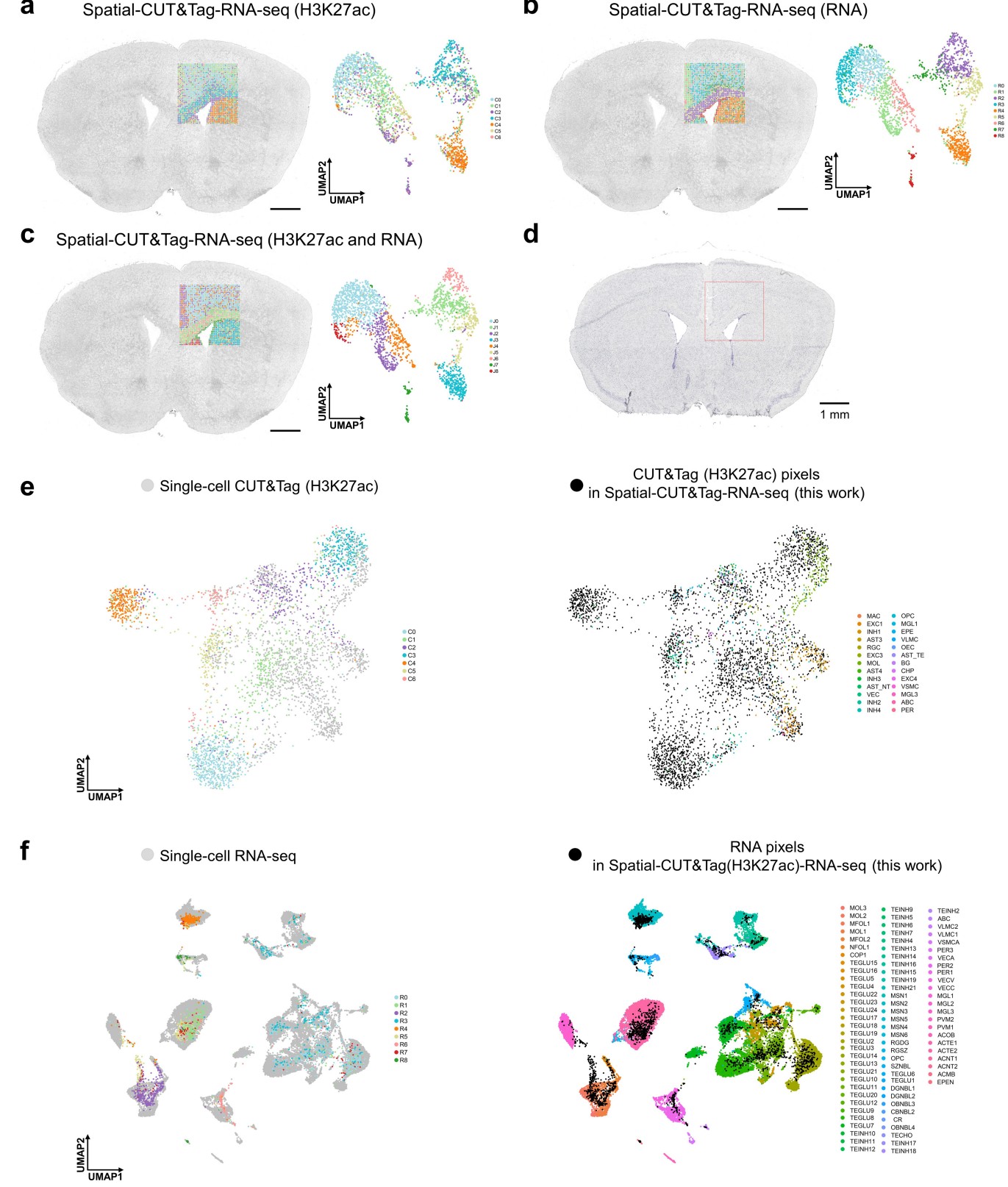

**Extended Data Fig. 8 | Spatial epigenome and transcriptome co-sequencing and integrative analysis of P21 mouse brain. a-c**, Spatial distribution and UMAP of all the clusters for CUT&Tag (H3K27ac) (**a**), RNA (**b**), and joint clustering of CUT&Tag (H3K27ac) and RNA (**c**) in spatial-CUT&Tag-RNA-seq for the mouse brain. Pixel size, 20 μm. Scale bar, 1 mm. **d**, Nissl-stained image from an adjacent tissue section of P21 mouse brain. Scale bar, 1 mm. **e**, Integration of CUT&Tag (H3K27ac) data in spatial-CUT&Tag-RNA-seq with scCUT&Tag (H3K27ac) data[37] from mouse brain. **f**, Integration of RNA data in spatial-CUT&Tag-RNA-seq with scRNA-seq data[32] from mouse brain.

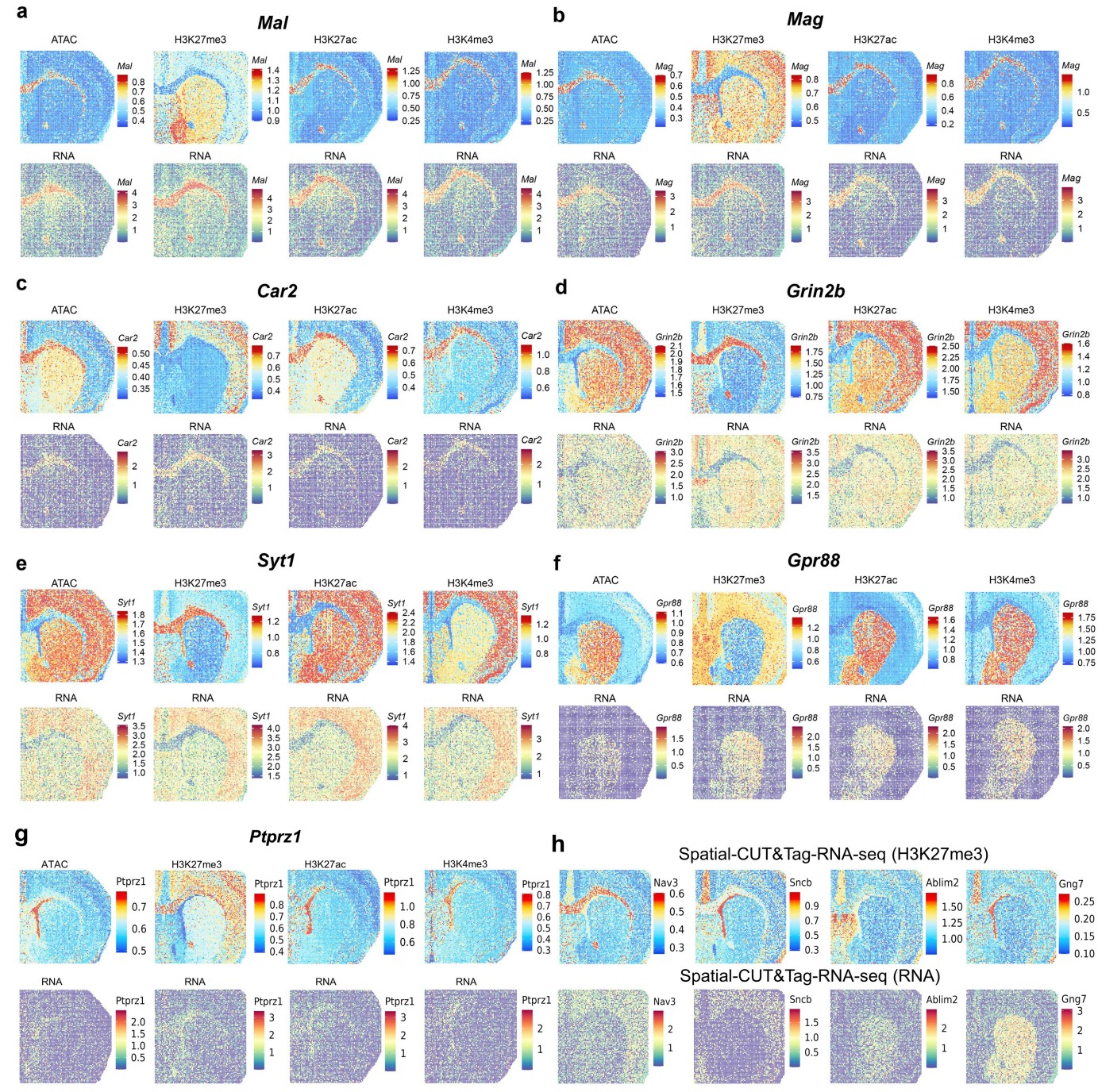

**Extended Data Fig. 9 | Spatial mapping of CSS, GAS, and gene expression of selected genes for P22 mouse brain. a-g**, Spatial mapping of CSS, GAS, and gene expression of *Mal* (**a**), *Mag* (**b**), *Car2* (**c**), *Grin2b* (**d**), *Syt1* (**e**), *Gpr88* (**f**), and *Ptprz1* (**g**) from ATAC and RNA in spatial-ATAC-RNA-seq, and CUT&Tag (H3K27me3, H3K27ac, or H3K4me3) and RNA in spatial-CUT&Tag-RNA-seq. **h**, Spatial mapping of CSS and gene expression of *Nav3*, *Sncb*, *Ablim2*, and *Gng7* in spatial-CUT&Tag(H3K27me3)-RNA-seq.

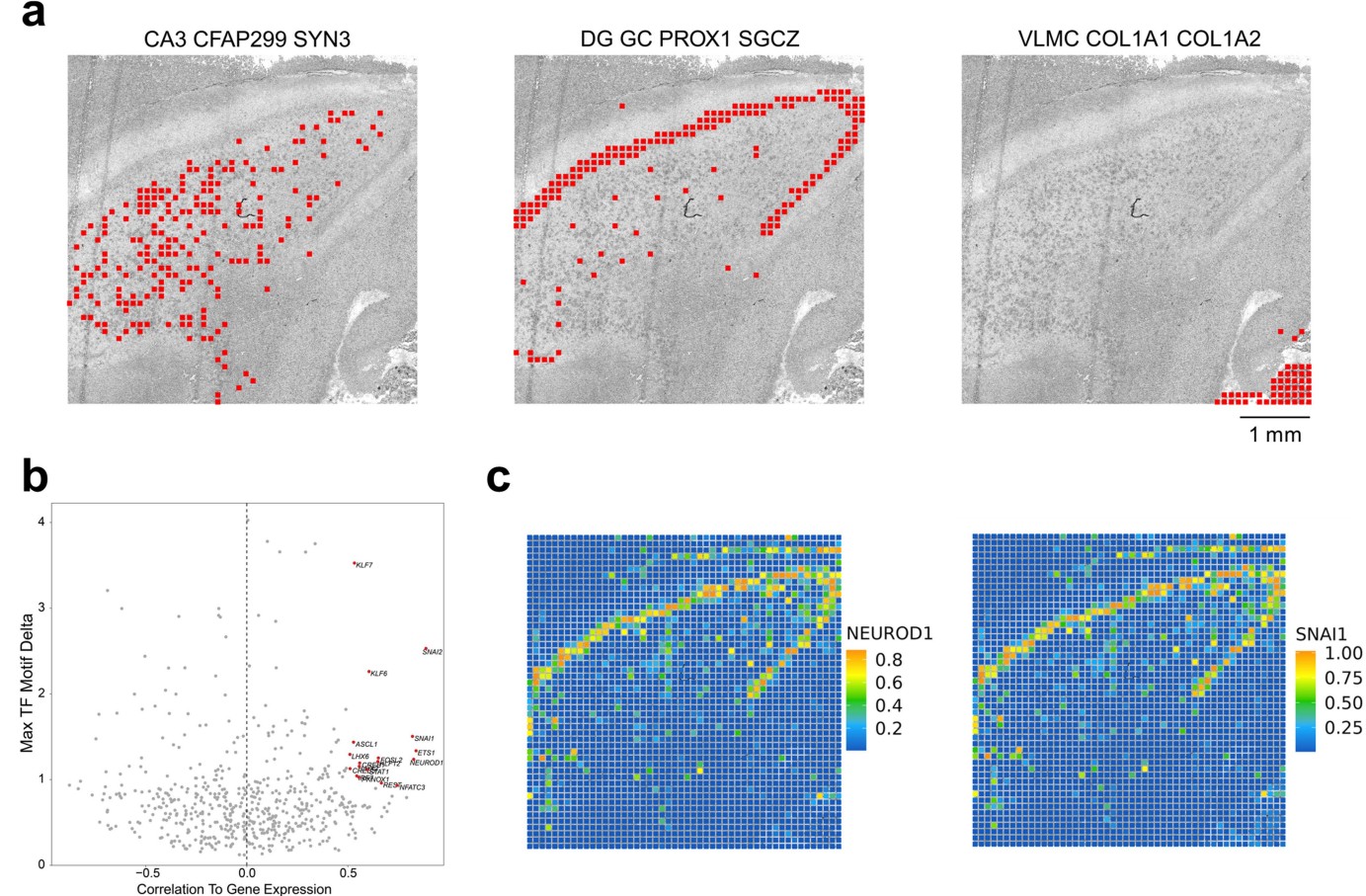

**Extended Data Fig. 10 | Further analysis of human hippocampus in spatial-ATAC-RNA-seq. a**, Spatial mapping of cell types identified by label transfer from scRNA-seq[43] to RNA data in spatial-ATAC-RNA-seq for human hippocampus. **b**, Dot plot showing the identification of positive TF regulators. **c**, Spatial mapping of deviation scores for selected TF motifs in spatial-ATAC-RNA-seq.

# Reporting Summary

Nature Research wishes to improve the reproducibility of the work that we publish. This form provides structure for consistency and transparency in reporting. For further information on Nature Research policies, see our Editorial Policies and the Editorial Policy Checklist.

## Statistics

For all statistical analyses, confirm that the following items are present in the figure legend, table legend, main text, or Methods section.

| n/a | Confirmed | |
|---|---|---|
| ☐ | ☒ | The exact sample size (*n*) for each experimental group/condition, given as a discrete number and unit of measurement |
| ☐ | ☒ | A statement on whether measurements were taken from distinct samples or whether the same sample was measured repeatedly |
| ☐ | ☒ | The statistical test(s) used AND whether they are one- or two-sided *Only common tests should be described solely by name; describe more complex techniques in the Methods section.* |
| ☐ | ☒ | A description of all covariates tested |
| ☐ | ☒ | A description of any assumptions or corrections, such as tests of normality and adjustment for multiple comparisons |
| ☐ | ☒ | A full description of the statistical parameters including central tendency (e.g. means) or other basic estimates (e.g. regression coefficient) AND variation (e.g. standard deviation) or associated estimates of uncertainty (e.g. confidence intervals) |
| ☐ | ☒ | For null hypothesis testing, the test statistic (e.g. *F*, *t*, *r*) with confidence intervals, effect sizes, degrees of freedom and *P* value noted *Give P values as exact values whenever suitable.* |
| ☒ | ☐ | For Bayesian analysis, information on the choice of priors and Markov chain Monte Carlo settings |
| ☒ | ☐ | For hierarchical and complex designs, identification of the appropriate level for tests and full reporting of outcomes |
| ☐ | ☒ | Estimates of effect sizes (e.g. Cohen's *d*, Pearson's *r*), indicating how they were calculated |

*Our web collection on statistics for biologists contains articles on many of the points above.*

## Software and code

Policy information about availability of computer code

| | |
|---|---|
| Data collection | EVOS FL Auto 2 Software Revision 2.0.2094.0, Illumina NovaSeq 6000 System. |
| Data analysis | R 4.1, ArchR 1.0.1, Seurat 4.1, Cell Ranger ATAC 1.2, clusterProfiler 4.2, GREAT 4.0.4, Matlab 2020b, Signac 1.8, Monocle 2.26, ST pipeline 1.7.2 <br><br> Scripts for data analysis were written in R and python with code available at https://github.com/di-0579/Spatial_epigenome-transcriptome_co-sequencing and archived at Zenodo (https://doi.org/10.5281/zenodo.7395313), and https://github.com/edicliuyang/Hiplex_proteome |

For manuscripts utilizing custom algorithms or software that are central to the research but not yet described in published literature, software must be made available to editors and reviewers. We strongly encourage code deposition in a community repository (e.g. GitHub). See the Nature Research guidelines for submitting code & software for further information.

## Data

Policy information about availability of data

All manuscripts must include a data availability statement. This statement should provide the following information, where applicable:
- Accession codes, unique identifiers, or web links for publicly available datasets
- A list of figures that have associated raw data
- A description of any restrictions on data availability

Raw and processed data reported in this paper are deposited in the Gene Expression Omnibus (GEO) with accession code GSE205055. These datasets are available as webresources and can be browsed within the tissue spatial coordinates in UCSC Cell and Genome Browser (https://brain-spatial-omics.cells.ucsc.edu) and our own data portal generated with AtlasXplore (https://web.atlasxomics.com/visualization/Fan).

The resulting fastq files were aligned to the human reference genome (GRCh38) (https://hgdownload.soe.ucsc.edu/goldenPath/hg38/chromosomes/) or mouse reference genome (GRCm38) (https://hgdownload.soe.ucsc.edu/goldenPath/mm10/chromosomes/).

Published data for integration and quality comparison are available online: ENCODE ATAC-seq (E13.5 mouse embryo) (https://www.encodeproject.org/search/?type=Experiment&status=released&related_series.@type=OrganismDevelopmentSeries&replicates.library.biosample.organism.scientific_name=Mus+musculus&assay_title=ATAC-seq&life_stage_age=embryonic%2013.5%20days), ENCODE RNA-seq (E13.5 mouse embryo): Forebrain (https://www.ncbi.nlm.nih.gov/geo/query/acc.cgi?acc=GSE78493);
Midbrain (https://www.ncbi.nlm.nih.gov/geo/query/acc.cgi?acc=GSE78456); Hindbrain (https://www.ncbi.nlm.nih.gov/geo/query/acc.cgi?acc=GSE78481), Mouse organogenesis cell atlas (MOCA) (https://oncoscape.v3.sttrcancer.org/atlas.gs.washington.edu.mouse.rna/downloads), Atlas of gene regulatory elements in adult mouse cerebrum (http://catlas.org/mousebrain/#!/downloads), Atlas of the Adolescent Mouse Brain (http://mousebrain.org/adolescent/downloads.html), Mouse brain scCUT&Tag H3K27ac data (https://www.ncbi.nlm.nih.gov/geo/query/acc.cgi?acc=GSM5949207), Mouse brain scCUT&Tag H3K27me3 data (https://www.ncbi.nlm.nih.gov/geo/query/acc.cgi?acc=GSM5949205), Mouse brain scCUT&Tag H3K4me3 data (https://www.ncbi.nlm.nih.gov/geo/query/acc.cgi?acc=GSE163532), Human hippocampus (snRNA-seq) (https://www.ncbi.nlm.nih.gov/geo/query/acc.cgi?acc=GSE186538), Human hippocampus (scATAC-seq) (https://www.ncbi.nlm.nih.gov/geo/query/a cc.cgi?acc=GSE147672).

# Field-specific reporting

Please select the one below that is the best fit for your research. If you are not sure, read the appropriate sections before making your selection.

☒ Life sciences          ☐ Behavioural & social sciences          ☐ Ecological, evolutionary & environmental sciences

For a reference copy of the document with all sections, see nature.com/documents/nr-reporting-summary-flat.pdf

# Life sciences study design

All studies must disclose on these points even when the disclosure is negative.

| | |
|---|---|
| Sample size | No directly relevant. No sample size calculation was performed. Samples sizes were chosen primarily based on experiment length, sample availability, and sequencing costs. The current manuscript mainly described the new methods for spatial epigenome-transcriptome co-profiling, the sample sizes are sufficient because each sample serves as a proof-of-concept for the new technologies. |
| Data exclusions | No data were excluded from the study. |
| Replication | All attempts at replication were successful. For spatial-ATAC-RNA-seq, two biological replicates have been done on P21 mouse brain to verify the reproducibility of the new technology. For spatial-CUT&Tag(H3K27ac)-RNA-seq, two biological replicates have been done on P21 mouse brain to verify the reproducibility the new technology. Other experiments were performed once to serve as a proof-of-concept for the new technologies. |
| Randomization | Randomization was not applicable because the focus of this paper is the development of new technologies for spatial epigenome-transcriptome co-profiling, and did not involve allocating samples/organisms/participants into experimental groups. |
| Blinding | Blinding was not applicable because the focus of this paper is the development of new technologies for spatial epigenome-transcriptome co-profiling, and did not involve group allocation, and by extension, blinding. |

# Reporting for specific materials, systems and methods

We require information from authors about some types of materials, experimental systems and methods used in many studies. Here, indicate whether each material, system or method listed is relevant to your study. If you are not sure if a list item applies to your research, read the appropriate section before selecting a response.

## Materials & experimental systems

| n/a | Involved in the study |
|---|---|
| ☐ | ☒ Antibodies |
| ☒ | ☐ Eukaryotic cell lines |
| ☒ | ☐ Palaeontology and archaeology |
| ☐ | ☒ Animals and other organisms |
| ☐ | ☒ Human research participants |
| ☒ | ☐ Clinical data |
| ☒ | ☐ Dual use research of concern |

## Methods

| n/a | Involved in the study |
|---|---|
| ☒ | ☐ ChIP-seq |
| ☒ | ☐ Flow cytometry |
| ☒ | ☐ MRI-based neuroimaging |

# Antibodies

| | |
|---|---|
| Antibodies used | Anti-H3K27ac antibody (clone number: EP16602), ab177178, Abcam; Secondary antibody (Guinea Pig anti-Rabbit IgG (Heavy & Light Chain) Antibody), ABIN101961, Antibodies-Online; Histone H3K4me3 antibody (pAb), 39159, Active Motif; Anti-H3K27me3 antibody (clone number: C36B11), 9733, Cell Signaling Technology. |

| Validation | Rabbit monoclonal [EP16602] to Histone H3 (acetyl K27)-ChIP Grade (ab177178), abcam, citation: Science 375, 681–686 (2022); Nat Commun 12:4618 (2021); Front Oncol 11:572585 (2021); Cell Death Dis 12:245 (2021); Cancer Cell 38:334-349.e9 (2020), et al. Please refer to manufacturer's description: https://www.abcam.com/histone-h3-acetyl-k27-antibody-ep16602-chip-grade-ab177178.html?productWallTab=ShowAll<br>Histone H3K4me3 antibody (pAb), 39159, Active Motif, This antibody has been validated for CUT&Tag using Active Motif's CUT&Tag-IT™ Assay Kit, Catalog No. 53160. Please refer to manufacturer's description: https://www.activemotif.com/catalog/details/39159<br>Tri-Methyl-Histone H3 (Lys27) (C36B11) Rabbit mAb #9733, Cell Signaling Technology, citation: Nat Cancer 3(9):1071-1087 (2022); Nat Commun 13(1):5883 (2022); Cancer Discov 12(7):1760-1781 (2022), et al. Please refer to manufacturer's description: https://www.cellsignal.com/products/primary-antibodies/tri-methyl-histone-h3-lys27-c36b11-rabbit-mab/9733<br>Guinea Pig anti-Rabbit IgG (Heavy & Light Chain) Antibody - Preadsorbed, Antibodies-Online, According to the manufacturer's description: ABIN101961 is tested via ELISA to ensure that the titer against the antigen (Rb IgG) is above a certain threshold. We also test to make sure the titer against potentially cross-reactive human IgG, goat IgG, and mouse IgG is below a certain threshold. In addition, we test ABIN101961 against anti-guinea pig Serum, rabbit IgG, and rabbit serum in an immunoelectrophoresis assay. Please refer to manufacturer's description: https://www.antibodies-online.com/antibody/101961/Guinea+Pig+anti-Rabbit+IgG+Heavy++Light+Chain+antibody+-+Preadsorbed/ |
|---|---|

## Animals and other organisms

Policy information about studies involving animals; ARRIVE guidelines recommended for reporting animal research

| Laboratory animals | The mouse line Sox10:Cre-RCE:LoxP (EGFP), on a C57BL/6xCD1 mixed genetic background. Animals of both sexes were used for experiments at P21/P22.<br>P22 (spatial-ATAC-RNA-Seq, 100 barcodes), Male<br>P22 (spatial-CUT&Tag(H3K27ac)-RNA-Seq, 100 barcodes), Female<br>P22 (spatial-CUT&Tag(H3K27me3)-RNA-Seq, 100 barcodes), Male<br>P22 (spatial-CUT&Tag(H3K4me3)-RNA-Seq, 100 barcodes), Female<br>P21 (spatial-ATAC-RNA-Seq, 50 barcodes, replica 1), Male<br>P21 (spatial-CUT&Tag(H3K27ac)-RNA-Seq, 50 barcodes, replica 1), Male<br>P21 (spatial-ATAC-RNA-Seq, 50 barcodes, replica 2), Female<br>P21 (spatial-CUT&Tag(H3K27ac)-RNA-Seq, 50 barcodes, replica 2), Female<br><br>All animals were free from mouse bacterial and viral pathogens, ectoparasites and endoparasites. The following light/dark cycle was kept for the mice: dawn 6:00-7:00, daylight 7:00-18:00, dusk 18:00-19:00, night 19:00-6:00. Mice were housed in individually ventilated cages with a maximum number of 5 per cage (IVC sealsafe GM500, tecniplast). General housing parameters such as temperature, ventilation, and relative humidity followed the European convention for the protection of vertebrate animals used for experimental and other scientific purposes. The air quality was controlled by using the stand-alone air handling units equipped with a HEPA filter. The consistent relative air humidity was 55%±10 with a temperature of 22 °C. The husbandry parameters were monitored with ScanClime® (Scanbur) units. The cages contained card box shelter, gnawing sticks, and nesting material (Scanbur), placed on a hardwood bedding (TAPVEI, Estonia). The mice were provided a regular chow diet and water was supplied with a water bottle and changed weekly. Cages were changed every two weeks in a laminar air-flow cabinet. |
|---|---|
| Wild animals | No wild animals were used in the study. |
| Field-collected samples | No field collected samples were used in the study. |
| Ethics oversight | All experimental procedures were conducted following the European directive 2010/63/EU, local Swedish directive L150/SJVFS/2019:9, Saknr L150, and Karolinska Institutet complementary guidelines for procurement and use of laboratory animals, Dnr. 1937/03-640. All the procedures described were approved by the local committee for ethical experiments on laboratory animals in Sweden (Stockholms Norra Djurförsöksetiska nämnd), lic. nr. 1995/2019 and 7029/2020. |

Note that full information on the approval of the study protocol must also be provided in the manuscript.

## Human research participants

Policy information about studies involving human research participants

| Population characteristics | 31-year-old Caucasian male, with no psychiatric or neurological diagnosis, toxicology negative for psychotropic medications and drugs. |
|---|---|
| Recruitment | We analyzed brain hippocampus tissue from a 31 year-old Caucasian male, with postmortem interval (PMI, time from demise to brain collection) of 6.5 hours, with no psychiatric or neurological diagnosis, who died of a traumatic accident, and had a high global functioning before death as measured by Global Assessment Scale (GAS) score which was 90 (score 1 to 100, with 100 the highest functioning), and with toxicology negative for psychotropic medications and drugs. Biases were not applicable because the focus of this paper is the development of the new methods for spatial epigenome-transcriptome co-profiling, and each sample serves as a proof-of-concept for the new technologies. |
| Ethics oversight | The human brain tissue was obtained from the Brain Collection of the New York State Psychiatric Institute (NYSPI) at Columbia University, which includes brain samples from the Republic of Macedonia. Brain tissue collection was conducted with NYSPI Institutional Review Board approval and informed consent obtained from next of kin who agreed to donate the brains and participated in psychological autopsy interviews. |

Note that full information on the approval of the study protocol must also be provided in the manuscript.

