## [Peer Review File · Nature]

Manuscript Title: Spatial epigenome-transcriptome co-profiling of mammalian tissues

Reviewer Comments & Author Rebuttals

Reviewer Reports on the Initial Version:

Referee expertise:

Referee #1: chromatin methods

Referee #2: spatial genomics

Referee #3: genomics methods, microfluidics

Referees' comments:

Referee #1 (Remarks to the Author):

The authors have extended their powerful dBIT microfluidics strategy for spatial transcriptomics and epigenomics for multi-modal chromatin/transcriptome profiling by adding RNA-seq to both ATAC-seq and CUT&Tag. Since these authors' single-modal spatial methods are already available as publications or preprints (RNA-seq: Cell PMID: 33188776; ATAC-seq: biorxiv 2021.06.06.447244; CUT&Tag: Science PMID: 35143307), I will focus my evaluation on the question of how much the current study adds to those previous studies. Unfortunately, I do not find enough added value to recommend publication in Nature.

1) Although Spatial-ATAC-seq-RNA-seq is described fully, including application to a wider-field microfluidic device, Spatial-CUT&Tag-RNA-seq seems to be last-minute add-on. Spatial-CUT&Tag-RNA-seq data are described in the text and in Figure 1 and interpreted without any indication of the antibody used. Different antibodies for example to different histone modifications report on different parts of epigenome, so the results are meaningless to a reader without knowing what antibody against what epitope was used. Only by Figure 3 is the mystery solved where it is stated that a single primary antibody against H3K27ac was used. This is a promoter/enhancer mark that should give roughly the same profile as ATAC-seq. But users of this method will not be interested in doing spatial-CUT&Tag-RNA-seq to do something that they can do using the spatial-ATAC-seq-RNA-seq, since it requires additional steps and is less familiar to most labs. For example in PMID: 33589836 (Nature Methods), which describes a multi-modal CUT&Tag/RNA-seq method also applied to mouse brain, 5 different histone modifications are profiled, each targeting a different feature of the epigenome. As there is no data using antibodies that provide information beyond what can be found more simply by the ATAC-seq version of the method, this aspect of the study is too preliminary for publication.

2) ATAC-seq, H3K27ac-CUT&Tag and RNA-seq should in principle be concordant at steady-state, and

as shown in the UMAPs of Figure 3, this is the case for these spatial implementations. Non-concordance might be interpreted as suggesting cause-effect dynamics, if for example there is an ATAC-seq signal without an RNA-seq signal. Evidence for this is shown in Figure 2e (2d and 2e are apparently mixed up in the text and figure legend relative to the figure panels), however the reads are quite sparse and noisy for most of the clusters (and there is no indication of what is the locus shown, enhancer? promoter? of what?) and not convincing. There is very little effort made to address this critical issue, as most of the data shown presents just the joint modalities. Thus it is not clear that there is much added value to adding spatial-RNA-seq to spatial-ATAC-seq or vice-versa. This is in contrast to other multi-modal studies; for example in PMID: 33589836 referred to above, there are UMAPs showing the chromatin-only and chromatin+RNA for direct comparison, where it is clear that the RNA-seq data substantially improved the clustering in general. One of the histone modifications was H3K27ac, and in that case there was little if any obvious improvement in UMAP cluster separation. Since H3K27ac is exactly the mark used here for Spatial-CUT&Tag, which marks the same features as ATAC-seq, there is reason to question whether RNA-seq provides substantial enough additional useful information about the mouse brain than what would be obtained by Spatial-ATAC-seq alone.

3) The authors make the bold claim that “The discordance between chromatin accessibility and gene expression proves the importance of spatial multi-omics co-profiling”. However, these claims seem largely unsupported by the data:

- In Figure 2d, ATAC signal for *Myt1l* and *Nrxn2* is observed at low-to-intermediate levels in places in the E13 mouse embryo where RNA signal is not detected in the same pixels. Whereas the authors' interpretation is that this indicates “lineage priming”, a more parsimonious explanation is that this is background ATAC signal, given that reads in peaks (13%) was relatively low for this sample. There is no indication that lineage-priming is being defined by robust peak-to-gene associations in the manner described by Ma et al. Cell 2020 PMID: 33098772, the article cited for this concept.
- In Figure 2h, a large proportion of the rows presented appear to have high RNA expression that precedes high ATAC signal on the pseudotemporal trajectory (assuming increasing pseudotime moves from left to right). This directly contradicts the associated claim that “the gene expression exhibited a similar temporal tendency as the chromatin accessibility, but showed a slower pace during the developmental process, such as *Sox11* and *Nfix*, in agreement with epigenetic lineage-priming of gene expression”. The examples in Figure 2i should therefore not be considered representative unless additional data is mustered to support this claim.
- The idea of lineage priming manifesting as promoter ATAC-seq preceding RNA-seq expression should be treated as rather extraordinary, since the prior expectation should be that promoter accessibility is dependent upon RNA PolIII residence/activity and not the other way around.

4) For a technology-focused paper, especially one published in a general interest journal, readers will be most interested if the methodology can be practically applied to their own problems of interest. In the case of biocomputing technologies, public availability of software is a requirement for publication at least to my knowledge. Otherwise, there will be only brief a flurry of interest but no lasting impact. This method requires microfluidic devices, and presumably some will become commercially available at some point in the future, but until then it is difficult to get as enthusiastic about this study as about the 3 previous single-modal papers when they first appeared.

Minor problems:

- Figure 2b is difficult to interpret in relation to Figure 2a. It is not clear how the clusters relate to each other here.
- Figure 3c: The “lineage priming” claim should be distinguished from spurious RNA-seq background, especially when only a single example is being used to support the claim.

Referee #2 (Remarks to the Author):

Professor Fan and colleagues have developed a new technology for joint spatial transcriptomic and epigenomic profiling in two versions. The first one, called spatial-ATAC-RNA-seq, is designed for joint profiling of RNA and open chromatin; and the second one, called spatial-CUT&Tag-RNA-seq, is designed for joint profiling of RNA and histone modifications. To my knowledge, this is the first sequencing-based spatial multi-omic technology for joint RNA and open chromatin/histone modification profiling. The data appears to have high quality, although it requires more rigorous evaluation through additional analysis (see detailed comments below). If confirmed, this will provide biologists a powerful tool for investigating the spatial patterns and regulatory mechanisms of cellular heterogeneity within a tissue. To demonstrate its utility, the authors applied this technology to study fetal and postnatal mouse brain as well as adult human brain. They were able to identify distinct cell clusters whose spatial distributions were consistent with tissue histology. By integrative analysis RNA and chromatin accessibility profiles, they identified key enhancers and TFs associated and each cell cluster. Interestingly, they identified a number of highly expressed genes that corresponded to low chromatin accessibility. Such information provides mechanistic insights that are not available from knowledge of individual modalities alone.

Major comments:

1. The raw data quality scores (ATACseq fragments and number of detected genes) are impressive and comparable to non-spatial counterparts. What is the average number of UMIs detected for each pixel? It is unclear to what extent the genome-wide expression and epigenetic profiles agree with previous data from the corresponding cell-types. The authors did try to address this issue by comparing with several public datasets. However, the UMAP co-embedding plots in Figure S4 should be taken with a grain of salt because the algorithm deliberately removes technical variations between platforms. To faithfully represent original data, correlation analysis and Genome Browser visualization will provide more accurate evaluation.
2. There is a non-negligible discrepancy between the RNA and ATACseq based pseudotime analyses results. This is especially evident in the bottom left corner of Figure 2f and g. Is the source of variation biological or technical in nature? If the source is biological, what would be the underlying mechanistic reason?
3. Figure 2h appears to indicate a significant number of genes have different temporal patterns in expression and chromatin profiles. The biological significance of such discrepancy is not discussed. Also, part of the discrepancy may be technical as the underlying trajectories from RNA and ATAC are

different. To remove this confounding factor, it would be more relevant to visualize the dynamic patterns under a common pseudotime coordinate system (either RNA or ATACseq based).

4. The integration analysis of spatial-ATACseq-RNAseq and spatial-CUT&Tag-RNAseq for P21 mouse brain is somewhat confusing. It seems that each dataset is analyzed independently, Consequently, the resulting clustering patterns do not match exactly. Also, the two datasets seem to target different regions in the brain, although they appear to be symmetrical. In any case, this analysis does not seem to provide a more coherent understanding of the mechanism than using each dataset alone.

5. The comparative analysis of RNA and ATAC dynamics for PROX1 is interesting. The authors discovered the discrepancy between RNA and ATAC patterns could be utilized to infer RNA turnover rate. I wonder if this relationship can be formalized as a quantitative model, although it might be beyond the scope of this paper.

Minor comments:

1. Gene activity score analysis is used for spatial-CUT&Tag-RNA-seq data, but I could not find where this score is defined.

2. Several technologies exist for spatial epigenomic profiling, including (sciMAP-ATAC Nat Commun 12, 1274 (2021)), merFISH (bioRxiv, doi: <https://doi.org/10.1101/2022.02.17.480825>), and seqFISH+ (Nature 590, 344–350 (2021)). The latter study also jointly profiled RNA and chromatin state from the same cells. It will be relevant to discuss the innovation and significance of this paper in the aforementioned (and possibly additional) literature context.

Referee #3 (Remarks to the Author):

This manuscript provides a novel and interesting technique, while serving as a biological reference catalogue for epigenetic and transcriptomic signatures in the mouse CNS during two developmental timepoints. Their biotechnology side involves a novel combination of epigenetics and transcriptomics from the same unit area, layered with spatial information, from PFA fixed tissues! The biological importance lies in ability to precede the transcriptomics and observe how epigenetics influences the transcriptomic outcome seen in embryonic versus adult tissues, while also translating to human hippocampus data.

This manuscript is incredibly original from techniques to data.

I would like to see more metrics regarding the technique, such as: during QC, how many barcode combinations are thrown away (as empty or just too poor of quality)? How many cells do you hypothesize to be captured by one barcode? Is there a supplementary video on how everything works that can be added? Are there any metrics for fidelity (which could be compromised just due to diffusion of cell contents) between pixels?

The overlap of the clusters from any given -omics dataset onto specific regions of the tissues is incredible. I would just like to see some scale bars on the images with clusters overlaid.

There is not so much a biological story throughout the manuscript, the data and way it is interpreted

sums out to be more of a reference, where focusing on and validating some gene candidates could provide a more interesting story.

Data analysis is well reported with availability of all scripts on a github page.

In terms of providing credit to other work, there could be more comparison with Visium, however, I am fine if just a reference sentence distinguishing their technique from Visium is thrown in to the introduction.

The abstract, introduction and conclusion are all clear and concise and the manuscript and data is presented well. In the results, I would like to see more interpretation/explanation in discrepancies seen in metrics in Figure 1, specifically the E13 data seems to often behave quite differently, do you think this can be explained by biology or is this a technical artifact?

Altogether, the technique seems great, the analysis seems solid and writing is clear, just more emphasis on some gene candidates could make this manuscript more interesting.

Author Rebuttals to Initial Comments:

We greatly appreciate the reviewers for all the constructive review comments, which contributed to a major improvement of our manuscript. We have fully addressed all the concerns from three reviewers, with an extensive set of experiments and new analysis. We have also now generated webresources in UCSC Cell and Genome Browser (<https://genome-test.gi.ucsc.edu/~max/rong/>) and our own data portal generated with AtlasXplore (<https://web.atlasxomics.com/>, Id: reviewer Password: Reviewer@fanlab1), where the data can be browsed, which will constitute a unique resource for the scientific community. Following please find our point-by-point responses to all review comments. Thank you so much!

Referee #1 (Remarks to the Author):

The authors have extended their powerful DBiT microfluidics strategy for spatial transcriptomics and epigenomics for multi-modal chromatin/transcriptome profiling by adding RNA-seq to both ATAC-seq and CUT&Tag. Since these authors' single-modal spatial methods are already available as publications or preprints (RNA-seq: Cell PMID: 33188776; ATAC-seq: biorxiv 2021.06.06.447244; CUT&Tag: Science PMID: 35143307), I will focus my evaluation on the question of how much the current study adds to those previous studies. Unfortunately, I do not find enough added value to recommend publication in Nature.

Response: We would like to thank the reviewer for the feedback regarding our manuscript!

The latest advances in spatial biology have enabled the profiling of proteins, transcriptome, and recently epigenome, creating a rich set of tools to empower future biological and biomedical research. However, most methods are single modality omics that capture only one layer of the omics information at a time on a tissue sample. It is highly desirable to develop spatial multi-omics techniques, especially, epigenome & transcriptome, for interrogating the mechanisms to control gene expression, cell fate decision and dynamics in the tissue context. This was highlighted as one of the seven technologies to watch in 2022 by Nature (<https://www.nature.com/articles/d41586-022-00163-x>). In the revised version of the manuscript, we demonstrated that the DBiT-Seq platform is unique in its versatility which allows the assessment of different genome-wide omic layers in the same tissue section. There are several key advances including (i) versatility to develop different spatial epigenome-transcriptome profiling protocols, (ii) high resolution at a near single cell level (20 μ m), (iii) new device design with larger mapping area (100x100 pixels) cover nearly an entire mouse brain hemisphere, (iv) up to three histone modifications (H3K27me3, H3K27ac, or H3K4me3, respectively) for spatial-CUT&Tag-RNA-seq, and (v) application to a variety of mammalian tissue types ranging from embryonic to postnatal mouse tissues and to highly complex human brain tissues (hippocampus). Thus, our manuscript is a major leap forward rather than an incremental next step to develop a new platform for spatially resolved genome-wide epigenome-transcriptome co-profiling.

With regard to the biological insights, this work demonstrated extensively the power of joint profiling for epigenome and transcriptome to examine the causal relationship between epigenetic state and transcriptional phenotype as well as to identify cell state and dynamics. Specifically, we (i) detected >20,000 significant peak-to-gene linkages between epigenetic regulatory elements and target genes, (ii) decipher the gene regulation mechanism and spatial dynamics during tissue development, (iii) epigenetic but not transcriptional memory of specific genes during brain development, (iv) dissect the principles of gene and epigenetic regulation that infer cell identity in diverse areas of the brain, uncovering for instance unexpected regional correlation between H3K27me3 and expression. Our results highlight the power of spatial epigenome-transcriptome co-profiling to dissect not only cell types or states but also the mechanisms controlling cell fate selection and dynamics. Thus, our spatial-epigenome-transcriptome co-sequencing dataset itself is a long-awaited and highly valuable resource.

Major comments

1) Although Spatial-ATAC-seq-RNA-seq is described fully, including application to a wider-field microfluidic device, Spatial-CUT&Tag-RNA-seq seems to be last-minute add-on. Spatial-CUT&Tag-RNA-seq data are described in the text and in Figure 1 and interpreted without any indication of the antibody used. Different antibodies for example to different histone modifications report on different parts of epigenome, so the results are meaningless to a reader without knowing what antibody against what epitope was used. Only by Figure 3 is the mystery solved where it is stated that a single primary antibody against H3K27ac was used. This is a promoter/enhancer mark that should give roughly the same profile as ATAC-seq. But users of this method will not be interested in doing spatial-CUT&Tag-RNA-seq to do something that they can do using the spatial-ATAC-seq-RNA-seq, since it requires additional steps and is less familiar to most labs. For example in PMID: 33589836 (Nature Methods), which describes a multi-modal CUT&Tag/RNA-seq method also applied to mouse brain, 5 different histone modifications are profiled, each targeting a different feature of the epigenome. As there is no data using antibodies that provide information beyond what can be found more simply by the ATAC-seq version of the method, this aspect of the study is too preliminary for publication.

Response: We thank the reviewer for the comments and we apologize for the oversight of not mentioning upfront that H3K27ac was the profiled histone modification. In our revised manuscript, we first designed a new device which could improve the number of barcodes from 50 (2,500 pixels, 20 μm pixel size) to 100 (10,000 pixels, 20 μm pixel size). The mapping area is 4 times larger than before (**Fig. 2a**, please also see below). The serpentine channel design in the new device can further realize the processing of up to 5 samples simultaneously.

This new device was used to conduct not only spatially resolved genome-wide joint mapping of epigenome and transcriptome by co-profiling chromatin accessibility and gene expression (spatial-ATAC-RNA-seq) but also now three histone modifications and gene expression (spatial-CUT&Tag-RNA-seq, H3K27ac, or H3K27me3, or H3K4me3 histone modifications, respectively) on the same mouse postnatal day 22 (P22) brain tissue, which can cover nearly the entire mouse brain hemisphere at a resolution near single cells (1~3 cells per pixel) (**Fig. 2b**, **Fig. 3a-c**, please also see below).

For spatial-CUT&Tag-RNA-seq, the application of the technology to H3K27me3 (repressed loci) and H3K4me3 (active promoters) clearly demonstrates the potential of the technology beyond spatial ATAC-RNA-seq. The detailed results of Spatial-CUT&Tag-RNA-seq for coprofiling of CUT&Tag (H3K27me3), CUT&Tag (H3K27ac), and CUT&Tag (H3K4me3) with transcriptome were shown in **Fig. 3** and **Fig. 4** (please also see below) and we did a comprehensive revision of sections “Juvenile mouse brain: spatial co-mapping of chromatin accessibility with transcriptome at the cellular level”, “Juvenile mouse brain: spatial co-mapping of histone modifications (H3K27me3, H3K27ac, and H3K4me3) with transcriptome at the cellular level” and added a new section “Spatial multi-omics deciphers region-specific epigenetic regulation of gene expression and cooperation in mouse brain” in our revised manuscript according to the reviewer’s suggestion.

Fig. 2. a, Design of the microfluidic chips for 100x100 barcodes with 20 μm channel size. **b**, Spatial UMAP and UMAP of all the clusters for ATAC and RNA in spatial-ATAC-RNA-seq for the mouse brain. Pixel size 20 μm . Scale bar, 1 mm.

Fig. 3. Spatial epigenome and transcriptome co-sequencing (Spatial-CUT&Tag-RNA-seq) and integrative analysis of P22 mouse brain. **a-c**, Spatial UMAP and UMAP of all the clusters for CUT&Tag (H3K27me3) and RNA (**a**), CUT&Tag (H3K27ac) and RNA (**b**), and CUT&Tag (H3K4me3) and RNA (**c**) in the mouse brain. Pixel size, 20 μ m. Scale bar, 1 mm. **d**, Integration of CUT&Tag (H3K27me3) data in spatial-CUT&Tag-RNA-seq with scCUT&Tag data from mouse brain. **e**, Integration of CUT&Tag (H3K27ac) data in spatial-CUT&Tag-RNA-seq with scCUT&Tag data from mouse brain. **f,g**, Integration of RNA data in spatial-CUT&Tag(H3K27me3)-RNA-seq, spatial-CUT&Tag(H3K27ac)-RNA-seq, and spatial-CUT&Tag(H3K4me3)-RNA-seq with scRNA-seq from mouse brain.

Fig. 4. Region specific epigenetic regulation of gene expression for spatial epigenome and transcriptome co-sequencing. **a**, Correlation of H3K27me3 CSS and RNA gene expression in corpus callosum. **b**, Correlation of H3K27ac GAS and RNA gene expression in corpus callosum. **c**, Correlation of H3K4me3 GAS and RNA gene expression in corpus callosum. **d**, Upset plot of H3K27me3 CSS and RNA gene expression in striatum, deeper and superficial cortical layer. Low CSS or gene expression (-), High CSS or gene expression (+). **e-g**, Venn diagrams showing the number of high (+) or low (-) CSS/GAS for different histone modifications in corpus callosum for common RNA marker genes. **h-k**, Spatial mapping of CSS, GAS, and gene expression for selected marker genes *Mag* (**h**), *Car2* (**i**), *Syt1* (**j**), *Gpr88* (**k**) in different clusters for ATAC, CUT&Tag and RNA in spatial-ATAC-RNA-seq and spatial-CUT&Tag-RNA-seq.

2) ATAC-seq, H3K27ac-CUT&Tag and RNA-seq should in principle be concordant at steady-state, and as shown in the UMAPs of Figure 3, this is the case for these spatial implementations. Non-concordance might be interpreted as suggesting cause-effect dynamics, if for example there is an ATAC-seq signal without an RNA-seq signal. Evidence for this is shown in Figure 2e (2d and 2e are apparently mixed up in the text and figure legend relative to the figure panels), however the reads are quite sparse and noisy for most of the clusters (and there is no indication of what is the locus shown, enhancer? promoter? of what?) and not convincing. There is very little effort made to address this critical issue, as most of the data shown presents just the joint modalities. Thus it is not clear that there is much added value to adding spatial-RNA-seq to spatial-ATAC-seq or vice-versa. This is in contrast to other multi-modal studies; for example in PMID: 33589836 referred to above, there are UMAPs showing the chromatin-only and chromatin+RNA for direct comparison, where it is clear that the RNA-seq data substantially improved the clustering in general. One of the histone modifications was H3K27ac, and in that case there was little if any obvious improvement in UMAP cluster separation. Since H3K27ac is exactly the mark used here for Spatial-CUT&Tag, which marks the same features as ATAC-seq, there is reason to question whether RNA-seq provides substantial enough additional useful information about the mouse brain than what would be obtained by Spatial-ATAC-seq alone.

Response: We thank the reviewer for the comments.

In order to verify the ATAC and RNA data quality of our technology, we did a comparison of aggregated profiles of chromatin accessibility from E13 mouse embryo with organ-specific ENCODE E13.5 ATAC-seq data. The aggregated profiles can reproduce the ENCODE bulk measurement accurately (**Fig. S3b**, please also see below). The signal-to-noise ratio of our ATAC signal is comparable with the ENCODE data. Furthermore, the peaks obtained from our ATAC data in spatial-ATAC-RNA-seq are consistent with ENCODE ATAC-seq data (**Fig. S3c**, please also see below). For our RNA data in spatial-ATAC-RNA-seq, we did a correlation analysis with organ-specific ENCODE E13.5 bulk RNA-seq data, which showed high reproducibility ($r=0.849$ (forebrain) and $r=0.834$ (hindbrain), r stands for Pearson correlation coefficient) (**Fig. S3a**). All the above results proved the good data quality of our technology. While we observed a strong correlation between replicates, we agree with the reviewer that challenges might occur regarding the depth of RNA detection, which we mention in the revised manuscript.

Fig. S3. Benchmarking of data quality in spatial-ATAC-RNA-seq. **a**, Comparison of transcriptional profiles between RNA in spatial-ATAC-RNA-seq and the ENCODE RNA-Seq data in brain of mouse embryo. **b**, Aggregated spatial chromatin accessibility profiles in spatial-ATAC-RNA-seq recapitulated ENCODE ATAC-seq profiles in brain of mouse embryo. **c**, Venn diagram showing the overlap of peaks between ATAC in spatial-ATAC-RNA-seq and ENCODE ATAC-seq profiles in brain of mouse embryo.

The Genome browser tracks were drawn with ArchR (*Nat Genet* 2021, 53, 403-411, <https://www.archrproject.com>). The open chromatin near the TSS (transcription start site) is where the promoter located, and the putative enhancers can be recognized by the peak-to-gene links. Also, co-accessibility of peaks was also calculated (using addCoAccessibility function in ArchR) to see the potential co-accessible between peaks (enhancers and promoters). For ATAC, the gene activity score (GAS, ArchR, Methods) was calculated to estimate the gene expression from chromatin accessibility based on the overall signal at a given locus. Following is the toy example provided in the ArchR tutorial to explain the default ArchR gene score model.

[Redacted figure below]

The regulatory and mechanistic links between different omics layers can only be deciphered by profiling multiple regulatory layers. Direct relation between chromatin and transcriptional regulation would be hidden if we will only look at a single regulatory layer. Changes of chromatin accessibility preceding gene expression have been observed in lineage priming from single-cell based methods, for instance it has been previously observed in mouse skin cells (Cell, 2020, 183(4): 1103-1116) and mouse brain oligodendroglia (Neuron, 2022, 110(7): 1193-1210). Only by coprofiled different modalities have we been able to identify lineage priming cases as reported by others.

To further address this point and also the reviewer's point number 3, we focused on the pseudotime analysis from radial glia cells to postmitotic premature neurons in the brainstem of the E13 mouse embryo. The pseudotime analysis was conducted under the ATAC coordinate system according to reviewer 2's suggestion, and the developmental trajectories were directly visualized in the spatial tissue map (**Fig. 1j**, please also see below). The gene activity scores for chromatin accessibility and gene expression along this developmental trajectory were computed and the dynamic changes in selected marker genes were presented (**Fig. 1k,l**, please also see below). Overall, the gene expression exhibited a similar temporal tendency as the chromatin accessibility. As expected, the expression levels of *Sox2*, *Pax6* and other genes involved in progenitor maintenance and proliferation (Gene Ontology (GO) *Fabp7* to *Pax6* in **Fig. S6a**, please also see below) were downregulated during the transition to postmitotic neurons. Interestingly the loss of chromatin accessibility at the *Pax6* and the radial glia marker *Fabp7* loci preceded the downregulation at the RNA level (**Fig. 1l**). In turn, genes involved in neuronal identity, axonogenesis and synapse organization (GO *Myt1l* to *Dnm3* in **Fig. S6a**, please also see below) as *Dcx* and *Tubb3* presented increased expression in the spatial pseudotime, but the chromatin accessibility at their loci was already elevated at much earlier stages, suggesting lineage-priming of gene expression (Cell, 2020, 183(4): 1103–1116; Neuron, 2022, 110(7): 1193-1210). We also found a cohort of genes whose expression quickly declined during the spatial pseudotime, but whose chromatin accessibility was maintained throughout the pseudotime, only declining at very late stages. Many of these genes such as *Ptprz1*, *Bcan*, *Luzp2* are characteristic of oligodendrocyte precursor cells, and indeed GO analysis highlighted biological processes as negative regulation of myelination and regulation of gliogenesis (GO Cp to *Sparcl1* in **Fig. S6a**, please also see below) suggesting that the neuronal lineage might retain the potential to acquire an oligodendrocyte identity even when they have already migrated away from the ventricular zone in the embryonic brain. Interestingly, Monocle2 pseudotime analysis indicated a bifurcation at the chromatin accessibility level that was not observed at the RNA level (**Fig. S6b,c**). One bifurcation path led to areas close to the ventricular zone (green pixels, **Fig. S6d,e**), while the other bifurcation path terminated in areas distal from the ventricular zone (blue pixels). In contrast to the green path, the blue path presented an increase in chromatin accessibility in genes involved in axonogenesis and dendrite formation (**Fig. S6e**, red box and **Fig. S6f**), suggesting that the chromatin state of neural cells distal to the ventricles is indeed more consistent with a terminally differentiated neuronal state. Thus, our spatial-ATAC-RNA-seq can be used to decipher the gene regulation mechanism and spatial dynamics during tissue development. The above contents have been added in the revised manuscript.

Fig. 1. **j**, Pseudotime analysis from radial glia to postmitotic premature neurons visualized at a spatial level. **k**, Heatmaps delineating gene expression and gene activity scores for marker genes. **l**, Dynamic changes of gene activity scores and gene expression across pseudotime.

Fig. S6 Further pseudotime analysis of radial glia and postmitotic premature neurons in spatial-ATAC-RNA-seq. a, GO enrichment analysis for genes from **Fig. 1k. b,c,** Pseudotime analysis from radial glia to postmitotic premature neurons with gene activity scores **(b)** and gene expression **(c)**. **d,** Monocle2 analyses showing different states in **(b)**. **e,** Heatmap of different states along the pseudotime trajectory. **f,** GO analysis of genes in red box of **(e)**.

We apologize for not being clear regarding the clustering. We followed the methods from Signac (Methods, https://stuartlab.org/signac/articles/pbmc_multiomic.html) to do unsupervised clustering for ATAC/CUT&Tag and RNA data, separately. First, we did the unsupervised clustering for RNA data (results from RNA clustering were obtained), and unsupervised clustering of ATAC/CUT&Tag data was performed separately or independently from RNA (results from ATAC/CUT&Tag clustering were obtained). Finally, joint UMAP (for E13 mouse embryo and P21 mouse brain) was done using

“FindMultiModalNeighbors” (results from clustering of joint modalities were obtained). In the end, the UMAPs for ATAC (for example, **Fig. 1e**, left, please also see below), RNA (for example, **Fig. 1e**, middle) and RNA+ATAC (for example, **Fig. 1e**, right) were visualized separately using “DimPlot”. We included this description in the Methods section of the revised version of the manuscript.

We identified 8 clusters from ATAC data and 14 clusters from RNA data for spatial-ATAC-RNA-seq of E13 mouse embryo. We provided the UMAP and spatial UMAP for ATAC (**Fig. 1e**, left) and RNA (**Fig. 1e**, middle). The joint clustering of ATAC and RNA as the reviewer mentioned is able to refine the spatial patterns (**Fig. 1e**, right). For example, we identified a new neuronal cluster (J10) in the joint clustering analysis, which was not readily resolved by single modalities alone (**Fig. 1e**, right). This cluster was recognized as granule neurons after label transfer with Mouse Organogenesis Cell Atlas (MOCA) (Nature, 2019, 566(7745): 496-502) scRNA-seq dataset. This result highlights the unique value to use joint multi-omics profiles for improving the cell-type-specific spatial mapping.

We also conducted spatial mapping of P22 mouse brain coronal sections (at Bregma 1) for chromatin accessibility jointly with transcriptome. In contrast to embryonic tissue, we were able to identify a larger number of clusters by ATAC (fourteen) than with RNA data (eleven) with unsupervised clustering, respectively. This might reflect the terminal differentiated status of most cells in the juvenile brain, in contrast the undifferentiated and multipotent state of cells during development. The distinct spatial distributions are also in agreement with anatomical annotation defined by the Nissl staining, reflecting arealization of the juvenile brain (**Fig. 2b**). Moreover, spatial clusters between ATAC and RNA showed strong concordance in cluster assignment (**Fig. S7c**).

According to the reviewer's suggestions, for spatial-CUT&Tag-RNA-seq, we expanded the histone modifications to H3K27me3 (marks repressed loci), H3K27ac (marks active promoter and/or enhancer), and (H3K4me3 is for active promoters). Unsupervised clustering was used separately to each modality, which identified 13 and 15 specific clusters for CUT&Tag (H3K27me3) and RNA, 12 and 13 clusters for CUT&Tag (H3K27ac) and RNA, 11 and 12 clusters for CUT&Tag (H3K4me3), respectively (**Fig. 3a-c**, please also see below). As can be seen from the UMAP and spatial UMAP of H3K27me3 (**Fig. 3a**, top), H3K27ac (**Fig. 3b**, top), and H3K4me3 (**Fig. 3c**, top), for CUT&Tag (H3K27me3, H3K27ac, and H3K4me3), we could get clusters agreed well with the anatomical regions defined by the Nissl staining from an adjacent tissue section without the need of integration with the corresponding RNA data. Also, there is good concordance between CUT&Tag and RNA in terms of spatial patterns (**Fig. 3a-c**).

Integration of our CUT&Tag (H3K27ac or H3K4me3) data with the corresponding scRNA-seq data also allowed for label transfer to assign epigenetic cell identities/states to spatial location (**Fig. S11f,g**). We observed an enrichment of MOL within the corpus callosum, a thin layer of EPEN in the lateral ventricle, excitatory neurons (TEGLU) in cerebral cortex, and MSN in the striatum, which are in consistent with the results obtained from RNA data (**Fig. S11f,g**). In particular, for spatial H3K27me3, while integration with scCUT&Tag could not clearly indicate the identity of several clusters in the epigenomic modalities (cluster 0, cluster 1, and cluster 3), integration of the spatial RNA in the same sections with scRNA-seq clearly indicated the cell identities in these clusters (**Fig. 3a,d** and **Fig. S11e**). This highlights the power of combining CUT&Tag and RNA-seq (spatial-CUT&Tag-RNA-seq) in the same tissue section. Also, proved the good clustering for our CUT&Tag data.

Moreover, to further understand the epigenetic regulation of gene expression, a new section “Spatial multi-omics deciphers region-specific epigenetic regulation of gene expression and cooperation in mouse brain” was added and more genome-wide analyses were done in this section (**Fig. 4**, please also see below, and **Fig. S15-22**) to decipher the mechanism of epigenetic regulation of gene expression in genome-wide.

Fig. 4. Region specific epigenetic regulation of gene expression for spatial epigenome and transcriptome co-sequencing. **a**, Correlation of H3K27me3 CSS and RNA gene expression in corpus callosum. **b**, Correlation of H3K27ac and RNA gene expression in corpus callosum. **c**, Correlation of H3K4me3 GAS and RNA gene expression in corpus callosum. **d**, Upset plot of H3K27me3 CSS and RNA gene expression in striatum, deeper and superficial cortical layer. Low CSS or gene expression (-), High CSS or gene expression (+). **e-g**, Venn diagrams showing the number of high (+) or low (-) CSS/GAS for different histone modifications in corpus callosum for common RNA marker genes. **h-k**, Spatial mapping of CSS, GAS, and gene expression for selected marker genes *Mag* (**h**), *Car2* (**i**), *Syt1* (**j**), *Gpr88* (**k**) in different clusters for ATAC, CUT&Tag and RNA in spatial-ATAC-RNA-seq and spatial-CUT&Tag-RNA-seq.

3) The authors make the bold claim that “The discordance between chromatin accessibility and gene expression proves the importance of spatial multi-omics co-profiling”. However, these claims seem largely unsupported by the data:

- In Figure 2d, ATAC signal for *Myt1l* and *Nrxn2* is observed at low-to-intermediate levels in places in the E13 mouse embryo where RNA signal is not detected in the same pixels. Whereas the authors' interpretation is that this indicates “lineage priming”, a more parsimonious explanation is that this is background ATAC signal, given that reads in peaks (13%) was relatively low for this sample. There is no indication that lineage-priming is being defined by robust peak-to-gene associations in the manner described by Ma et al. Cell 2020 PMID: 33098772, the article cited for this concept.
- In Figure 2h, a large proportion of the rows presented appear to have high RNA expression that precedes high ATAC signal on the pseudotemporal trajectory (assuming increasing pseudotime moves from left to right). This directly contradicts the associated claim that “the gene expression exhibited a similar temporal tendency as the chromatin accessibility, but showed a slower pace during the developmental process, such as *Sox11* and *Nfix*, in agreement with epigenetic lineage-priming of gene expression”. The examples in Figure 2i should therefore not be considered representative unless additional data is mustered to support this claim.

Response: We thank the reviewer for this comment, which we address in point 2, namely in the pseudotime analysis at E13. Nevertheless, we removed the sentence “The discordance between chromatin accessibility and gene expression proves the importance of spatial multi-omics co-profiling” from the corresponding section according to the reviewer’s comment.

As the reviewer mentioned, in a recent published study (Cell, 2020, PMID: 33098772), a method is described to predict the possible lineage priming by calculating DORCs (domains of regulatory chromatin), which are domains that have significantly overlap with super-enhancers. They found at DORCs, that chromatin accessibility precedes gene expression during lineage commitment. Since DORCs could predict the possible lineage priming loci, we calculate DORCs for our E13 spatial-ATAC-RNA-seq data, *Myt1l* belongs to DORCs, it may indicate the lineage priming. We removed *Nrxn2* in this sentence.

C
While the overall spatial RNA pixel distribution of these genes overlapped with their chromatin accessibility, we observed clear differences at the level of expression, with some regions with abundant ATAC signal being devoid of transcription of the corresponding gene (**Fig. 1h**, please also see below, and **Fig. S4b**). We found that some of the marker genes identified from ATAC data were enriched in part of the embryonic brain (for example, *Pax6*, *Sox2*, and *Myt1l*) but not highly expressed in RNA data (**Fig. 1h** and **Fig. S4b**), which may indicate the lineage priming²⁶ of these genes in embryonic brain (**Fig. S4c**)¹⁰. While we observed a strong correlation between replicates (**Fig. S2c**, please also see below), this could be due to technical challenges regarding the depth of RNA detection, which we mention in the revised version of the manuscript. Nevertheless, these differences highlight the potential of the presented technologies to identify differential correlation between epigenomic information and transcription in different regions.

- 10 Ma, S. *et al.* Chromatin Potential Identified by Shared Single-Cell Profiling of RNA and Chromatin. *Cell* **183**, 1103-1116 e1120, doi:10.1016/j.cell.2020.09.056 (2020).
- 26 Meijer, M. *et al.* Epigenomic priming of immune genes implicates oligodendroglia in multiple sclerosis susceptibility. *Neuron* **110**, 1193-1210.e1113, doi:<https://doi.org/10.1016/j.neuron.2021.12.034> (2022).

- The idea of lineage priming manifesting as promoter ATAC-seq preceding RNA-seq expression should be treated as rather extraordinary, since the prior expectation should be that promoter accessibility is dependent upon RNA PolII residence/activity and not the other way around.

Response: We thank the reviewer for the comments. Lineage priming is that the changes in chromatin accessibility may precede gene expression during gene transcription, leading to the non-concordance or asynchronous changes between ATAC and RNA signals. The lineage priming at enhancers and/or at promoters we observe is not unique, since it has also been observed by other studies. For instance, in mouse skin cells (*Cell*, 2020, 183(4): 1103-1116), mouse brain oligodendroglia (*Neuron*, 2022, 110(7): 1193-1210). Therefore, the chromatin accessibility signal can be applied to predict cell lineage states before the activation of the transcriptional programs.

4) For a technology-focused paper, especially one published in a general interest journal, readers will be most interested if the methodology can be practically applied to their own problems of interest. In the case of biocomputing technologies, public availability of software is a requirement for publication at least to my knowledge. Otherwise, there will be only brief a flurry of interest but no lasting impact. This method requires microfluidic devices, and presumably some will become commercially available at some point in the future, but until then it is difficult to get as enthusiastic about this study as about the 3 previous single-modal papers when they first appeared.

Response: We thank the reviewer for the comments. We are aiming to make our technologies available to solve readers' problems of interest, and will actively work on this.

To make our technology more user-friendly, the detailed experimental procedure for our spatial-ATAC-RNA-seq and spatial-CUT&Tag-RNA-seq technologies were provided in the Methods. We will publish a video protocol as well as a step-by-step protocol for our spatial-ATAC-RNA-seq and spatial-CUT&Tag-RNA-seq technologies. Besides, we have published a video protocol for our DBiT-seq technology (*STAR protocols*, 2021, 2(2): 100532), which includes detailed procedures on fabrication of PDMS microfluidic chips, assembly of the microfluidic device, and how the reagents were added. The labs without any microfluidic experiment can set up this platform easily. We also added this reference (Su G, Qin X, Enniful A, et al. Spatial multi-omics sequencing for fixed tissue via DBiT-seq[J]. *STAR protocols*, 2021, 2(2): 100532) in the Methods section. Additionally, a startup company (<https://www.atlasxomics.com/>) has licensed this technology, which will be adopted quickly by many people in the research community.

The computational analysis part has been performed by using published pipelines and packages. The detailed information was provided in the Methods. All the codes are available at Github: https://github.com/di-0579/Spatial_epigenome-transcriptome_co-sequencing.

All the raw and processed data reported in this manuscript are deposited in the Gene Expression Omnibus (GEO) with accession code GSE205055 (reviewer token: alkvqimobtyjxol).

Furthermore, in order to constitute a unique resource for the scientific community, we have generated the webresources where our data can be browsed easily by the readers, one is UCSC Cell and Genome Browser (<https://genome-test.gi.ucsc.edu/~max/rong/>), another is our own data portal generated with AtlasXplore (<https://web.atlasxomics.com>, Id: reviewer Password: Reviewer@fanlab1). It allows for visualizing spatial pattern of genes or motifs, for example, by selecting a gene in the table of top ranked genes/motifs to show on the search bar to display (click the show function to view one by one). For ATAC and CUT&Tag data, the peaks, raw fragments per cluster, and gene model for the corresponding gene of interest can be viewed (click the Peak Viewer icon in the lower left panel). The motifs can be viewed by switching the icon Gene on the top right panel to Motif followed by selecting motif in the table to show on the search bar in a way similar to visualizing spatial gene expression map. The sequence logo can be displayed by clicking the Peak Viewer icon. The detailed tutorial of AtlasXplore for interactive analysis of spatial omics data is available at <https://docs.atlasxomics.com/projects/AtlasXplore/en/latest/>. The uploaded datasets will be available for browsing upon publication, and we are also building the CellXGene webresource, which we aim to release at the same time.

Minor problems:

- Figure 2b is difficult to interpret in relation to Figure 2a. It is not clear how the clusters relate to each other here.

Response: Thank you for the comments. We changed the color of each cluster to make it consistent between Figure 2a (showed as **Fig. 1e** in the revised manuscript) and Figure 2b (showed as **Fig. 1f** in the revised manuscript, please also see below).

- Figure 3c: The “lineage priming” claim should be distinguished from spurious RNA-seq background, especially when only a single example is being used to support the claim.

Response: We thank the reviewer for the comments. According to the reviewer’s comments, we calculated the gene activity scores (GAS) of the same genes (*Sox2*, *Pax6*, *Notch1*, *Sox10*, and *Neurod6*) for spatial-ATAC-RNA-seq with 10,000 and 2,500 pixels device on mouse brain (P22 and P21, respectively). The results were shown in **Fig. 2e**, **Fig. S7a** and **Fig. S10g** (please also see below). As can be seen from the results, from both datasets, while the overall spatial RNA pixel distribution of these genes overlapped with their chromatin accessibility, we could observe clear differences at the level of expression for ATAC and RNA, with some regions with abundant ATAC signal being devoid of

transcription of the corresponding gene. We revised the manuscript for the lineage priming part and the content has already been added in our response to reviewer's third question.

(1) GAS for 10,000 pixels on P22 mouse brain

(2) GAS for 2,500 pixels on P21 mouse brain

Referee #2 (Remarks to the Author):

Professor Fan and colleagues have developed a new technology for joint spatial transcriptomic and epigenomic profiling in two versions. The first one, called spatial-ATAC-RNA-seq, is designed for joint profiling of RNA and open chromatin; and the second one, called spatial-CUT&Tag-RNA-seq, is designed for joint profiling of RNA and histone modifications. To my knowledge, this is the first sequencing-based spatial multi-omic technology for joint RNA and open chromatin/histone modification profiling. The data appears to have high quality, although it requires more rigorous evaluation through additional analysis (see detailed comments below). If confirmed, this will provide biologists a powerful tool for investigating the spatial patterns and regulatory mechanisms of cellular heterogeneity within a tissue. To demonstrate its utility, the authors applied this technology to study fetal and postnatal mouse brain as well as adult human brain. They were able to identify distinct cell clusters whose spatial distributions were consistent with tissue histology. By integrative analysis RNA and chromatin accessibility profiles, they identified key enhancers and TFs associated and each cell cluster. Interestingly, they identified a number of highly expressed genes that corresponded to low chromatin accessibility. Such information provides mechanistic insights that are not available from knowledge of individual modalities alone.

Response: We are grateful to the reviewer for the positive feedback regarding our technologies!

Major comments:

1. The raw data quality scores (ATACseq fragments and number of detected genes) are impressive and comparable to non-spatial counterparts. What is the average number of UMIs detected for each pixel? It is unclear to what extent the genome-wide expression and epigenetic profiles agree with previous data from the corresponding cell-types. The authors did try to address this issue by comparing with several public datasets. However, the UMAP co-embedding plots in Figure S4 should be taken with a grain of salt because the algorithm deliberately removes technical variations between platforms. To faithfully represent original data, correlation analysis and Genome Browser visualization will provide more accurate evaluation.

Response: We thank the reviewer for the comments! For 50 μm pixel size device (50x50 barcodes), the detected UMIs per pixel were found to be an average of 3,603 UMIs (mouse E13) and 2,809 UMIs (human hippocampus) (**Fig. 1c**). For 20 μm pixel size device (P21 mouse brain, 50x50 barcodes), an average of 2,391 UMIs (spatial-ATAC-RNA-seq) and 2,938 UMIs (spatial-CUT&Tag(H3K27ac)-RNA-seq) (**Fig. S2f**) were detected per pixel. To further increase the mapping area, we developed a new device to perform in tissue barcoding of 100x100 pixels with a 20 μm pixel size. We performed spatial-ATAC-RNA-seq and spatial-CUT&Tag-RNA-seq (H3K27me3, H3K27ac or H3K4me3) on P22 mouse brain using the new device, an average of 2,358 UMIs (spatial-ATAC-RNA-seq), 4,734 UMIs (spatial-CUT&Tag(H3K27me3)-RNA-seq), 3,580 UMIs (spatial-CUT&Tag(H3K27ac)-RNA-seq), and 2,885 UMIs (spatial-CUT&Tag(H3K4me3)-RNA-seq) were detected per pixel (**Fig. 1c**). According to the reviewer's suggestion, the average number of UMIs per pixel were added in the manuscript. Also, the metrics for all the samples we processed were summarized in **Table S1** and **Table S2** in the revised manuscript (please also see below).

According to the reviewer's suggestion, we did a validation of the data quality from our technology using our E13 mouse embryo data of spatial-ATAC-RNA-seq. To evaluate the ATAC data quality, we did a comparison of aggregated profiles of chromatin accessibility with organ-specific ENCODE ATAC-seq E13.5 mouse embryos data. The aggregated profiles can reproduce the ENCODE bulk measurement accurately (**Fig. S3b**). Additionally, the peaks obtained from our ATAC data in spatial-ATAC-RNA-seq are consistent with ENCODE ATAC-seq data (**Fig. S3c**). To verify the data quality of our RNA data in spatial-ATAC-RNA-seq from E13 mouse embryo, we did a correlation analysis with organ-specific ENCODE E13.5 bulk RNA-seq data, which showed high reproducibility ($r = 0.849$ (forebrain) and $r = 0.834$ (hindbrain), r stands for Pearson correlation coefficient) (**Fig. S3a**).

All the above results proved the good data quality of our technology. We added "To evaluate the ATAC data quality, we did a comparison of aggregated profiles of chromatin accessibility with organ-specific ENCODE E13.5 ATAC-seq data. The aggregated profiles can reproduce the ENCODE bulk measurement accurately (**Fig. S3b**). Additionally, the peaks obtained from our ATAC data in spatial-

ATAC-RNA-seq are consistent with ENCODE ATAC-seq data (Fig. S3c).”, “To verify the data quality of our RNA data in spatial-ATAC-RNA-seq, we did a correlation analysis with organ-specific ENCODE E13.5 bulk RNA-seq data, which showed high reproducibility (Fig. S3a).” and Fig. S3a,b,c in our revised manuscript.

Fig. S3. Benchmarking of data quality in spatial-ATAC-RNA-seq. **a**, Comparison of transcriptional profiles between RNA in spatial-ATAC-RNA-seq and the ENCODE RNA-Seq data in brain of mouse embryo. **b**, Aggregated spatial chromatin accessibility profiles in spatial-ATAC-RNA-seq recapitulated ENCODE ATAC-seq profiles in brain of mouse embryo. **c**, Venn diagram showing the overlap of peaks between ATAC in spatial-ATAC-RNA-seq and ENCODE ATAC-seq profiles in brain of mouse embryo.

Table S1 Summary of metrics for ATAC and RNA in spatial-ATAC-RNA-seq for all the samples.

Spatial-ATAC		E13 mouse embryo	P22 mouse brain	P21 mouse brain	P21 mouse brain	Human brain (50 barcode)

-RNA-seq			(50 barcodes, 50 μ m pixel size)	(100 barcodes, 20 μ m pixel size)	(replica 1, 50 barcodes, 20 μ m pixel size)	(replica 2, 50 barcodes, 20 μ m pixel size)	s, 50 μ m pixel size)
	ATAC	Number of Unique fragments	18,079	14,284	10,857	14,385	9,898
		TSS fragments	16%	19%	20%	19%	15%
		FRiP	11%	26%	24%	26%	11%
		Mitochondrial fragments	0.96%	4.6%	9%	8.4%	20%
	RNA	Number of genes per pixel	1,255	1,073	1,005	1,600	1,200
		Number of UMIs per pixel	3,603	2,358	2,391	3,811	2,809
		Number of unique genes present	20,900	22,914	19,859	20,046	29,293
		Pixels on tissue	2,187	9,215	2,373	2,498	2,500

Table S2 Summary of metrics for CUT&Tag and RNA in spatial-CUT&Tag-RNA-seq for all the samples.

Spatial-CUT&Tag-RNA-seq			P22 mouse brain (H3K27me3) (100 barcodes, 20 μ m pixel size)	P22 mouse brain (H3K27ac) (100 barcodes, 20 μ m pixel size)	P22 mouse brain (H3K4me3) (100 barcodes, 20 μ m pixel size)	P21 mouse brain (H3K27ac, replica 1) (50 barcodes, 20 μ m pixel size)	P21 mouse brain (H3K27ac, replica 2) (50 barcodes, 20 μ m pixel size)
	CUT&Tag	Number of Unique fragments	10,644	10,002	2,507	4,756	5,022

		TSS fragments	12%	17%	67%	19%	20%
		FRiP	12%	21%	54%	19%	18%
		Mitochondrial fragments	0.2%	0.3%	3.6%	0.1%	0.02%
	RNA	Number of genes per pixel	2,011	1,513	1,329	1,145	752
		Number of UMIs per pixel	4,734	3,580	2,885	2,938	1,890
		Number of unique genes present	25,881	23,415	22,731	19,831	18,718
		Pixels on tissue	9,752	9,370	9,548	2,387	2,499

2. There is a non-negligible discrepancy between the RNA and ATACseq based pseudotime analyses results. This is especially evident in the bottom left corner of Figure 2f and g. Is the source of variation biological or technical in nature? If the source is biological, what would be the underlying mechanistic reason?

Response: We thank the reviewer for the comments! According to the reviewer's suggestion, we have re-done the pseudotime analysis under the ATAC pseudotime coordinate system (**Fig. j,k,l**). Here, we focused on brainstem of E13 mouse embryo and chose the differentiation trajectory from radial glia cells to postmitotic premature neurons. In our previous manuscript, we used the cell identities obtained after label transfer from scRNA-seq (Nature, 2019, 566(7745): 496-502) data without taking the regions into consideration, which may cause discrepancy between ATAC and RNA data. To further address the reviewer's comments, we provided the detailed explanation of the revised pseudotime results in our response to reviewer's third question.

3. Figure 2h appears to indicate a significant number of genes have different temporal patterns in expression and chromatin profiles. The biological significance of such discrepancy is not discussed. Also, part of the discrepancy may be technical as the underlying trajectories from RNA and ATAC are different. To remove this confounding factor, it would be more relevant to visualize the dynamic patterns under a common pseudotime coordinate system (either RNA or ATACseq based).

Response: We thank the reviewer for the comments! According to the reviewer's suggestion, we have re-done the pseudotime analysis under the ATAC pseudotime coordinate system. We chose the differentiation trajectory from radial glia cells to postmitotic premature neurons (**Fig. j,k,l**, please also see below).

The developmental trajectories were directly visualized in the spatial tissue map (**Fig. 1j**). The gene activity scores for chromatin accessibility and gene expression along this developmental trajectory were computed and the dynamic changes in selected marker genes were presented (**Fig. 1k,l**). Overall, the gene expression exhibited a similar temporal tendency as the chromatin accessibility. As expected, the expression levels of *Sox2*, *Pax6* and other genes involved in progenitor maintenance and proliferation (Gene Ontology (GO) *Fabp7* to *Pax6* in **Fig. S6a**) were downregulated during the transition to

postmitotic neurons. Interestingly the loss of chromatin accessibility at the *Pax6* and the radial glia marker *Fabp7* loci preceded the downregulation at the RNA level (**Fig. 1l**). In turn, genes involved in neuronal identity, axonogenesis and synapse organization (GO *Myt1l* to *Dnm3* in **Fig. S6a**) as *Dcx* and *Tubb3* presented increased expression in the spatial pseudotime, but the chromatin accessibility at their loci was already elevated at much earlier stages, suggesting lineage-priming of gene expression^{10,26}. We also found a cohort of genes whose expression quickly declined during the spatial pseudotime, but whose chromatin accessibility was maintained throughout the pseudotime, only declining at very late stages. Many of these epigenetic genes such as *Ptprz1*, *Bcan*, *Luzp2* are characteristic of oligodendrocyte precursor cells, and indeed GO analysis highlighted biological processes as negative regulation of myelination and regulation of gliogenesis (GO *Cp* to *Sparcl1* in **Fig. S6a**) suggesting that the neuronal lineage might retain the potential to acquire an oligodendrocyte identity even when they have already migrated away from the ventricular zone in the embryonic brain. Interestingly, Monocle2 pseudotime analysis indicated a bifurcation at the chromatin accessibility level that was not observed at the RNA level (**Fig. S6b,c**). One bifurcation path led to areas close to the ventricular zone (green pixels, **Fig. S6d,e**), while the other bifurcation path terminated in areas distal from the ventricular zone (blue pixels). In contrast to the green path, the blue path presented an increase in chromatin accessibility in genes involved in axonogenesis and dendrite formation (**Fig. S6e**, red box and **Fig. S6f**), suggesting that the chromatin state of neural cells distal to the ventricles is indeed more consistent with a terminally differentiated neuronal state. Thus, our spatial-ATAC-RNA-seq can be used to decipher the gene regulation mechanism and spatial dynamics during tissue development.

The above explanation has been added in the Mouse embryo: spatial co-mapping of chromatin accessibility and transcriptome to identify tissue feature, cell state and developmental dynamics section according to the reviewer's comment.

Fig. 1. **j**, Pseudotime analysis from radial glia to postmitotic premature neurons visualized at a spatial level. **k**, Heatmaps delineating gene expression and gene activity scores for marker genes. **l**, Dynamic changes of gene activity scores and gene expression across pseudotime.

Fig. S6 Further pseudotime analysis of radial glia and postmitotic premature neurons in spatial-ATAC-RNA-seq. a, GO enrichment analysis for genes from **Fig. 1k. b,c,** Pseudotime analysis from radial glia to postmitotic premature neurons with gene activity scores (**b**) and gene expression (**c**). **d,** Monocle2 analyses showing different states in (**b**). **e,** Heatmap of different states along the pseudotime trajectory. **f,** GO analysis of genes in red box of (**e**).

10 Ma, S. *et al.* Chromatin Potential Identified by Shared Single-Cell Profiling of RNA and Chromatin. *Cell* **183**, 1103-1116 e1120, doi:10.1016/j.cell.2020.09.056 (2020).

26 Meijer, M. *et al.* Epigenomic priming of immune genes implicates oligodendroglia in multiple sclerosis susceptibility. *Neuron* **110**, 1193-1210.e1113, doi:https://doi.org/10.1016/j.neuron.2021.12.034 (2022).

4. The integration analysis of spatial-ATACseq-RNAseq and spatial-CUT&Tag-RNAseq for P21 mouse

brain is somewhat confusing. It seems that each dataset is analyzed independently, Consequently, the resulting clustering patterns do not match exactly. Also, the two datasets seem to target different regions in the brain, although they appear to be symmetrical. In any case, this analysis does not seem to provide a more coherent understanding of the mechanism than using each dataset alone.

Response: Thanks for pointing out this confusion. For the clustering of spatial-ATAC-RNA-seq on P21 mouse brain data (2,500 pixels) (Figure 3a, now is shown as **Fig. S10a-c** in our revised manuscript, please also see below), we followed the methods from Signac (Methods, https://stuartlab.org/signac/articles/pbmc_multiomic.html). First, we did the clustering for RNA data (results from RNA clustering were obtained), Then, the ATAC clustering was done separately or independently from RNA (results from ATAC clustering were obtained). Finally, joint UMAP was done using “FindMultiModalNeighbors” (results from clustering of joint modalities were obtained). In the end, the UMAPs for ATAC (**Fig. S10a**), RNA (**Fig. S10b**), and RNA+ATAC (**Fig. S10c**) were visualized separately using “DimPlot”. We included this description in the Methods section of the revised version of the manuscript.

Fig. S10 a-c, Spatial UMAP and UMAP of all the clusters for ATAC (a), RNA (b), and joint clustering of ATAC and RNA (c) in spatial-ATAC-RNA-seq for the mouse brain. Pixel size, 20 µm. Scale bar, 1 mm.

We did the clustering using the same method for spatial-CUT&Tag-RNA (H3K27ac)-RNA-seq data on P21 mouse brain (2,500 pixels), the results are shown in **Fig. S14a-c** (Figure 3b before revision, please also see below). We could see a slight increase of spatial resolution after joint clustering of ATAC+RNA (**Fig. S10c**) and CUT&Tag+RNA (**Fig. S14c**).

Fig. S14 a-c, Spatial UMAP and UMAP of all the clusters for CUT&Tag (H3K27ac) (a), RNA (b), and joint clustering of CUT&Tag and RNA (c) in spatial-CUT&Tag-RNA-seq for the mouse brain. Pixel size, 20 µm. Scale bar, 1 mm.

Also, according to the reviewer’s comment, we did an integration of ATAC and CUT&Tag (H3K27ac) data, the results are shown in the following figures, it could be seen that there is a slight increase of the spatial resolution for CUT&Tag (H3K27ac) in the ventricular zone after integration but no further

improvement for other regions which already matched well with the corresponding anatomical annotations before integration.

For our new generated spatial-ATAC-RNA-seq and spatial-CUT&Tag-RNA-seq (H3K27me3, H3K27ac, or H3K4me3 histone modifications) data with larger mapping area on P22 mouse brain (10,000 pixels, 20 μm pixel size), we could also get good clustering from each modality (ATAC/CUT&Tag and the corresponding RNA) alone (Fig. 2a,b and Fig. 3a-c, please also see below). The distinct spatial distributions for all the datasets are in agreement with anatomical annotation defined by the Nissl staining, reflecting arealization of the juvenile brain (Fig. 2b, Fig. 3a-c). So we don't do joint clustering for ATAC+RNA, and CUT&Tag+RNA for our P22 mouse brain (10,000 pixels) data.

Fig. 2. a, Design of the microfluidic chips for 100x100 barcodes with 20 μm channel size. **b**, Spatial UMAP and UMAP of all the clusters for ATAC and RNA in spatial-ATAC-RNA-seq for the mouse brain. Pixel size, 20 μm . Scale bar, 1 mm.

Fig. 3. a-c, Spatial UMAP and UMAP of all the clusters for CUT&Tag (H3K27me3) and RNA (a), CUT&Tag (H3K27ac) and RNA (b), and CUT&Tag (H3K4me3) and RNA (c) in the mouse brain. Pixel size, 20 μ m. Scale bar, 1 mm.

Also, we did the integration of our ATAC/CUT&Tag and RNA data with the corresponding single cell data for validation or label transfer. In the revised manuscript, we projected the integrated datasets together as gray and black dots to make the results clearer for both P22 (10,000 pixels) (**Fig. 2c,d, Fig. 3d,e,f,g,** and **Fig. S11a-d,** please also see below) and P21 (2,500 pixels) (**Fig. S10e,f** and **Fig. S14e,f,** please also see below) mouse brain data. As can be seen from the results, our data from each modality matched well with the corresponding single cell data.

Fig. 2. c, Integration of ATAC data in spatial-ATAC-RNA-seq with scATAC-seq data from mouse brain (10,000 pixels). **d,** Integration of RNA data in spatial-ATAC-RNA-seq with scRNA-seq from mouse brain (10,000 pixels).

Fig. 3. d, Integration of CUT&Tag (H3K27me3) data in spatial-CUT&Tag-RNA-seq with scCUT&Tag data (10,000 pixels). **e,** Integration of CUT&Tag (H3K27ac) data in spatial-CUT&Tag-RNA-seq with scCUT&Tag data from mouse brain (10,000 pixels).

f,g, Integration of RNA data in spatial-CUT&Tag(H3K27me3)-RNA-seq, spatial-CUT&Tag(H3K27ac)-RNA-seq, and spatial-CUT&Tag(H3K4me3)-RNA-seq with scRNA-seq from mouse brain (10,000 pixels).

Fig. S11 a, Integration of RNA data in spatial-CUT&Tag(H3K27me3)-RNA-seq with scRNA-seq from P22 mouse brain (10,000 pixels). **b**, Integration of RNA data in data in spatial-CUT&Tag(H3K27ac)-RNA-seq with scRNA-seq data from mouse brain (10,000 pixels). **c**, Integration of CUT&Tag (H3K4me3) data in spatial-CUT&Tag-RNA-seq with scCUT&Tag data from mouse brain (10,000 pixels). **d**, Integration of RNA data in spatial-CUT&Tag(H3K4me3)-RNA-seq with scRNA-seq from mouse brain (10,000 pixels).

Fig. S10 e, Integration of ATAC data in spatial-ATAC-RNA-seq with scATAC-seq data from mouse brain (2,500 pixels). **f**, Integration of RNA data in spatial-ATAC-RNA-seq with scRNA-seq from mouse brain (2,500 pixels).

Fig. S14 e, Integration of CUT&Tag data in spatial-CUT&Tag-RNA-seq with scCUT&Tag data from mouse brain (2,500 pixels). **f**, Integration of RNA data in spatial-CUT&Tag-RNA-seq with scRNA-seq from mouse brain (2,500 pixels).

Furthermore, the regulatory and mechanistic links between different omics layers can only be deciphered by profiling multiple regulatory layers. Direct relation between chromatin and transcriptional regulation would be hidden if we will only look at a single regulatory layer. The pseudotime analysis for E13 mouse embryo in our response to reviewer's former comments proved that spatial epigenome-transcriptome co-mapping can be used to decipher the gene regulation mechanism and spatial dynamics during tissue development.

Moreover, to further understand the epigenetic regulation of gene expression, a new section "Spatial multi-omics deciphers region-specific epigenetic regulation of gene expression and cooperation in mouse brain" was added and more genome-wide analyses were added in this section (**Fig. 4**, please also see below, and **Fig. S15-22**) to decipher the mechanism of epigenetic regulation of gene expression genome-wide.

Fig. 4. Region specific epigenetic regulation of gene expression for spatial epigenome and transcriptome co-sequencing. **a**, Correlation of H3K27me3 CSS and RNA gene expression in corpus callosum. **b**, Correlation of H3K27ac GAS and RNA gene expression in corpus callosum. **c**, Correlation of H3K4me3 GAS and RNA gene expression in corpus callosum. **d**, Upset plot of H3K27me3 CSS and RNA gene expression in striatum, deeper and superficial cortical layer. Low CSS or gene expression (-), High CSS or gene expression (+). **e-g**, Venn diagrams showing the number of high (+) or low (-) CSS/GAS for different histone modifications in corpus callosum for common RNA marker genes. **h-k**, Spatial mapping of CSS, GAS, and gene expression for selected marker genes *Mag* (**h**), *Car2* (**i**), *Syt1* (**j**), *Gpr88* (**k**) in different clusters for ATAC, CUT&Tag and RNA in spatial-ATAC-RNA-seq and spatial-CUT&Tag-RNA-seq.

[Redacted figure below]

5. The comparative analysis of RNA and ATAC dynamics for PROX1 is interesting. The authors discovered the discrepancy between RNA and ATAC patterns could be utilized to infer RNA turnover rate. I wonder if this relationship can be formalized as a quantitative model, although it might be beyond the scope of this paper.

Response: We thank the reviewer for the comments. Indeed, development of a quantitative model for this discrepancy will be very interesting. However, the quantitative model is beyond the scope of this manuscript as the reviewer kindly mentioned, since our focus here is the developing of new technologies for spatial co-profiling of epigenome and transcriptome. We will actively work on this according to the reviewer's comments.

Minor comments:

1. Gene activity score analysis is used for spatial-CUT&Tag-RNA-seq data, but I could not find where this score is defined.

Response: We thank the reviewer for the comments. According to the reviewer's suggestion, we made a definition of gene scores for CUT&Tag in spatial-CUT&Tag-RNA-seq and ATAC in spatial-ATAC-RNA-seq. For H3K27me3, chromatin silencing score (CSS) was calculated for the prediction of gene expression. The high CSS represents repressed genes because of the transcriptional repression function of H3K27me3. For ATAC, H3K27ac and H3K4me3, gene activity scores (GAS) were calculated to show cluster specific active genes.

Both CSS and GAS were calculated with ArchR (*Nat Genet* 2021, 53, 403-411, <https://www.archrproject.com/bookdown/calculating-gene-scores-in-archr.html>). Please find below the toy example provided in the ArchR tutorial to explain the default ArchR gene score model. The method of calculation was added in the Methods section of our revised manuscript.

2. Several technologies exist for spatial epigenomic profiling, including (sciMAP-ATAC *Nat Commun* 12, 1274 (2021)), merFISH (bioRxiv, doi:<https://doi.org/10.1101/2022.02.17.480825>), and seqFISH+ (*Nature* 590, 344–350 (2021)). The latter study also jointly profiled RNA and chromatin state from the same cells. It will be relevant to discuss the innovation and significance of this paper in the aforementioned (and possibly additional) literature context.

Response: We thank the reviewer for the suggestion. According to the reviewer's kind suggestion, we already added "Spatial omics technologies (spatial epigenomics, transcriptomics, and proteomics) based either on next generation sequencing (NGS)^{5-8,43-45} or imaging^{9,46}, offer great opportunity to get insight into the gene regulation at a spatial level. However, the comprehensive understanding of the mechanism of gene regulation will need different layers of information simultaneously, which has been realized by single-cell multi-omics technologies¹⁻⁴. Imaging-based DNA seqFISH+ combined with RNA seqFISH also provide the spatial chromatin and gene expression image of cells instead of tissue⁴⁷. However, there is no current technology that can achieve genome-wide co-mapping of different epigenome modalities and transcriptome on the same tissue section at cellular level." and the corresponding references in the Discussion section.

- 1 Allaway, K. C. *et al.* Genetic and epigenetic coordination of cortical interneuron development. *Nature* **597**, 693-697, doi:10.1038/s41586-021-03933-1 (2021).
- 2 Chen, S., Lake, B. B. & Zhang, K. High-throughput sequencing of the transcriptome and chromatin accessibility in the same cell. *Nat Biotechnol* **37**, 1452-1457, doi:10.1038/s41587-019-0290-0 (2019).
- 3 Cao, J. *et al.* Joint profiling of chromatin accessibility and gene expression in thousands of single cells. *Science* **361**, 1380-1385, doi:10.1126/science.aau0730 (2018).
- 4 Trevino, A. E. *et al.* Chromatin and gene-regulatory dynamics of the developing human cerebral cortex at single-cell resolution. *Cell* **184**, 5053-5069 e5023, doi:10.1016/j.cell.2021.07.039 (2021).
- 5 Liu, Y. *et al.* High-Spatial-Resolution Multi-Omics Sequencing via Deterministic Barcoding in Tissue. *Cell* **183**, 1665-1681 e1618, doi:10.1016/j.cell.2020.10.026 (2020).
- 6 Deng, Y. *et al.* Spatial-CUT&Tag: Spatially resolved chromatin modification profiling at the cellular level. *Science* **375**, 681-686, doi:10.1126/science.abg7216 (2022).
- 7 Deng, Y. *et al.* Spatial profiling of chromatin accessibility in mouse and human tissues. *Nature* **609**, 375-383, doi:10.1038/s41586-022-05094-1 (2022).
- 8 Chen, A. *et al.* Spatiotemporal transcriptomic atlas of mouse organogenesis using DNA nanoball-patterned arrays. *Cell* **185**, 1777-1792.e1721, doi:<https://doi.org/10.1016/j.cell.2022.04.003> (2022).
- 9 Lu, T., Ang, C. E. & Zhuang, X. Spatially resolved epigenomic profiling of single cells in complex tissues. *Cell*, doi:<https://doi.org/10.1016/j.cell.2022.09.035> (2022).
- 43 Thornton, C. A. *et al.* Spatially mapped single-cell chromatin accessibility. *Nature Communications* **12**, 1274, doi:10.1038/s41467-021-21515-7 (2021).
- 44 Cho, C.-S. *et al.* Microscopic examination of spatial transcriptome using Seq-Scope. *Cell* **184**, 3559-3572.e3522, doi:<https://doi.org/10.1016/j.cell.2021.05.010> (2021).
- 45 Fu, X. *et al.* Continuous Polony Gels for Tissue Mapping with High Resolution and RNA Capture Efficiency. *bioRxiv*, 2021.2003.2017.435795, doi:10.1101/2021.03.17.435795 (2021).
- 46 Fang, R. *et al.* Conservation and divergence of cortical cell organization in human and mouse revealed by MERFISH. *Science* **377**, 56-62, doi:10.1126/science.abm1741 (2022).
- 47 Takei, Y. *et al.* Integrated spatial genomics reveals global architecture of single nuclei. *Nature* **590**, 344-350, doi:10.1038/s41586-020-03126-2 (2021).

[Redacted figure below]

Referee #3 (Remarks to the Author):

This manuscript provides a novel and interesting technique, while serving as a biological reference catalogue for epigenetic and transcriptomic signatures in the mouse CNS during two developmental timepoints. Their biotechnology side involves a novel combination of epigenetics and transcriptomics from the same unit area, layered with spatial information, from PFA fixed tissues! The biological importance lies in ability to precede the transcriptomics and observe how epigenetics influences the transcriptomic outcome seen in embryonic versus adult tissues, while also translating to human hippocampus data. This manuscript is incredibly original from techniques to data.

Response: We really appreciate the reviewer for the positive feedback regarding our work and manuscript!

- I would like to see more metrics regarding the technique, such as: during QC, how many barcode combinations are thrown away (as empty or just too poor of quality)? How many cells do you hypothesize to be captured by one barcode? Is there a supplementary video on how everything works that can be added? Are there any metrics for fidelity (which could be compromised just due to diffusion of cell contents) between pixels?

Response: We thank the reviewer for the comments!

We first identified pixels on tissue from the bright field image taken from the same tissue using MATLAB, the barcodes didn't on the tissues were thrown away. The detailed processing procedure to select the pixels on tissue was put in Github: (https://github.com/ediciuyang/Hiplex_proteome). The Github link were added in the Methods section according to the reviewer's comments.

The cell numbers captured by one barcode are various, which are decided by pixel size and tissue types. According to the reviewer's comment, we added "The cell numbers captured by the barcode depended on the pixel size and tissue type." in the data quality section of our manuscript.

For 50 μm pixel size, we previously did a counting of cell numbers for E10 mouse embryo, there are about 15~30 cells per pixel with an average cell number of 25.1 (Cell, 2020, 183(6): 1665-1681). We added "For 50 μm pixel size, we obtained an average of 25 cells for E10 mouse embryo" in data quality section of our revised manuscript.

We also used 50 μm pixel size for the human brain hippocampus sample, according to the Nissl staining of an adjacent tissue (we draw a grid with a pixel size of 50 μm , scale bar, 1 mm), each pixel contains 1~9 cells since the cells in tissue are various in size and density. We added "1~9 cells for human hippocampus" in the data quality section of our revised manuscript.

[Redacted figure below]

In order to increase the resolution, we used 20 μm pixel size chip for all the mouse brain samples. According to the Nissl staining of the P21 mouse brain sample (we draw a grid with a pixel size of 20 μm , scale bar, 200 μm , this image was added as **Fig. S2b** in the revised manuscript), most of the pixels contain 1~3 cells in the region of interesting we covered, except for the ventricular layer that contains 5~9 cells in some pixels. We added “most of the pixels contain 1~3 cells per 20 μm pixel size” in data quality section of our revised manuscript.

A video protocol is a great suggestion! We will publish a video protocol as well as a step-by-step protocol for our spatial-ATAC-RNA-seq and spatial-CUT&Tag-RNA-seq technologies.

Besides, we have published a video protocol for our DBiT-seq technology (STAR protocols, 2021, 2(2): 100532.), which includes detailed procedures on fabrication of PDMS microfluidic chips, assembly of the microfluidic device, and how the reagents were added. The labs without any microfluidic experiment can set up this platform easily. We also added this reference “Su G, Qin X, Enniful A, et al. Spatial multi-omics sequencing for fixed tissue via DBiT-seq[J]. STAR protocols, 2021, 2(2): 100532.” in the Methods section.

The possibility of DNA diffusion has been evaluated by analyzing a 3D fluorescence confocal image in our previous study (Cell, 2020, 183(6): 1665-1681), which validated negligible leakage signal

[Redacted figure below]

throughout the tissue section thickness (please see the following figure). It was found to be $4.5 \pm 1 \mu\text{m}$ for $50 \mu\text{m}$ channels and $0.9 \pm 0.2 \mu\text{m}$ for $10 \mu\text{m}$ channels, which confirmed spatially confined delivery and binding of DNA barcodes in tissue with our microfluidic device.

- The overlap of the clusters from any given -omics dataset onto specific regions of the tissues is incredible. I would just like to see some scale bars on the images with clusters overlaid.

Response: We appreciate the reviewer for the comments! We added scale bars on images of tissues for **Fig. 1e**, **Fig. 2b,c**, **Fig. 3a-c**, **Fig. 5a**, **Fig. S10a-c**, and **Fig. S14a-c**.

- There is not so much a biological story throughout the manuscript, the data and way it is interpreted sums out to be more of a reference, where focusing on and validating some gene candidates could provide a more interesting story.

Response: We thank the reviewer for the comments! According to the reviewer's comments, we discussed more genes and focused more on the biological story in the revised manuscript.

According to the reviewer's suggestion, we have focus on the pseudotime analysis of the E13 dataset and we chose the differentiation trajectory from radial glia cells to postmitotic premature neurons (**Fig. j,k,l**, please also see below). The developmental trajectories were directly visualized in the spatial tissue map (**Fig. 1j**). The gene activity scores for chromatin accessibility and gene expression along this developmental trajectory were computed and the dynamic changes in selected marker genes were presented (**Fig. 1k,l**). Overall, the gene expression exhibited a similar temporal tendency as the chromatin accessibility. As expected, the expression levels of *Sox2*, *Pax6* and other genes involved in progenitor maintenance proliferation (Gene Ontology (GO) *Fabp7* to *Pax6* in **Fig. S6a**) were downregulated during the transition to postmitotic neurons. Interestingly the loss of chromatin accessibility at the *Pax6* and the radial glia marker *Fabp7* loci preceded the downregulation at the RNA level (**Fig. 1l**). In turn, genes involved in neuronal identity, axonogenesis and synapse organization (GO *Myt1l* to *Dnm3* in **Fig. S6a**) as *Dcx* and *Tubb3* presented increased expression in the spatial pseudotime, but the chromatin accessibility at their loci was already elevated at much earlier stages, suggesting lineage-priming of gene expression^{10,26}. We also found a cohort of genes whose expression quickly declined during the spatial pseudotime, but whose chromatin accessibility was maintained throughout the pseudotime, only declining at very late stages. Many of these genes such as *Ptprz1*, *Bcan*, *Luzp2* are characteristic of oligodendrocyte precursor cells, and indeed GO analysis highlighted biological processes as negative regulation of myelination and regulation of gliogenesis (GO *Cp* to *Sparcl1* in **Fig. S6a**) suggesting that the neuronal lineage might retain the potential to acquire an oligodendrocyte identity even when they have already migrated away from the ventricular zone in the embryonic brain. Interestingly, Monocle2 pseudotime analysis indicated a bifurcation at the chromatin accessibility level that was not observed at the RNA level (**Fig. S6b,c**). One bifurcation path led to areas close to the ventricular zone (green pixels, **Fig. S6d,e**), while the other bifurcation path terminated in areas distal from the ventricular zone (blue pixels). In contrast to the green path, the blue path presented an increase in chromatin accessibility in genes involved in axonogenesis and dendrite formation (**Fig. S6e**, red box and **Fig. S6f**), suggesting that the chromatin state of neural cells distal to the ventricles is indeed more consistent with a terminally differentiated neuronal state. Thus, our spatial-ATAC-RNA-seq can be used to decipher the gene regulation mechanism and spatial dynamics during tissue development.

The above explanation has been added in the Mouse embryo: spatial co-mapping of chromatin accessibility and transcriptome to identify tissue feature, cell state and developmental dynamics section according to the reviewer's comment.

Fig. 1. **j**, Pseudotime analysis from radial glia to postmitotic premature neurons visualized at a spatial level. **k**, Heatmaps delineating gene expression and gene activity scores for marker genes. **l**, Dynamic changes of gene activity scores and gene expression across pseudotime.

Fig. S6 Further pseudotime analysis of radial glia and postmitotic premature neurons in spatial-ATAC-RNA-seq. a, GO enrichment analysis for genes from **Fig. 1k. b,c,** Pseudotime analysis from radial glia to postmitotic premature neurons with gene activity scores (**b**) and gene expression (**c**). **d,** Monocle2 analyses showing different states in (**b**). **e,** Heatmap of different states along the pseudotime trajectory. **f,** GO analysis of genes in red box of (**e**).

10 Ma, S. *et al.* Chromatin Potential Identified by Shared Single-Cell Profiling of RNA and Chromatin. *Cell* **183**, 1103-1116 e1120, doi:10.1016/j.cell.2020.09.056 (2020).

26 Meijer, M. *et al.* Epigenomic priming of immune genes implicates oligodendroglia in multiple sclerosis susceptibility. *Neuron* **110**, 1193-1210.e1113, doi:https://doi.org/10.1016/j.neuron.2021.12.034 (2022).

We also added a new section “Spatial multi-omics deciphers region-specific epigenetic regulation of gene expression and cooperation in mouse brain” and more genome-wide analyses were done in this section (**Fig. 4**, please also see below, and **Fig. S15-22**) to emphasize on epigenetic regulation of gene expression and the biological story as the reviewer kindly suggested.

Besides, we discussed more gene candidates, for example, **Fig. 2e** (please also see below) and **Fig. S7a** for our spatial-ATAC-RNA-seq; **Fig. 4b-d,h-k**, **Fig. S12a-c** and **Fig. S15a-b** for our spatial-CUT&Tag-RNA-seq (H3K27me3, H3K27ac, or H3K4me3, respectively) on P22 mouse brain with 100x100 barcodes. The corresponding discussions have been added in the corresponding sections according to the reviewer’s kind suggestion.

Furthermore, in order to constitute a unique resource of reference as the reviewer kindly mentioned, we have generated the webresources where our data can be browsed easily by the readers, one is UCSC Cell and Genome Browser (<https://cells-test.gi.ucsc.edu/?ds=mouse-brain-spatial-atac+spatial>), another is our own data portal generated with AtlasXplore (AtlasXomics) (<https://web.atlasxomics.com>, Id: reviewer Password: Reviewer@fanlab1). It allows for visualizing spatial pattern of genes or motifs, for example, by selecting a gene in the table of top ranked genes/motifs to show on the search bar to display (click the show function to view one by one). For ATAC and CUT&Tag data, the peaks, raw fragments per cluster, and gene model for the corresponding gene of interest can be viewed (click the Peak Viewer icon in the lower left panel). The motifs can be viewed by switching the icon Gene on the top right panel to Motif followed by selecting motif in the table to show on the search bar in a way similar to visualizing spatial gene expression map. The sequence logo can be displayed by clicking the Peak Viewer icon. The detailed tutorial of AtlasXplore for interactive analysis of spatial omics data is available at <https://docs.atlasxomics.com/projects/AtlasXplore/en/latest/>. The uploaded datasets will be available for browsing upon publication, and we are also building the CellXGene webresource, which we aim to release at the same time.

Fig. 2. e, Spatial mapping of GAS and gene expression for selected marker genes in different clusters for ATAC and RNA in spatial-ATAC-RNA-seq.

Fig. 4. Region specific epigenetic regulation of gene expression for spatial epigenome and transcriptome co-sequencing. **a**, Correlation of H3K27me3 CSS and RNA gene expression in corpus callosum. **b**, Correlation of H3K27ac GAS and RNA gene expression in corpus callosum. **c**, Correlation of H3K4me3 GAS and RNA gene expression in corpus callosum. **d**, Upset plot of H3K27me3 CSS and RNA gene expression in striatum, deeper and superficial cortical layer. Low CSS or gene expression (-), High CSS or gene expression (+). **e-g**, Venn diagrams showing the number of high (+) or low (-) CSS/GAS for different histone modifications in corpus callosum for common RNA marker genes. **h-k**, Spatial mapping of CSS, GAS, and gene expression for selected marker genes *Mag* (**h**), *Car2* (**i**), *Syt1* (**j**), *Gpr88* (**k**) in different clusters for ATAC, CUT&Tag and RNA in spatial-ATAC-RNA-seq and spatial-CUT&Tag-RNA-seq.

- Data analysis is well reported with availability of all scripts on a github page. In terms of providing credit to other work, there could be more comparison with Visium, however, I am fine if just a reference sentence distinguishing their technique from Visium is thrown in to the introduction.

Response: We thank the reviewer for the comments and suggestions. According to the reviewer's suggestion, we added "The 10x Visium platform also allows the co-detection of transcriptomics and proteins by either using immunofluorescence staining or polyadenylated antibody-derived tag-conjugated (ADT-conjugated) antibodies for proteins of interest before spatial transcriptome workflow¹¹." in the Introduction section.

11 Ben-Chetrit, N. *et al.* Integrated protein and transcriptome high-throughput spatial profiling. *bioRxiv*, 2022.2003.2015.484516, doi:10.1101/2022.03.15.484516 (2022).

- The abstract, introduction and conclusion are all clear and concise and the manuscript and data is presented well. In the results, I would like to see more interpretation/explanation in discrepancies seen in metrics in Figure 1, specifically the E13 data seems to often behave quite differently, do you think this can be explained by biology or is this a technical artifact?

Response: We appreciate the reviewer for this comment! The metrics in **Fig. 1b,c** and **Fig. S2e,f** (originally in Figure 1) includes Unique fragments, TSS fragments, FRiP (fraction of reads in peaks) and Mitochondria fragments for ATAC/CUT&Tag data and number of genes/UMIs per pixel for RNA data.

The discrepancies among metrics may mainly come from the tissues in our manuscript. The tissue types and tissue qualities may influence the metrics here. The tissue quality on epigenome and transcriptome are different among different tissue types. Besides, we are using fresh frozen tissues from different sources, the methods of tissue acquisition and conditions for tissue preservation will influence the tissue quality, especially the mRNA quality since mRNAs are quite easy to degrade.

Furthermore, according to the reviewer's suggestion, in order to figure out the reproducibility of our technologies within the same type of tissues, we added the experiments of biological replica using P21 mouse brain from different batches for both spatial-ATAC-RNA-seq and spatial-CUT&Tag(H3K27ac)-RNA-seq. The correlation analysis between replicates were shown in **Fig. S2c,d** (please also see below), which showed high reproducibility ($r=0.98$ for ATAC and $r=0.98$ for RNA in spatial-ATAC-RNA-seq; $r=0.96$ for CUT&Tag (H3K27ac) and $r=0.89$ for RNA in spatial-CUT&Tag-RNA-seq). The metrics of the replicas were also shown in **Fig. S2e,f** (the corresponding data are provided in **Table S1, S2**, please also see below). As can be seen from the results, there is good consistency for each technology with the same tissue types. This proved that the discrepancies were not led by technical artifacts. The **Fig. S2c,d,e,f**, and **Table S1,S2** were added in the revised manuscript and "Thus, the tissue types and quality may influence the metrics." was added in the data quality section.

Fig. S2 c, The reproducibility of spatial-ATAC-RNA-seq between biological replicates on ATAC data (left) and RNA data (right) for P21 mouse brain. **d**, The reproducibility of spatial-CUT&Tag-RNA-seq between biological replicates on CUT&Tag data (left) and RNA data (right) for P21 mouse brain.

Fig. S2 e, Comparison of number of unique fragments, TSS fragments, fraction of reads in peaks (FRIP), and fraction of mitochondrial fragments between biological replicates for spatial-ATAC-RNA-seq and spatial-CUT&Tag(H3K27ac)-RNA-seq. **f**, Gene and UMI count distribution between biological replicates for spatial-ATAC-RNA-seq and spatial-CUT&Tag(H3K27ac)-RNA-seq.

To further increase the mapping area, we developed a new device to perform in tissue barcoding of 100 x 100 pixels with a 20 μm pixel size, and performed spatial-ATAC-RNA-seq and spatial-CUT&Tag-RNA-seq (H3K27me3, H3K27ac or H3K4me3) on mouse postnatal day 22 (P22) brain (**Fig. 2a**, please also see below). This new device allows a unique combination of barcodes A_i and B_j ($i = 1-100$, $j = 1-100$) with a total of 10,000 barcoded pixels, which allowed to cover almost the entire mouse brain hemisphere at this stage. The new device also generated good metrics (**Fig. 1b,c** and **Table S1,S2**, please also see below) and the discussion about the new metrics were added in the data quality section (please also see below).

a

For the ATAC (spatial-ATAC-RNA-seq) or CUT&Tag (spatial-CUT&Tag-RNA-seq) data, we obtained a median of 14,284 (ATAC), 10,644 (H3K27me3), 10,002 (H3K27ac), and 2,507 (H3K4me3) unique fragments per pixel, of which 19% (ATAC), 12% (H3K27me3), 17% (H3K27ac), and 67% (H3K4me3) of fragments overlapped with TSS regions, and 26% (ATAC), 12% (H3K27me3), 21% (H3K27ac), and 54% (H3K4me3) located in peaks (**Fig. 1b,d** and **Table S1,S2**). The proportion of mitochondrial fragments was 0.2% (H3K27me3), 0.3% (H3K27ac), and 3.6% (H3K4me3) for spatial-CUT&Tag-RNA-seq or 4.6% for spatial-ATAC-RNA-seq. For the RNA portion of spatial-ATAC-RNA-seq and spatial-CUT&Tag-RNA-seq, totally 22,914 genes (spatial-ATAC-RNA-seq) or 25,881 (H3K27me3), 23,415 (H3K27ac), and 22,731 (H3K4me3) genes (spatial-CUT&Tag-RNA-seq) were detected with an average of 1,073 genes (spatial-ATAC-RNA-seq) and 2,011 (H3K27me3), 1,513 (H3K27ac), and 1,329 (H3K4me3) genes per pixel (spatial-CUT&Tag-RNA-seq) or 2,358 UMIs (spatial-ATAC-RNA-seq) and 4,734 (H3K27me3), 3,580 (H3K27ac), and 2,885 (H3K4me3) UMIs (spatial-CUT&Tag-RNA-seq) per pixel (**Fig. 1c** and **Table S1,S2**) in the mouse brain tissues.

Fig. 1. b, Comparison of number of unique fragments, TSS fragments, fraction of reads in peaks (FRIP), and fraction of mitochondrial fragments between spatial-ATAC-RNA-seq and spatial-CUT&Tag-RNA-seq. **c**, Gene and UMI (unique molecular identifier) count distribution between spatial-ATAC-RNA-seq and spatial-CUT&Tag-RNA-seq.

Table S1 Summary of metrics for ATAC and RNA in spatial-ATAC-RNA-seq for all the samples.

		E13 mouse embryo (50 barcodes, 50 μ m pixel size)	P22 mouse brain (100 barcodes, 20 μ m pixel size)	P21 mouse brain (replica 1, 50 barcodes, 20 μ m pixel size)	P21 mouse brain (replica 2, 50 barcodes, 20 μ m pixel size)	Human brain (50 barcodes, 50 μ m pixel size)	
Spatial-ATAC-RNA-seq	ATAC	Number of Unique fragments	18,079	14,284	10,857	14,385	9,898
		TSS fragments	16%	19%	20%	19%	15%
		FRiP	11%	26%	24%	26%	11%
		Mitochondrial fragments	0.96%	4.6%	9%	8.4%	20%
	RNA	Number of genes per pixel	1,255	1,073	1,005	1,600	1,200
		Number of UMIs per pixel	3,603	2,358	2,391	3,811	2,809
		Number of unique genes present	20,900	22,914	19,859	20,046	29,293
		Pixels on tissue	2,187	9,215	2,373	2,498	2,500

Table S2 Summary of metrics for CUT&Tag and RNA in spatial-CUT&Tag-RNA-seq for all the samples.

		P22 mouse brain (H3K27me3) (100 barcodes, 20 μ m pixel size)	P22 mouse brain (H3K27ac) (100 barcodes, 20 μ m pixel size)	P22 mouse brain (H3K4me3) (100 barcodes, 20 μ m pixel size)	P21 mouse brain (H3K27ac, replica 1) (50 barcodes, 20 μ m pixel size)	P21 mouse brain (H3K27ac, replica 2) (50 barcodes, 20 μ m pixel size)
Spatial-CUT&Tag-RNA-seq						

	CUT&Tag	Number of Unique fragments	10,644	10,002	2,507	4,756	5,022
		TSS fragments	12%	17%	67%	19%	20%
		FRiP	12%	21%	54%	19%	18%
		Mitochondrial fragments	0.2%	0.3%	3.6%	0.1%	0.02%
	RNA	Number of genes per pixel	2,011	1,513	1,329	1,145	752
		Number of UMIs per pixel	4,734	3,580	2,885	2,938	1,890
		Number of unique genes present	25,881	23,415	22,731	19,831	18,718
		Pixels on tissue	9,752	9,370	9,548	2,387	2,499

- Altogether, the technique seems great, the analysis seems solid and writing is clear, just more emphasis on some gene candidates could make this manuscript more interesting.

Response: We appreciate the reviewer for the positive comments and the great suggestion! According to the reviewer's comments, we discussed more genes and focused more on the biological story in the revised manuscript. As we discussed in our responses to reviewer's previous comment, we added a new section "Spatial multi-omics deciphers region-specific epigenetic regulation of gene expression and cooperation in mouse brain" to emphasize on discussion of epigenetic regulation of gene expression genome-wide and the biological story according to the reviewer's suggestion (**Fig. 4** and **Fig. S15-22**). Besides, we discussed more gene candidates, for example, **Fig. 2e** and **Fig. S7a** for our spatial-ATAC-RNA-seq; **Fig. 4b-d,h-k**, **Fig. S12a-c** and **Fig. S15a-b** for our spatial-CUT&Tag-RNA-seq (H3K27me3, H3K27ac, or H3K4me3, respectively) on P22 mouse brain with 100x100 barcodes. The corresponding discussions have been added in the corresponding sections. In addition, the generated webresources could also help the readers browse our data easily, and we are also actively improving the webresources according to the reviewer's suggestions.

Reviewer Reports on the First Revision:

Referees' comments:

Referee #1 (Remarks to the Author):

The authors have addressed the issues raised with extensive additional data and more meaningful analyses, and I support publication.

Referee #2 (Remarks to the Author):

I commend the authors for doing an excellent job addressing all the issues raised in my (and possibly other) previous review. The new analyses included in this revision have clearly demonstrated high data quality. The two spatial epigenomics/transcriptomics technologies by Prof. Fan and colleagues provide new powerful tools for biologists to characterize the spatial heterogeneity and to investigate the underlying gene regulatory mechanisms, as demonstrated by their integrated analyses of mouse and human brains.

Referee #3 (Remarks to the Author):

I do appreciate how the authors addressed my comments. I was satisfied by all responses , except for the following:

I would still like the authors to specifically address the technical issues which may be setting the E13 tissue apart, and maybe append this to the sentence about reproducibility (which I do recognize as a great improvement)

The new biological storyline section outlining progenitor to post-mitotic and oligodendrocyte states is exactly what I was looking for, thank you!

I am also very impressed by the accessibility offered by the atlasxomics website, and think this is wonderful and deserves to be better recognized.

Altogether, my comments were satisfactorily addressed and I believe this manuscript is great product of extensive interdisciplinary collaboration.

Author Rebuttals to First Revision:

We greatly appreciate the reviewers for all the constructive review comments, which contributed to a major improvement of our manuscript. We have fully addressed all the concerns from the reviewers. Following please find our point-by-point responses to all review comments. Thank you so much!

Referee #1 (Remarks to the Author):

The authors have addressed the issues raised with extensive additional data and more meaningful analyses, and I support publication.

Response: We would like to thank the reviewer for the positive feedback regarding our work and manuscript!

Referee #2 (Remarks to the Author):

I commend the authors for doing an excellent job addressing all the issues raised in my (and possibly other) previous review. The new analyses included in this revision have clearly demonstrated high data quality. The two spatial epigenomics/transcriptomics technologies by Prof. Fan and colleagues provide new powerful tools for biologists to characterize the spatial heterogeneity and to investigate the underlying gene regulatory mechanisms, as demonstrated by their integrated analyses of mouse and human brains.

Response: We really appreciate the reviewer for the positive feedback regarding our technologies and manuscript!

Referee #3 (Remarks to the Author):

- I do appreciate how the authors addressed my comments. I was satisfied by all responses, except for the following:

I would still like the authors to specifically address the technical issues which may be setting the E13 tissue apart, and maybe append this to the sentence about reproducibility (which I do recognize as a great improvement)

Response: We thank the reviewer for the comment! As we mentioned in our previous response, the tissue types and tissue quality may influence the metrics here.

The E13 mouse embryo is a heterogenous tissue type. The region we covered for this tissue (Fig. 1d) includes several embryonic organs with distinct cell types, for instance, brain, eye, limb, spine, etc. So the metrics might be difference.

Also, besides the influence of tissue quality from tissue acquisition and preservation, the methods of tissue preparation might also influence the metrics. We used fresh frozen tissues from different sources, and they may have different tissue preparation protocols, which could affect the quality of the tissue sections. According to the reviewer's suggestions, we added the details of tissue preparation for all the tissues in the Methods section (please also see below), and the sentence "Tissue type, preparation, and quality may influence analytical metrics (**Methods**)." has been added in the reproducibility section of our manuscript.

The tissue preparation for the E13 mouse embryo:

Mouse C57 Embryo Sagittal Frozen Sections (MF-104-13-C57) were purchased from Zyagen (San Diego, CA). The freshly harvested E13 mouse embryos were snap-frozen in OCT blocks and sectioned with 7–10 μm thickness. The tissue sections were collected on poly-L-lysine coated glass slides.

The tissue preparation for the juvenile mouse brain:

Juvenile mouse brain tissue (P21–P22) was obtained from *Sox10:Cre-RCE:LoxP* (EGFP) line on a C57BL/6xCD1 mixed genetic background maintained at Karolinska Institutet. Mice were sacrificed at P21/P22 by anesthesia with ketamine (120 mg/kg of body weight) and xylazine (14 mg/kg of body weight), followed by transcranial perfusion with cold oxygenated artificial cerebrospinal fluid aCSF (87 mM NaCl, 2.5 mM KCl, 1.25 mM NaH_2PO_4 , 26 mM NaHCO_3 , 75 mM Sucrose, 20 mM Glucose, 1 mM $\text{CaCl}_2 \cdot 2\text{H}_2\text{O}$ and 2 mM $\text{MgSO}_4 \cdot 7\text{H}_2\text{O}$ in dH_2O). Upon isolation, the brains were kept for a minimal time period in aCSF until embedding in Tissue-Tek® O.C.T. compound (Sakura) and snap-freezing using a mixture of dry ice and ethanol. Coronal cryosections of 10 μm were mounted on poly-L-lysine coated glass slides.

The tissue preparation for the human brain:

The human brain tissue was obtained from the Brain Collection of the New York State Psychiatric Institute (NYSPI) at Columbia University. The anterior hippocampal region was dissected from a fresh

frozen coronal section (20 μ m thickness) of the right brain hemisphere. The dentate gyrus region (around 10 mm x 10 mm) of the anterior hippocampal region was selected. The cryosections of 10 μ m were collected on poly-L-lysine coated glass slides.

- The new biological storyline section outlining progenitor to post-mitotic and oligodendrocyte states is exactly what I was looking for, thank you!

I am also very impressed by the accessibility offered by the atlasxomics website, and think this is wonderful and deserves to be better recognized.

Altogether, my comments were satisfactorily addressed and I believe this manuscript is great product of extensive interdisciplinary collaboration.

Response: We are grateful to the reviewer for the positive feedback regarding our work and manuscript!